# Federated TD Learning with Linear Function Approximation under Environmental Heterogeneity

**Han Wang**                                                        *hw2786@columbia.edu*
*Department of Electrical Engineering*
*Columbia University*

**Aritra Mitra**                                                    *amitra2@ncsu.edu*
*Department of Electrical and Computer Engineering*
*North Carolina State University*

**Hamed Hassani**                                                   *hassani@seas.upenn.edu*
*Department of Electrical and Systems Engineering*
*University of Pennsylvania*

**George J. Pappas**                                                *pappasg@seas.upenn.edu*
*Department of Electrical and Systems Engineering*
*University of Pennsylvania*

**James Anderson**                                                  *james.anderson@columbia.edu*
*Department of Electrical Engineering*
*Columbia University*

**Reviewed on OpenReview:** *https://openreview.net/forum?id=hdQspgyFrk*

## Abstract

We initiate the study of federated reinforcement learning under environmental heterogeneity by considering a policy evaluation problem. Our setup involves $N$ agents interacting with environments that share the same state and action space but differ in their reward functions and state transition kernels. Assuming agents can communicate via a central server, we ask: *Does exchanging information expedite the process of evaluating a common policy?* To answer this question, we provide the first comprehensive finite-time analysis of a federated temporal difference (TD) learning algorithm with linear function approximation, while accounting for Markovian sampling, heterogeneity in the agents' environments, and multiple local updates to save communication. Our analysis crucially relies on several novel ingredients: (i) deriving perturbation bounds on TD fixed points as a function of the heterogeneity in the agents' underlying Markov decision processes (MDPs); (ii) introducing a virtual MDP to closely approximate the dynamics of the federated TD algorithm; and (iii) using the virtual MDP to make explicit connections to federated optimization. Putting these pieces together, we prove that in a low-heterogeneity regime, exchanging model estimates leads to linear convergence speedups in the number of agents. Our theoretical contribution is significant in that it is the first result of its kind in multi-agent/federated reinforcement learning that complements the numerous analogous results in heterogeneous federated optimization.

## 1 Introduction

In the popular federated learning (FL) paradigm (Konečnỳ et al., 2016; McMahan et al., 2017), a set of agents aim to find a common statistical model that explains their collective observations. The motivation to collaborate stems from the fact that if the underlying distributions generating the agents' observations are "similar", then each agent can end up learning a "better" model than if it otherwise used just its own

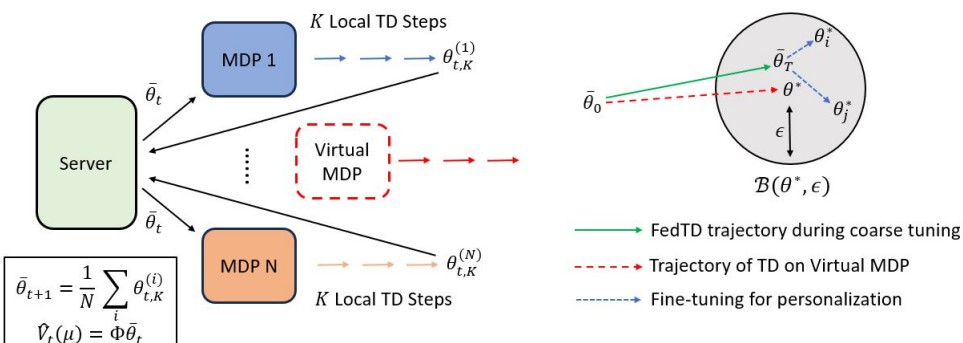

Figure 1: (**Left**) Illustration of how `FedTD(0)` works. Each agent performs $K$ local TD update steps on its own MDP, and transmits its updated model to a server. The virtual MDP serves to approximate the dynamics of `FedTD(0)`. The global model $\bar{\theta}_t$ at the server is used to construct a linearly parameterized approximation of the value function associated with a policy $\mu$. (**Right**) `FedTD(0)` helps each agent converge *quickly* to a ball $\mathcal{B}(\theta^*, \epsilon)$ centered around the optimal parameter $\theta^*$ of the virtual MDP. Here, $\epsilon$ captures the heterogeneity in the agents' MDPs. Using the output $\bar{\theta}_T$ of `FedTD(0)`, each agent $i$ can then fine-tune based on its own data to converge *exactly* to its own optimal parameter $\theta_i^*$.

data. This idea has been formalized by the canonical FL algorithm `FedAvg` (and its many variants) where agents communicate local models via a central server while keeping their raw data private. To achieve communication-efficiency - a key consideration in FL - the agents perform multiple local model-updates between successive communication rounds. There is a rich literature that analyzes the performance of `FedAvg`, focusing primarily on the aspect of *statistical heterogeneity* that originates from differences in the agents' underlying data distributions (Sahu et al., 2018; Khaled et al., 2019; 2020; Li et al., 2019; Koloskova et al., 2020; Woodworth et al., 2020b; Malinovskiy et al., 2020; Pathak & Wainwright, 2020; Wang et al., 2020; Karimireddy et al., 2020b; Acar et al., 2021; Gorbunov et al., 2021; Mitra et al., 2021; Mishchenko et al., 2022). Notably, the above works focus on supervised learning problems that are modeled within the framework of distributed optimization. However, for sequential decision-making with multiple agents interacting with *potentially different environments*, little to nothing is known about the effect of heterogeneity. This is the gap we seek to fill with our work.

The recent survey paper (Qi et al., 2021) describes a federated reinforcement learning (FRL) framework which incorporates some of the key ideas from FL into reinforcement learning (RL); applications of FRL in robotics (Liu et al., 2019), autonomous driving (Chen et al., 2015), and edge computing (Wang et al., 2019) are discussed in detail in this paper. As RL algorithms often require many samples to achieve acceptable accuracy, FRL aims to achieve *sample-efficiency* by leveraging information from multiple agents interacting with similar environments. Importantly, as in standard FL, the FRL framework requires agents to keep their raw data (e.g., rewards, states, and actions) private, and adhere to stringent communication constraints.

**Motivation and Scope of this Work.** While FRL is a promising idea, in reality, it will rarely be the case that different agents end up interacting with *exactly the same* environment. Unfortunately, this is the running assumption in almost all multi-agent RL (MARL) and FRL works (Doan et al., 2019; Liu & Olshevsky, 2021a; Khodadadian et al., 2022; Shen et al., 2023). Departing from this somewhat unrealistic yet prevalent assumption, **the main motivation of this paper is to build a systematic theoretical framework for reasoning about what to expect when one mixes information from non-identical Markov processes**. The nature of this question is fundamental, and while we motivate its study from the perspective of FRL[1], it can just as easily be connected to stochastic control and estimation problems where one seeks to "fuse" data generated from non-identical dynamical systems with noisy inputs (Wang et al., 2022b; Guo et al., 2023; Xin et al., 2023).

---

[1]Just as statistical heterogeneity is a major challenge in FL, *environmental heterogeneity* is identified as a key open challenge in FRL (Qi et al., 2021).

To initiate a principled study of heterogeneity in FRL, we focus on the simplest RL problem, namely *policy evaluation.* Our setup involves $N$ agents where each agent interacts with an environment modeled as a MDP. The agents' MDPs share the same state and action space but have different reward functions and state transition kernels, thereby capturing environmental heterogeneity. Each agent seeks to compute the discounted cumulative reward (value function) associated with a common policy $\mu$. Notably, the value functions induced by $\mu$ may differ across environments. This leads to the central question we investigate: *Can an agent expedite the process of learning its own value function by leveraging information from potentially different MDPs?* As we explain shortly, this is a non-trivial question to answer even for policy evaluation; hence, our focus on policy evaluation as a starting point. That said, recent works have shown that with minor modifications to the analysis of TD learning for policy evaluation (Srikant & Ying, 2019), one can analyze Q-learning for control (Chen et al., 2019). As such, we envision that the developments in this paper can be suitably extended to control algorithms like Q-learning as well.

A typical application of the above FRL setup is that of an autonomous driving system where vehicles in different geographical locations share local models capturing their learned experiences to train a shared model that benefits from the collective exploration data of all vehicles. The vehicles (agents) essentially have the same operations (e.g., steering, braking, accelerating, etc.), but can be exposed to different environments (e.g., road and weather conditions, routes, driving regulations etc.).

## 1.1   Our Contributions

We study a federated version of the temporal difference (TD) learning algorithm `TD`(0) (Sutton, 1988). The structure of this algorithm, which we call `FedTD`(0), is as follows. At each iteration, each agent plays an action according to the policy $\mu$, observes a reward, and transitions to a new state based on its *own* MDP. It then uses `TD`(0) with linear function approximation to update a local model that approximates its own value function. To benefit from other agents' data in a communication-efficient manner, each agent periodically synchronizes with a central server, and performs multiple local model-updates in between - as depicted in Figure 1. Notably, as in FL, agents only exchange models but never their personal observations. We perform a comprehensive analysis of `FedTD`(0) under environmental heterogeneity, and make the following contributions:

1. **Effect of heterogeneity on TD(0) fixed points.** Towards understanding the behavior of `FedTD`(0), we start by asking: *How does heterogeneity in the transition kernels and reward functions of MDPs manifest into differences in the long-term behavior of TD(0) (with linear function approximation) on such MDPs?* Theorem 1 provides an answer by characterizing how perturbing a MDP perturbs the `TD`(0) fixed point for that MDP. To arrive at this result, we combine results from the perturbation theories of Markov chains and linear equations. Theorem 1 establishes the first perturbation result for `TD`(0) fixed points, and complements results of a similar flavor in the RL literature, such as the *Simulation Lemma* due to Kearns & Singh (2002). As such, Theorem 1 can serve as a tool of independent interest in RL.

2. **The Virtual MDP framework.** In FL algorithms such as `FedAvg`, the average of the negative gradients of the agents' loss functions drives the iterates of `FedAvg` towards the minimizer of a global loss function. In our setting, there is no such global loss function. *So by averaging TD(0) update directions of different MDPs, where do we end up?* To answer this question, we construct a virtual MDP in Section 3.1, and characterize several important properties of this fictitious MDP that aid our subsequent analysis. Along the way, we derive a simple yet key result (Proposition 1) pertaining to convex combinations of Markov matrices associated with aperiodic and irreducible Markov chains. This result appears to be new, and may be of independent interest.

3. **Linear Speedup under Markovian Sampling and Heteroegenity.** Our most significant contribution is to provide the *first analysis of a federated RL algorithm, FedTD(0), that simultaneously accounts for linear function approximation, Markovian sampling, multiple local updates, and heterogeneity.* In Theorem 2, we prove that after $T$ communication rounds with $K$ local model-updating steps per round, `FedTD`(0) guarantees convergence at a rate of $\tilde{O}(1/NKT)$ to a neighborhood of each agent's optimal parameter. The size of the neighborhood depends on the level of heterogeneity in the

agents' MDPs. *The key implication of this result is that in a low-heterogeneity regime, each agent can enjoy an N-fold linear speed-up in convergence via collaboration, and converge quickly to a vicinity of its own optimal parameter.* One can view this as a "coarse tuning phase". As is typically done in FL (Collins et al., 2022), each agent can use the solution of FedTD(0) to then fine-tune (personalize) based on its own data. This is visually illustrated in Figure 1. Theorem 2 is significant in that it is the first result in FRL that complements the myriad of federated optimization results that account for the effects of heterogeneity (Sahu et al., 2018; Khaled et al., 2019; 2020; Li et al., 2019; Koloskova et al., 2020; Woodworth et al., 2020b; Malinovskiy et al., 2020; Pathak & Wainwright, 2020; Wang et al., 2020; Karimireddy et al., 2020b; Acar et al., 2021; Gorbunov et al., 2021; Mitra et al., 2021; Mishchenko et al., 2022).

4. **Novel Proof Framework.** One might be tempted to think that the proof of Theorem 2 is a simple combination of the standard FedAvg analysis with that of TD learning. We briefly explain here why this isn't quite the case, and defer a more elaborate explanation to Section 5.1. First, in the centralized TD analysis (Bhandari et al., 2018; Srikant & Ying, 2019), and in the existing analysis for MARL/FRL (Doan et al., 2019; Liu & Olshevsky, 2021a; Khodadadian et al., 2022) with identical MDPs, the dynamics of the update rules correspond to one single MDP. *In our setup, the dynamics of* FedTD(0) *may not correspond to any MDP at all!* Thus, we need new tools relative to existing RL analyses. Second, while existing FL analyses are essentially distributed optimization proofs, *federated TD learning does not correspond to minimizing any fixed loss function.* Moreover, unlike the i.i.d. data model in FL, the data tuples observed by each agent in FedTD(0) are part of a single Markovian trajectory. This creates complex time-correlations that are challenging to deal with even in a single-agent setting. Thus, we cannot directly employ FL proofs either. As such, we introduce a new analysis framework where we argue that the dynamics of FedTD(0) can be approximated by that of TD(0) on a virtual MDP, up to an error term that captures heterogeneity in the agents' MDPs. Carefully tracking how this error term propagates over time accounts for the effect of heterogeneity; establishing linear speedup under Markovian sampling and local steps requires much more work.

5. **Bias introduced by Heterogeneity.** Our convergence result in Theorem 2 features a bias term due to heterogeneity that cannot be eliminated even by making the step-size arbitrarily small. *Is such a term unavoidable?* We explore this question in Theorem 3 by studying a "steady-state" deterministic version of FedTD(0). Even for this simple case, we prove that a bias term depending on a natural measure of heterogeneity shows up *inevitably* in the long-term dynamics of FedTD(0). This result sheds further light on the effect of heterogeneity in FRL.

## 1.2 Related Work

In what follows, we discuss the most relevant threads of literature.

1. **Finite-Time Analysis of TD Learning Algorithms.** In their seminal paper, Tsitsiklis & Van Roy (1997) provided an asymptotic convergence analysis of the temporal difference (TD) learning algorithm (Sutton, 1988; Sutton et al., 1998) with value function approximation, using tools from stochastic approximation theory. Several years later, the work by Korda & La (2015) provided finite-time rates for TD learning. However, the authors in Narayanan & Szepesvári (2017) noted some issues with the proofs in Korda & La (2015). Under the i.i.d. observation model described in Section 5, Dalal et al. (2018) and Lakshminarayanan & Szepesvári (2017) were able to resolve the issues in Korda & La (2015). Even so, a non-asymptotic convergence analysis for the challenging Markovian setting (that we consider in this paper) remained elusive till the work by Bhandari et al. (2018). While the authors in Bhandari et al. (2018) made some elegant connections between the dynamics of TD learning and gradient descent, an alternative proof technique using Stein's method was developed by Srikant & Ying (2019). Yet another interesting interpretation was provided by Liu & Olshevsky (2021b): they argued that the steady-state temporal difference direction acts as a "gradient-splitting" of an appropriately chosen function. Recently, a short proof of TD learning with linear function approximation and more general nonlinear contractive stochastic approximation schemes was provided by Mitra (2024) based on a novel inductive proof technique. While all the

above works provide upper-bounds for the task of policy evaluation, for minimax lower bounds, we refer the reader to the work of Khamaru et al. (2021).

2. **Multi-Agent and Federated RL.** In Doan et al. (2019) and Liu & Olshevsky (2021a), the authors analyze multi-agent TD learning with linear function approximation over peer-to-peer networks. Neither approach accounts for local steps or Markovian sampling. In Shen et al. (2023), the authors study a parallel version of asynchronous actor-critic algorithms, and establish a linear speedup result - albeit under an i.i.d. sampling assumption. Very recently, the authors in Khodadadian et al. (2022) and Dal Fabbro et al. (2023) studied the effect of Markovian sampling for federated TD learning. However, all of the above papers consider a *homogeneous setting with identical* MDPs for all agents. In contrast, our work has to tackle the challenge of understanding *the long-term effects of mixing TD update directions from non-identical MDPs.* The only two papers we are aware of that perform any theoretical analysis of heterogeneity in FRL are Jin et al. (2022) and Xie & Song (2023). However, their analyses are limited to the much simpler tabular setting with no function approximation. In particular, the work of Xie & Song (2023) only comes with asymptotic results, i.e., they do not provide finite-time rates. Moreover, unlike us, neither Jin et al. (2022) nor Xie & Song (2023) provide any explicit linear speedup result. In conclusion, we are the first to establish a **finite-time theory** for FRL under function approximation, environmental heterogeneity, and Markovian sampling. Considering different settings, Zhang et al. (2024) proposed the `FEDSARSA` algorithm to solve the on-policy FRL problem and Wang et al. (2023) proposed `FedLQR` to solve the federated control design problem. A more detailed description of related work on federated learning is relegated to the Appendix.

## 2 Model and Problem Formulation

We consider a Markov Decision Process (MDP) (Sutton et al., 1998) defined by the tuple $\mathcal{M} = \langle \mathcal{S}, \mathcal{A}, \mathcal{R}, \mathcal{P}, \gamma \rangle$, where $\mathcal{S}$ is a finite state space of size $n$, $\mathcal{A}$ is a finite action space, $\mathcal{P}$ is a set of action-dependent Markov transition kernels, $\mathcal{R}$ is a reward function, and $\gamma \in (0, 1)$ is the discount factor. We consider the problem of evaluating the value function $V_\mu$ of a given policy $\mu$, where $\mu : \mathcal{S} \to \mathcal{A}$. The policy $\mu$ induces a Markov reward process (MRP) characterized by a transition matrix $P_\mu$, and a reward function $R_\mu$. Under the action of the policy $\mu$ at an initial state $s$, $P_\mu(s, s')$ is the probability of transitioning from state $s$ to state $s'$, and $R_\mu(s)$ is the expected instantaneous reward. The discounted expected cumulative reward obtained by playing policy $\mu$ starting from initial state $s$ is:

$$V_\mu(s) = \mathbb{E}\left[\sum_{t=0}^{\infty} \gamma^t R_\mu(s_t) | s_0 = s\right],$$

where $s_t$ is the state of the Markov chain at time $t$. From Tsitsiklis & Van Roy (1997), we know that $V_\mu$ is the fixed point of the policy-specific Bellman operator $T_\mu : \mathbb{R}^n \to \mathbb{R}^n$, i.e., $T_\mu V_\mu = V_\mu$, where for any $V \in \mathbb{R}^n$,

$$(T_\mu V)(s) = R_\mu(s) + \gamma \sum_{s' \in \mathcal{S}} P_\mu(s, s') V(s'), \ \forall s \in \mathcal{S}.$$

**TD learning with linear function approximation.** We consider the setting where the number of states is very large, making it practically infeasible to compute the value function $V_\mu$ directly. To mitigate the curse of dimensionality, a common approach (Sutton et al., 1998) is to consider a low-dimensional linear function approximation of the value function $V_\mu$. Let $\{\Phi_k\}_{k=1}^{d}$ be a set of $d$ linearly independent basis vectors in $\mathbb{R}^n$, and $\Phi \in \mathbb{R}^{n \times d}$ be a matrix with these basis vectors as its columns, i.e., the $k$-th column of $\Phi$ is $\Phi_k$. A parametric approximation $\hat{V}_\theta$ of $V_\mu$ in the span of $\{\Phi_k\}_{k=1}^{d}$ is then given by $\hat{V}_\theta = \Phi\theta$, where $\theta \in \mathbb{R}^d$ is a parameter vector to be learned. Notably, this is tractable since $d \ll n$. We denote the $s$-th row of $\Phi$ by $\phi(s) \in \mathbb{R}^d$, and refer to it as the fixed feature vector corresponding to state $s$. We write $\hat{V}_\theta(s) = \phi(s)^\top \theta$ and make the standard assumption (Bhandari et al., 2018) that $\|\phi(s)\|^2 \leq 1, \forall s \in \mathcal{S}$.

The objective is to find the best linear approximation of $V_\mu$ in the span of $\{\Phi_k\}_{k=1}^{d}$. More precisely, we seek a parameter vector $\theta^*$ that minimizes the distance between $\hat{V}_\theta$ and $V_\mu$ (in a suitable sense). When the underlying

MDP is *unknown*, one of the most popular techniques to achieve this goal is the classical TD(0) algorithm. TD(0) starts from an initial guess $\theta_0 \in \mathbb{R}^d$. Subsequently, at the $t$-th iteration, upon playing the given policy $\mu$, a new data tuple $O_t = (s_t, r_t = R_\mu(s_t), s_{t+1})$ comprising of the current state, the instantaneous reward, and the next state is observed. Let us define the TD(0) update direction as $g_t(\theta_t) \triangleq \left( r_t + \gamma \phi(s_{t+1})^\top \theta_t - \phi(s_t)^\top \theta_t \right) \phi(s_t)$. Using a step-size $\alpha_t \in (0, 1)$, the parameter $\theta_t$ is then updated as

$$\theta_{t+1} = \theta_t + \alpha_t g_t(\theta_t).$$

Under some mild technical assumptions, it was shown in Tsitsiklis & Van Roy (1997) that the TD(0) iterates converge asymptotically almost surely to a vector $\theta^*$, where $\theta^*$ is the unique solution of the projected Bellman equation $\Pi_D T_\mu(\Phi\theta^*) = \Phi\theta^*$. Here, $D$ is a diagonal matrix with entries given by the elements of the stationary distribution $\pi$ of the Markov matrix $P_\mu$. Furthermore, $\Pi_D(\cdot)$ is the projection operator onto the subspace spanned by $\{\phi_k\}_{k=1}^d$ with respect to the inner product $\langle \cdot, \cdot \rangle_D$.[2]

**Objective.** We study a multi-agent RL problem where agents interact with similar, but *non-identical* MDPs that share the same state and action space. All agents seek to evaluate the same policy. Our goal is to understand: *Can an agent evaluate the value function of its own MDP in a more sample-efficient way by leveraging data from other agents?* Existing FL analyses that study statistical heterogeneity in supervised learning/empirical risk minimization fall short of answering this question, since *our problem does not involve minimizing a static loss function.* As such, the question we have posed above is non-trivial, and requires several new ideas and tools. In the next section, we will start building these tools in a systematic manner by accomplishing the following goals.

**Goal 1.** Formally defining what we mean by model heterogeneity in the agents' MDPs.

**Goal 2.** Characterizing how such model heterogeneity translates to differences in the *fixed points* of the TD(0) algorithm when run on the agents' MDPs.

**Goal 3.** Introducing the notion of a virtual MDP that will play a crucial role in reasoning about the *long-term* behavior of algorithms that combine information from non-identical MDPs.

## 3 Heterogeneous Federated RL

We consider a federated RL setting comprising of $N$ agents that interact with potentially different environments. Agent $i$'s environment is characterized by the following MDP: $\mathcal{M}^{(i)} = \langle \mathcal{S}, \mathcal{A}, \mathcal{R}^{(i)}, \mathcal{P}^{(i)}, \gamma \rangle$. While all agents share the same state and action space, the reward functions and state transition kernels of their environments can differ. We focus on a policy evaluation problem where all agents seek to evaluate a common policy $\mu$ that induces $N$ Markov reward processes characterized by the tuples $\{P_\mu^{(i)}, R_\mu^{(i)}\}_{i \in [N]}$.[3] Agent $i$ aims to find a linearly parameterized approximation of its *own* value function $V_\mu^{(i)}$. Trivially, agent $i$ can do so without interacting with any other agent by simply running TD(0). However, the key question we ask pertains to the **value of side-information**: *By using data from other agents, can it achieve a desired level of approximation with fewer samples relative to when it acts alone?* Naturally, the answer to the above question depends on the level of heterogeneity in the agents' MDPs. Accordingly, we introduce the following definitions.

**Assumption 1.** *(Markov Kernel Heterogeneity) There exists an $\epsilon > 0$ such that for all agents $i, j \in [N]$, it holds that $|P^{(i)}(s, s') - P^{(j)}(s, s')| \leq \epsilon |P^{(i)}(s, s')|, \forall s, s' \in \mathcal{S}$. Here, for each $i \in [N]$, $P^{(i)}(s, s')$ represents the $(s, s')$-th element of the matrix $P^{(i)}$.*

**Assumption 2.** *(Reward Heterogeneity) There exists an $\epsilon_1 > 0$ such that for all $i, j \in [N]$, it holds that $\|R^{(i)} - R^{(j)}\| \leq \epsilon_1$.*

Clearly, smaller values of $\epsilon$ and $\epsilon_1$ capture more similarity in the agents' MDPs. Suppose all agents can communicate via a central server. Via such communication, the standard FL task is to find one common

---

[2]We will use $\|\cdot\|_D$ to denote the norm induced by the matrix $D$, and $\|\cdot\|$ to represent the standard Euclidean norm for vectors and $\ell_2$ induced norm for matrices.

[3]We will henceforth drop the dependence of $P^{(i)}$ and $R^{(i)}$ on the policy $\mu$.

model that "fits" the data of all agents. In a similar spirit, our goal is to find a common parameter $\theta$ such that $\hat{V}_\theta = \Phi\theta$ approximates each $V_\mu^{(i)}, i \in [N]$. The role of this common $\theta$ will be to quickly (i.e., by leveraging samples of *all* agents) provide a coarse model that the agents can then use as a warm-start to fine-tune based on personal data. There is a natural tension here. While federation can help converge *faster* to a coarse model, such a model may not *accurately* capture the value function of *any* agent if the agents' MDPs are very dissimilar. *So does more data help or hurt?*

**Impact of Heterogeneity on TD fixed points.** To answer the above question, we need to carefully understand how the structural heterogeneity assumptions on the MDPs (namely, Assumptions 1 and 2) manifest into differences in the long-term dynamics of TD(0) on these MDPs. Since long-term dynamics are intimately tied to fixed points, we first set out to characterize the "closeness" in TD(0) fixed points across different MDPs. To proceed, we make the following standard assumption.

**Assumption 3.** *For each $i \in [N]$, the Markov chain induced by the policy $\mu$, corresponding to the state transition matrix $P^{(i)}$, is aperiodic and irreducible.*

The above assumption implies the existence of a unique stationary distribution $\pi^{(i)}$ for each $i \in [N]$; let $D^{(i)}$ be a diagonal matrix with the entries of $\pi^{(i)}$ on its diagonal. For each agent $i$, we then use $\theta_i^*$ to denote the solution of the projected Bellman equation $\Pi_{D^{(i)}} T_\mu^{(i)}(\Phi\theta_i^*) = \Phi\theta_i^*$ for agent $i$. In words, $\theta_i^*$ is the best linear approximation of $V_\mu^{(i)}$ in the span of $\{\phi_k\}_{k=1}^d$. From Section 2, we know that the iterates of TD(0) on agent $i$'s MRP will converge to $\theta_i^*$ asymptotically almost surely. Our goal is to bound the gap $\|\theta_i^* - \theta_j^*\|$ as a function of the heterogeneity parameters $\epsilon$ and $\epsilon_1$ appearing in Assumptions 1 and 2. The key observation we will exploit is that for each $i \in [N]$, $\theta_i^*$ is the unique solution of the linear equation $\bar{A}_i \theta_i^* = \bar{b}_i$, where $\bar{A}_i = \Phi^\top D^{(i)}(\Phi - \gamma P^{(i)}\Phi)$ and $\bar{b}_i = \Phi^\top D^{(i)} R^{(i)}$. For an agent $j \neq i$, viewing $\bar{A}_j$ and $\bar{b}_j$ as perturbed versions of $\bar{A}_i$ and $\bar{b}_i$, we can now appeal to results from the perturbation theory of linear equations (Horn & Johnson, 2012a, Chapter 5.8) to bound $\|\theta_i^* - \theta_j^*\|$. To that end, we first recall a result from the perturbation theory of Markov chains (O'cinneide, 1993) which shows that under Assumption 1, the stationary distributions $\pi^{(i)}$ and $\pi^{(j)}$ are close for any pair $i, j \in [N]$.

**Lemma 1.** *(Perturbation bound on Stationary Distributions) Suppose Assumption 1 holds. Then, for any pair of agents $i, j \in [N]$, the stationary distributions $\pi^{(i)}$ and $\pi^{(j)}$ satisfy:*

$$\|\pi^{(i)} - \pi^{(j)}\|_1 \leq 2(n-1)\epsilon + \mathcal{O}(\epsilon^2). \tag{1}$$

We will now use the above result to bound $\|\bar{A}_i - \bar{A}_j\|$ and $\|\bar{b}_i - \bar{b}_j\|$. To state our results, we make the standard assumption that for each $i \in [N]$, it holds that $|R^{(i)}(s)| \leq R_{\max}, \forall s \in \mathcal{S}$, i.e., the rewards are uniformly bounded. In (Tsitsiklis & Van Roy, 1997), it was shown that $-\bar{A}_i$ is a negative definite matrix; thus, $\exists \delta_1 > 0$ such that $\|\bar{A}_i\| \geq \delta_1, \forall i \in [N]$. We also assume that $\exists \delta_2 > 0$ such that $\|\bar{b}_i\| \geq \delta_2, \forall i \in [N]$. In our first technical result, stated below, we provide a bound on the perturbation of TD fixed points.

**Theorem 1.** *(Perturbation bounds on TD(0) fixed points) For all $i, j \in [N]$, we have:*

1. $\|\bar{A}_i - \bar{A}_j\| \leq A(\epsilon) \triangleq \gamma\sqrt{n}\epsilon + (1+\gamma)\left(2(n-1)\epsilon + \mathcal{O}(\epsilon^2)\right).$

2. $\|\bar{b}_i - \bar{b}_j\| \leq b(\epsilon, \epsilon_1) \triangleq R_{\max}\left(2(n-1)\epsilon + \mathcal{O}(\epsilon^2)\right) + \mathcal{O}(\epsilon_1).$

3. *Suppose $\exists H > 0$ s.t. $\|\theta_i^*\| \leq H$, $\forall i \in [N]$. Let $\kappa(\bar{A}_i)$ be the condition number of $\bar{A}_i$. Then:*

$$\|\theta_i^* - \theta_j^*\| \leq \Gamma(\epsilon, \epsilon_1) \triangleq \max_{i \in [N]} \left\{ \frac{\kappa(\bar{A}_i)H}{1 - \kappa(\bar{A}_i)\frac{A(\epsilon)}{\delta_1}} \left( \frac{A(\epsilon)}{\delta_1} + \frac{b(\epsilon, \epsilon_1)}{\delta_2} \right) \right\}.$$

**Discussion.** Theorem 1 reveals how heterogeneity in the rewards and transition kernels of MDPs can be mapped to differences in the limiting behavior of TD(0) on such MDPs from a fixed-point perspective. It formalizes the intuition that if the level of heterogeneity - as captured by $\epsilon$ and $\epsilon_1$ - is small, then so is the

gap in the TD(0) limit points of the agents' MDPs. This result is novel, and complements similar perturbation results in the RL literature such as the *Simulation Lemma* (Kearns & Singh, 2002).[4]

In what follows, we will introduce the key concept of a virtual MDP, and build on Theorem 1 to relate properties of this virtual MDP to those of the agents' individual MDPs.

### 3.1 Virtual Markov Decision Process

In a standard FL setting, the goal is to typically minimize a global loss function $f(x) = (1/N) \sum_{i \in [N]} f_i(x)$ composed of the local loss functions of $N$ agents; here, $f_i(x)$ is the local loss function of agent $i$. In FL, due to heterogeneity in the agents' loss functions, there is a "drift" effect (Charles & Konečný, 2020; Karimireddy et al., 2020b): the local iterates of each agent $i$ drift towards the minimizer of $f_i(x)$. However, when the heterogeneity is moderate, the average of the agents' iterates converges towards the minimizer of $f(x)$. To develop an analogous theory for FRL, we need to first answer: *When we average TD(0) update directions from different MDPs, where does the average TD(0) update direction lead us?* It is precisely to answer this question that we introduce the concept of a *virtual MDP*. To model a virtual environment that captures the "average" of the agents' individual environments, we construct an MDP $\bar{\mathcal{M}} = \langle \mathcal{S}, \mathcal{A}, \bar{\mathcal{R}}, \bar{\mathcal{P}}, \gamma \rangle$, where $\bar{\mathcal{P}} = (1/N) \sum_{i=1}^{N} \mathcal{P}^{(i)}$, and $\bar{\mathcal{R}} = (1/N) \sum_{i=1}^{N} \mathcal{R}^{(i)}$. Note that the virtual MDP is a fictitious MDP that we construct solely for the purpose of analysis, and it may not coincide with any of the agents' MDPs, in general.

**Properties of the Virtual MDP.** When applied to $\bar{\mathcal{M}}$, let the policy $\mu$ that we seek to evaluate induce a virtual MRP characterized by the tuple $\{\bar{P}, \bar{R}\}$. It is easy to see that $\bar{P} = (1/N) \sum_{i=1}^{N} P^{(i)}$, and $\bar{R} = (1/N) \sum_{i=1}^{N} R^{(i)}$. The following result shows how the virtual MRP inherits certain basic properties from the individual MRPs; the result is quite general and may be of independent interest.

**Proposition 1.** *(**Convex combinations of Markov matrices**) Let $\{P^{(i)}\}_{i=1}^{N}$ be a set of Markov matrices associated with Markov chains that share the same states, and are each aperiodic and irreducible. Then, for any set of weights $\{w_i\}_{i=1}^{N}$ satisfying $w_i \geq 0, \forall i \in [N]$ and $\sum_{i \in [N]} w_i = 1$, the Markov chain corresponding to the matrix $\sum_{i \in [N]} w_i P^{(i)}$ is also aperiodic and irreducible.*

The above result immediately tells us that the Markov chain corresponding to $\bar{P}$ is aperiodic and irreducible. Thus, there exists an unique stationary distribution $\bar{\pi}$ of this Markov chain; let $\bar{D}$ be the corresponding diagonal matrix. As before, let us define $\bar{A} \triangleq \Phi^\top \bar{D}(\Phi - \gamma \bar{P}\Phi)$, $\bar{b} \triangleq \Phi^\top \bar{D}\bar{R}$, and use $\theta^*$ to denote the solution to the equation $\bar{A}\theta^* = \bar{b}$. Our next result is a consequence of Theorem 1, and characterizes the gap between $\theta_i^*$ and $\theta^*$, for each $i \in [N]$.

**Proposition 2.** *(**Virtual MRP is "close" to Individual MRPs**) Fix any $i \in [N]$. Using the same definitions as in Theorem 1, we have $\|\bar{A}_i - \bar{A}\| \leq A(\epsilon)$, $\|\bar{b}_i - \bar{b}\| \leq b(\epsilon, \epsilon_1)$ and $\|\theta_i^* - \theta^*\| \leq \Gamma(\epsilon, \epsilon_1)$.*

We will later argue that the federated TD algorithm (to be introduced in Section 4) converges to a ball centered around the TD(0) fixed point $\theta^*$ of the virtual MRP. Proposition 2 is thus particularly important since it tells us that in a low-heterogeneity regime, by converging close to $\theta^*$, we also converge close to the optimal parameter $\theta_i^*$ of each agent $i$. This justifies studying the convergence behavior of FedTD(0) on the virtual MRP. Define $\Sigma_v \triangleq \Phi^\top \bar{D}\Phi$. The smallest eigenvalue of this matrix will end up dictating the convergence rate of our proposed algorithm. We end this section with a result showing that this eigenvalue is bounded away from zero.

**Proposition 3.** *For the virtual MRP, it holds that $\lambda_{\max}(\Sigma_v) \leq 1$, and $\exists \bar{\omega} > 0$ s.t. $\lambda_{\min}(\Sigma_v) \geq \bar{\omega}$.*

## 4 Federated TD Algorithm

In this section, we describe the FedTD(0) algorithm (outlined in Algorithm 1). The goal of FedTD(0) is to generate a model $\theta$ such that $\hat{V}_\theta$ is a good approximation of each agent $i$'s value function $V_\mu^{(i)}$, corresponding

---

[4]The simulation lemma tells us that if two MDPs with the same state and action spaces are similar, then so are the value functions induced by a common policy on these MDPs.

---

**Algorithm 1** Description of `FedTD`$(0)$

---

1: **Input:** Policy $\mu$, local step-size $\alpha_l$, global step-size $\alpha_g^{(t)}$ that depends on communication round $t$
2: **Initialize:** $\bar{\theta}_0 = \theta_0$ and $s_{0,0}^{(i)} = s_0, \forall i \in [N]$
3: **for** each round $t = 0, \ldots, T-1$ **do**
4:     **for** each agent $i \in [N]$ **do**
5:         **for** $k = 0, \ldots, K-1$ **do** with initial model $\theta_{t,0}^{(i)} = \bar{\theta}_t$
6:             Agent $i$ plays $\mu(s_{t,k}^{(i)})$, observes $O_{t,k}^{(i)} = (s_{t,k}^{(i)}, r_{t,k}^{(i)}, s_{t,k+1}^{(i)})$, and updates local model:

$$\theta_{t,k+1}^{(i)} = \theta_{t,k}^{(i)} + \alpha_l g_i(\theta_{t,k}^{(i)}), \text{ where } g_i(\theta_{t,k}^{(i)}) \triangleq \left( r_{t,k}^{(i)} + \gamma \phi(s_{t,k+1}^{(i)})^\top \theta_{t,k}^{(i)} - \phi(s_{t,k}^{(i)})^\top \theta_{t,k}^{(i)} \right) \phi(s_{t,k}^{(i)})$$

7:         **end for**
8:         Agent $i$ sends $\Delta_t^{(i)} = \theta_{t,K}^{(i)} - \bar{\theta}_t$ back to the server
9:     **end for**
10:     Server broadcasts the following global model: $\bar{\theta}_{t+1} = \Pi_{2,\mathcal{H}}(\bar{\theta}_t + (\alpha_g^{(t)}/N) \sum_{i \in [N]} \Delta_t^{(i)})$
11: **end for**

---

to the policy $\mu$. In line with both standard FL algorithms, and also works in MARL/FRL (in homogeneous settings) (Doan et al., 2019; Khodadadian et al., 2022), the agents keep their raw observations (i.e., their rewards, states, and actions) private, and only exchange local models. In each round $t$, each agent $i \in [N]$ starts from a common global model $\bar{\theta}_t$ and uses its local data to perform $K$ local updates of the following form: at each local iteration $k$, agent $i$ takes action $\mu(s_{t,k}^{(i)})$ and observes a data tuple $O_{t,k}^{(i)}$ based on its *own* MRP, i.e., $\{P^{(i)}, R^{(i)}\}$; we note here that *observations are independent across agents.* Using its data tuple, agent $i$ then updates its own local model $\theta_{t,k}^{(i)}$ along the direction $g_i(\theta_{t,k}^{(i)})$ in line 6. Since each agent seeks to benefit from the samples acquired by the other agents, there is intermittent communication via the server. However, such communication needs to be limited as communication-efficiency is a key concern in FL. As such, the agents upload their local models' difference $\Delta_t^{(i)}$ to the server only once every $K$ time-steps. The server averages these model differences and performs a projection to construct a global model $\bar{\theta}_{t+1}$ that is then broadcast to all agents (line 10). Here, we use $\Pi_{2,\mathcal{H}}(\cdot)$ to denote the standard Euclidean projection on to a convex compact subset $\mathcal{H} \subset \mathbb{R}^d$ that is assumed to contain each $\theta_i^*, i \in [N]$, and also $\theta^*$. Such a projection step ensures that the global models do not blow up, and is common in stochastic approximation (Borkar, 2009) and RL (Bhandari et al., 2018; Doan et al., 2019). Each agent then resumes its local updating process from this global model.

We note that the structure of `FedTD`$(0)$ mirrors that of `FedAvg` (and its many variants) where agents perform multiple local model-updates in isolation using their own data (to save communication), and synchronize periodically via a server. However, there are significant differences in the *dynamics* of standard FL algorithms and `FedTD`$(0)$, making it quite challenging to derive finite-time convergence results for the latter. In the next section where we analyze `FedTD`$(0)$, we will explain the nature of these challenges, and discuss how we overcome them.

## 5 Main Result and Analysis

To state our main convergence result for `FedTD`$(0)$, we need to introduce a few objects. First, let $H$ denote the radius of the set $\mathcal{H}$ in line 10 of Algorithm 1. Also, define $G \triangleq R_{\max} + 2H$ and $\nu \triangleq (1-\gamma)\bar{\omega}$, where $\bar{\omega}$ is as in Proposition 3. In our analysis, we will make use of the geometric mixing property of finite-state, aperiodic, and irreducible Markov chains (Levin & Peres, 2017). Specifically, under Assumption 3, for each $i \in [N]$, there exists some $m_i \geq 1$ and $\rho_i \in (0,1)$, such that for all $t \geq 0$ and $0 \leq k \leq K-1$:

$$d_{TV}\left(\mathbb{P}\left(s_{t,k}^{(i)} = \cdot \mid s_{0,0}^{(i)} = s\right), \pi^{(i)}\right) \leq m_i \rho_i^{tK+k}, \forall s \in \mathcal{S}.$$

Here, we use $d_{TV}(P, Q)$ to denote the total-variation distance between two probability measures $P$ and $Q$. For any $\bar{\epsilon} > 0$, let us define the mixing time for $P^{(i)}$ as $\tau_i^{\text{mix}}(\bar{\epsilon}) \triangleq \min\{t \in \mathbb{N}_0 \mid m_i \rho_i^t \leq \bar{\epsilon}\}$. Finally, let $\tau(\bar{\epsilon}) = \max_{i \in [N]} \tau_i^{\text{mix}}(\bar{\epsilon})$ represent the mixing time corresponding to the Markov chain that mixes the slowest. As one might expect, and as formalized by our main result below, it is this slowest-mixing Markov chain that dictates certain terms in the convergence rate of `FedTD(0)`.

**Theorem 2.** *(**Main Result**) There exists a decreasing global step-size sequence $\{\alpha_g^{(t)}\}$, a fixed local step-size $\alpha_l$, and a set of convex weights, such that a convex combination $\tilde{\theta}_T$ of the global models $\{\bar{\theta}_t\}$ satisfies the following for each agent $i \in [N]$ after $T$ rounds:*

$$\mathbb{E}\left[\left\|V_{\tilde{\theta}_T} - V_{\theta_i^*}\right\|_{\bar{D}}^2\right] \leq \tilde{\mathcal{O}}\left(\frac{\tau^2 G^2 + K^2}{K^2 T^2} + \frac{c_{quad}(\tau)}{\nu^2 NKT} + \frac{c_{lin}(\tau)}{\nu^4 KT^2} + Q(\epsilon, \epsilon_1)\right), \tag{2}$$

*where $\tau = \lceil \frac{\tau^{\text{mix}}(\alpha_T^2)}{K} \rceil$, $\alpha_T = K\alpha_l \alpha_g^{(T)}$, and $c_{quad}(\tau)$ and $c_{lin}(\tau)$ are quadratic and linear functions in $\tau$, respectively. Moreover, $B(\epsilon, \epsilon_1) = H\left(\sqrt{n}\epsilon + 2(n-1)\epsilon + \mathcal{O}(\epsilon^2) + \mathcal{O}(\epsilon_1)\right)$, $\Gamma(\epsilon, \epsilon_1)$ is as defined in Theorem 1, and $Q(\epsilon, \epsilon_1) = \tilde{\mathcal{O}}(\frac{B(\epsilon, \epsilon_1)G}{\nu} + \Gamma^2(\epsilon, \epsilon_1))$.*

The proof of the above result is deferred to Appendix I. We now discuss its impplications.

**Discussion.** To parse Theorem 2, let us start by noting that the term $Q(\epsilon, \epsilon_1)$ in Eq. (2) captures the effect of heterogeneity; we will comment on this term later. When $T \gg N$, the dominant term among the first three terms in Eq. (2) is $c_{\text{quad}}(\tau)/(\nu^2 NKT)$. To appreciate the tightness of this term, we note that in a centralized setting (i.e., when $N = 1$), given access to $KT$ samples, the convergence rate of `TD(0)` is $O(1/(\nu^2 KT))$ (Bhandari et al., 2018). Our analysis thus reveals that by communicating just $T$ times in $KT$ iterations, each agent *i can achieve a linear speedup w.r.t. the number of agents*. In a low-heterogeneity regime, i.e., when $Q(\epsilon, \epsilon_1)$ is small, we note that by combining data from different MDPs, `FedTD(0)` guarantees *fast* convergence to a model that is a good approximation of each agent's value function; by fast, we imply a $N$-fold speedup over the rate each agent would have achieved had it not communicated at all. Thus with little communication, `FedTD(0)` quickly provides each agent with a good model that it can then fine-tune for personalization. *Theorem 2 is significant in that it is the first result of its kind in MARL/FRL with heterogeneous environments, and complements the numerous analogous results in heterogeneous federated optimization (Sahu et al., 2018; Khaled et al., 2019; 2020; Li et al., 2019; Koloskova et al., 2020; Woodworth et al., 2020b; Malinovskiy et al., 2020; Pathak & Wainwright, 2020; Wang et al., 2020; Karimireddy et al., 2020b; Acar et al., 2021; Gorbunov et al., 2021; Mitra et al., 2021; Mishchenko et al., 2022).*

When all the MDPs are identical, $Q(\epsilon, \epsilon_1) = 0$. But when the MDPs are different, should we expect such a term? To further understand the effect of heterogeneity, it suffices to get rid of all the randomness in our setting. As such, suppose we replace the random `TD(0)` direction $g_i(\theta_{t,k}^{(i)})$ of each agent $i$ in Algorithm 1 by its *steady-state* deterministic version $\bar{g}_i(\theta_{t,k}^{(i)}) = \bar{b}_i - \bar{A}_i \theta_{t,k}^{(i)}$, where $\bar{A}_i$ and $\bar{b}_i$ are as in Section 3. We call the resulting deterministic algorithm *mean-path* `FedTD(0)`. For simplicity, we skip the projection step. In our next result, we exploit the affine nature of the steady-state `TD(0)` directions to characterize the effect of heterogeneity in the limiting behavior of `FedTD(0)`.

**Theorem 3.** *(**Heterogeneity Bias**) Suppose $N = 2$ and $K = 1$. Let the step-size $\alpha = \alpha_l \alpha_g^{(t)}$ be chosen such that $I - \alpha\hat{A}$ is Schur stable, where $\hat{A} = (\bar{A}_1 + \bar{A}_2)/2$. Define $e_{i,t} \triangleq \bar{\theta}_t - \theta_i^*, i \in \{1, 2\}$. The output of mean-path `FedTD(0)` then satisfies:*

$$\lim_{t \to \infty} e_{1,t} = \frac{1}{2}\hat{A}^{-1}\bar{A}_2(\theta_1^* - \theta_2^*); \lim_{t \to \infty} e_{2,t} = \frac{1}{2}\hat{A}^{-1}\bar{A}_1(\theta_2^* - \theta_1^*). \tag{3}$$

**Discussion:** For the setting described in Theorem 3, the mean-path `FedTD(0)` updates follow the deterministic recursion $\bar{\theta}_{t+1} = (I - \alpha\hat{A})\bar{\theta}_t + \alpha\hat{b}$, where $\hat{b} = (1/2)(\bar{b}_1 + \bar{b}_2)$. This is a discrete-time linear time-invariant system (LTI). The dynamics of this system are stable if and only if the state transition matrix $(I - \alpha\hat{A})$ is Schur stable, justifying the choice of $\alpha$ in Theorem 3. The main message conveyed by this result is that the gap between the limit point of mean-path `FedTD(0)` and the optimal parameter $\theta_i^*$ of either of the two

MRPs bears a dependence on *the difference in the optimal parameters of the MRPs - a natural indicator of heterogeneity between the two MRPs.* Furthermore, this term has no dependence on the step-size $\alpha$, i.e., the effect of the heterogeneity-induced bias *cannot be eliminated* by making $\alpha$ arbitrarily small. Aligning with this observation, notice that $Q(\epsilon, \epsilon_1)$ in Eq. (2) is also step-size independent. The above discussion sheds some light on the fact that a term of the form $Q(\epsilon, \epsilon_1)$ is to be expected in Theorem 2. *Notably, the bias term in Eq. (3) persists even when the number of local steps is just one, i.e., even when the agents communicate with the server at all time steps.* This is a key difference with the standard FL setting where the effect of heterogeneity manifests itself *only* when the number of local steps $K$ strictly exceeds 1 (Charles & Konečný, 2021; Karimireddy et al., 2020b; Mitra et al., 2021).

## 5.1 Main Technical Challenges and Overview of the Novel Ingredients in Our Analysis

**Challenges.** We summarize the major technical challenges that show up in the analysis of Theorem 2. First, the FedTD(0) update direction may not correspond to the TD(0) update direction of *any* MDP. This challenge is unique to our setting, and neither shows up in the centralized TD(0) analysis (Bhandari et al., 2018; Srikant & Ying, 2019), nor in the existing MARL/FRL analyses with homogeneous MDPs (Doan et al., 2019; Khodadadian et al., 2022). Second, unlike standard FL analyses that deal with i.i.d. observations for each agent, our setting is complicated by the fact that each agent's data is generated from a Markov chain. Moreover, for each agent $i$, the parameter sequence $\{\theta_{t,k}^{(i)}\}$ and the data tuples $\{O_{t,k}^{(i)}\}$ are intricately coupled. Third, the synchronization step in FedTD(0) creates complex statistical dependencies between the local parameter of any given agent and the past observations of *all* other agents. Fourth, controlling the gradient bias $(1/NK) \sum_{i=1}^{N} \sum_{k=0}^{K-1} \left( g_i(\theta_{t,k}^{(i)}, O_{t,k}^{(i)}) - \bar{g}_i(\theta_{t,k}^{(i)}) \right)$ and the gradient norm $\mathbb{E}\|(1/NK) \sum_{i=1}^{N} \sum_{k=0}^{K-1} g_i(\theta_{t,k}^{(i)})\|^2$ requires a very delicate analysis when one seeks to establish the linear speedup property w.r.t. the number of agents $N$, i.e., the $\mathcal{O}(1/NKT)$-type rate. In particular, naively bounding terms using the projection radius (as in the centralized analysis (Bhandari et al., 2018)) will not yield the linear speedup property. Finally, we need to control the "client-drift" effect due to environmental heterogeneity under the strong coupling between the different random variables discussed above.

**Proof Sketch for Theorem 2.** Our first key innovation is to build on the results in Section 3 to show that the mean-path (steady-state) FedTD(0) update direction $(1/N) \sum_{i=1}^{N} \bar{g}_i(\theta)$ is "close" to the mean-path TD(0) update direction $\bar{g}(\theta) = \bar{b} - \bar{A}\theta$ of the virtual MRP we constructed in Section 3.1; here, $\bar{b}, \bar{A}$ are as defined in Section 3.1. Formally, we have the following result.

**Lemma 2.** *(Steady-state Pseudo-Gradient Heterogeneity) For each $\theta \in \mathcal{H}$, we have:*

$$\left\| \bar{g}(\theta) - \frac{1}{N} \sum_{i=1}^{N} \bar{g}_i(\theta) \right\| \leq B(\epsilon, \epsilon_1), \tag{4}$$

*where $B(\epsilon, \epsilon_1)$ is as in Theorem 2, and $\bar{g}(\theta)$ is the steady-state TD(0) direction of the virtual MRP.*

From Bhandari et al. (2018), we know that $\bar{g}(\theta)$ acts like a pseudo-gradient pointing towards the optimal model $\theta^*$ of the virtual MRP. Since based on Proposition 2, we know that $\theta^*$ is close to $\theta_i^*, \forall i \in [N]$, Lemma 2 tells us that at least in the steady-state, the iterates of FedTD(0) will converge to a neighborhood of each agent's optimal model, where the size of the neighborhood depends on the level of heterogeneity. While this helps build intuition, all the valuable insights conveyed by Lemma 2 only pertain to the *steady state* dynamics of FedTD(0), i.e., all the statistical challenges we alluded to still need to be resolved. In particular, as mentioned earlier, we cannot naively use a projection bound of the form $\mathbb{E}\left[\|(1/NK) \sum_{i=1}^{N} \sum_{k=0}^{K-1} g_i(\theta_{t,k}^{(i)})\|^2\right] = \mathcal{O}(G^2)$ from the centralized analysis in Bhandari et al. (2018), since the local models may not belong to the set $\mathcal{H}$. Also, this will obscure the linear speedup effect. We overcome this difficulty by decomposing the random TD direction of each agent $i$ as $g_i(\theta_{t,k}^{(i)}) = b_i(O_{t,k}^{(i)}) - A_i(O_{t,k}^{(i)})\theta_{t,k}^{(i)}$. Since $A_i(O_{t,k}^{(i)})$ and $b_i(O_{t,k}^{(i)})$ only depend on the randomness from the Markov chain, and $O_{t,k}^{(i)}$ and $O_{t,k}^{(j)}$ are independent, we can show that the variances of $(1/NK) \sum_{i=1}^{N} \sum_{k=0}^{K-1} A_i(O_{t,k}^{(i)})$ and $(1/NK) \sum_{i=1}^{N} \sum_{k=0}^{K-1} b_i(O_{t,k}^{(i)})$ get scaled down by $NK$ (up to higher order terms). Furthermore, to account for the fact that $A_i(O_{t,k}^{(i)})$ and $b_i(O_{t,k}^{(i)})$ differ across agents, we appeal to

**Lemma 2.** Putting these pieces together in a careful manner yields the final rate in Theorem 2. The detailed analysis, along with some simulations, are deferred to the Appendix.

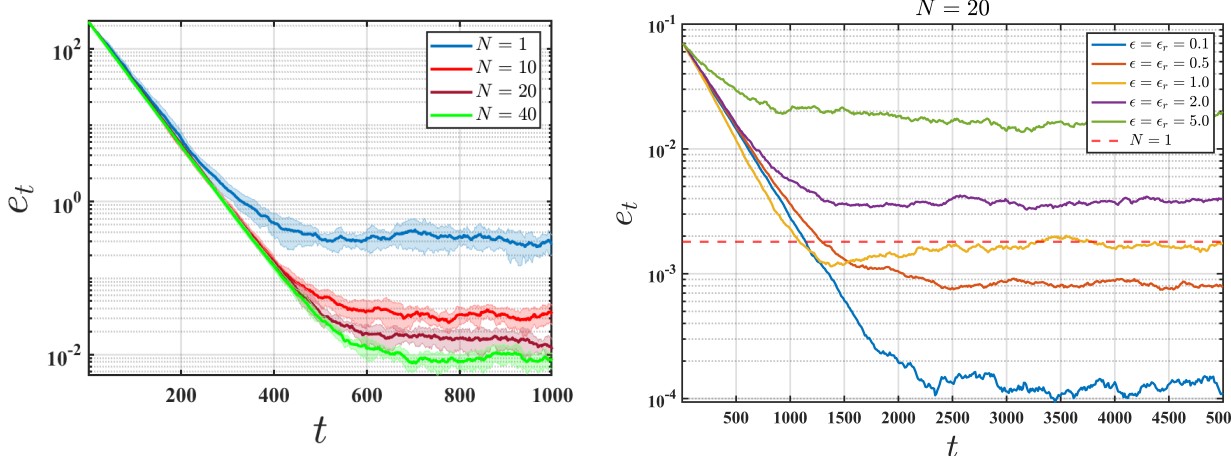

(a) Simulations on the effect of the linear speedup     (b) Simulations on the effect of the heterogeneity level

Figure 2: Performance of FedTD(0) under Markovian sampling. (*a*) Performance of FedTD(0) for varying number of agents $N$. The MDP $\mathcal{M}^{(1)}$ of the first agent is randomly generated with a state space of size $n = 100$. The remaining MDPs are perturbations of $\mathcal{M}^{(1)}$ with the heterogeneity levels $\epsilon = 0.05$ and $\epsilon_1 = 0.1$. We evaluate the convergence in terms of the running error $e_t = \|\bar{\theta}_t - \theta_1^*\|^2$. (*b*) Performance of FedTD(0) for varying heterogeneity level, with a fixed number of agents $N = 20$. Complying with theory, increasing $N$ reduces the error, and increasing the level of heterogeneity increases the size of the ball to which FedTD(0) converges. We choose the number of local steps as $K = 10$ in both plots.

## 6 Conclusion

In this work, we have studied the problem of federated reinforcement learning under environmental heterogeneity and explored the following question: *Can an agent expedite the process of learning its own value function by using information from agents interacting with potentially different MDPs?* To answer this question, we studied the convergence of a federated TD(0) algorithm with linear function approximation, where $N$ agents under different environments collaboratively evaluate a common policy. The main differences from the existing works are: (i) proposing a new definition of environmental heterogeneity; (ii) characterizing the effect of heterogeneity on TD(0) fixed points; (iii) introducing a virtual MDP to analyze the long-term behavior of the FedTD(0) algorithm; and (iv) making an explicit connection between federated reinforcement learning and federated supervised learning/optimization by leveraging the virtual MDP. With these elements, we proved that if the environmental heterogeneity between agents' environments is small, then FedTD(0) can achieve a linear speedup under both i.i.d and Markovian settings, and with multiple local updates.

A few interesting extensions to this work are as follows. First, it is natural to study federated variants of other RL algorithms beyond the TD(0) algorithm. Second, it would be interesting to investigate whether the personalization techniques used in the traditional FL optimization literature can be applied to solve federated RL problems. Instead of learning a common value function/policy, can we design personalized value functions/policies that might perform better in high-heterogeneity regimes? We leave the exploration of this interesting question as future work.

**Acknowledgments**

Anderson and Wang are partially supported by the NSF under awards 2144634 & 2231350 from the EPCN program.

Hamed Hassani is supported by The Institute for Learning-enabled Optimization at Scale (TILOS), under award number NSF-CCF-2112665.

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

# Contents

# A    Additional Literature Survey

**Federated Learning Algorithms**. The literature on algorithmic developments in federated learning is vast; as such, we only cover some of the most relevant/representative works here. The most popularly used FL algorithm, `FedAvg`, was first introduced in McMahan et al. (2017). Several works went on to provide a detailed theoretical analysis of `FedAvg` both in the homogeneous case when all clients minimize the same objective function (Stich, 2018; Wang & Joshi, 2018; Spiridonoff et al., 2020; Reisizadeh et al., 2020; Haddadpour et al., 2019; Woodworth et al., 2020a), and also in the more challenging heterogeneous setting (Khaled et al., 2019; 2020; Haddadpour & Mahdavi, 2019; Li et al., 2019; Koloskova et al., 2020). In the latter scenario, it was soon realized that `FedAvg` suffers from a "client-drift" effect that hurts its convergence performance (Charles & Konečnỳ, 2020; 2021; Karimireddy et al., 2020a).

Since then, a lot of effort has gone into improving the convergence guarantees of `FedAvg` via a variety of technical approaches: proximal methods in `FedProx` (Sahu et al., 2018); operator-splitting in `FedSplit` (Pathak & Wainwright, 2020); variance-reduction in `Scaffold` (Karimireddy et al., 2020a) and `S-Local-SVRG` (Gorbunov et al., 2021); gradient-tracking in `FedLin` (Mitra et al., 2021); dynamic regularization in Acar et al. (2021); and ADMM in `FedADMM` (Wang et al., 2022a). While these methods improved upon `FedAvg` in various ways, they all fell short of providing any theoretical justification for performing multiple local updates under arbitrary statistical heterogeneity. Very recently, the authors in Mishchenko et al. (2022) introduced the `ProxSkip` algorithm, and showed that it can indeed lead to communication savings via local steps, despite arbitrary heterogeneity.

Some other approaches to tackling heterogeneous statistical distributions in FL include personalization (Deng et al., 2020; Fallah et al., 2020; T Dinh et al., 2020; Hanzely et al., 2020; Tan et al., 2022), clustering (Ghosh et al., 2020; Sattler et al., 2020; Su et al., 2022), representation learning (Collins et al., 2021), and the use of quantiles (Laguel et al., 2021).

# B   Perturbation bounds for TD(0) fixed points

## B.1   Proof of Theorem 1

In this section, we prove the perturbation bounds on TD(0) fixed points shown in Theorem 1. We start by observing that:

$$
\begin{aligned}
\|\bar{A}_i - \bar{A}_j\| &= \|\Phi^\top D^{(i)}(\Phi - \gamma P^{(i)}\Phi) - \Phi^\top D^{(j)}(\Phi - \gamma P^{(j)}\Phi)\| \\
&\leq \|\Phi^\top D^{(i)}(\Phi - \gamma P^{(i)}\Phi) - \Phi^\top D^{(i)}(\Phi - \gamma P^{(j)}\Phi) + \\
&\qquad \Phi^\top D^{(i)}(\Phi - \gamma P^{(j)}\Phi) - \Phi^\top D^{(j)}(\Phi - \gamma P^{(j)}\Phi)\| \\
&\leq \|\Phi^\top D^{(i)}(\Phi - \gamma P^{(i)}\Phi) - \Phi^\top D^{(i)}(\Phi - \gamma P^{(j)}\Phi)\| + \\
&\qquad \|\Phi^\top D^{(i)}(\Phi - \gamma P^{(j)}\Phi) - \Phi^\top D^{(j)}(\Phi - \gamma P^{(j)}\Phi)\| \\
&\overset{(a)}{\leq} \gamma\|\Phi\|^2\|D^{(i)}\|\|P^{(i)} - P^{(j)}\| + \|\Phi\|^2\|D^{(i)} - D^{(j)}\|\|(I - \gamma P^{(j)})\| \\
&\overset{(b)}{\leq} \gamma\sqrt{n}\epsilon + (1 + \gamma)[2(n-1)\epsilon + \mathcal{O}(\epsilon^2)],
\end{aligned}
\tag{5}
$$

where (a) follows from the triangle inequality. The first term in (b) uses the fact that $\|\Phi\| \leq 1$, $\|D^{(i)}\| \leq 1$, and

$$
\|P^{(i)} - P^{(j)}\| \leq \sqrt{n}\|P^{(i)} - P^{(j)}\|_\infty \leq \epsilon\sqrt{n}\|P^{(i)}\|_\infty = \epsilon\sqrt{n},
$$

where we use Assumption 1 in the second inequality. The second term in (b) uses the the facts that $\|I - \gamma P^{(j)}\| \leq 1 + \gamma$, $\|D^{(i)} - D^{(j)}\| \leq \|D^{(i)} - D^{(j)}\|_1 \leq \|\pi^{(i)} - \pi^{(j)}\|_1$, along with Lemma 1.

Next, we bound

$$
\begin{aligned}
\|\bar{b}_i - \bar{b}_j\| &= \|\Phi D^{(i)} R^{(i)} - \Phi D^{(j)} R^{(j)}\| \\
&\leq \|\Phi D^{(i)} R^{(i)} - \Phi D^{(i)} R^{(j)}\| + \|\Phi D^{(i)} R^{(j)} - \Phi D^{(j)} R^{(j)}\| \\
&\leq \|\Phi\|\|D^{(i)}\|\|R^{(i)} - R^{(j)}\| + \|\Phi\|\|D^{(i)} - D^{(j)}\|\|R^{(j)}\| \\
&\leq \epsilon_1 + R_{\max}\left(2(n-1)\epsilon + \mathcal{O}(\epsilon^2)\right),
\end{aligned}
\tag{6}
$$

where we use Assumption 2 in the last inequality and follow the same reasoning as we used to bound $\|\bar{A}_i - \bar{A}_j\|$ above.

We are now ready to bound the gap between fixed points as:

$$
\frac{\|\theta_i^* - \theta_j^*\|}{\|\theta_i^*\|} \leq \frac{\kappa(\bar{A}_i)}{1 - \kappa(\bar{A}_i)\frac{\|\bar{A}_i - \bar{A}_j\|}{\|\bar{A}_i\|}} \left( \frac{\|\bar{A}_i - \bar{A}_j\|}{\|\bar{A}_i\|} + \frac{\|\bar{b}_i - \bar{b}_j\|}{\|\bar{b}_i\|} \right).
\tag{7}
$$

Here, we leveraged the perturbation theory of linear equations in (Horn & Johnson, 2012b) Section 5.8. Finally, for any $\|\theta_i^*\| \leq H$, we have

$$
\|\theta_i^* - \theta_j^*\| \leq \Gamma(\epsilon, \epsilon_1) \triangleq \frac{\kappa(\bar{A}_i)H}{1 - \kappa(\bar{A}_i)\frac{A(\epsilon)}{\delta_1}} \left( \frac{A(\epsilon)}{\delta_1} + \frac{b(\epsilon, \epsilon_1)}{\delta_2} \right),
$$

where we used the fact that $\delta_1$ and $\delta_2$ are positive constants that lower bound $\|\bar{A}_i\|$ and $\|\bar{b}_i\|$, respectively.

## C  Properties of the Virtual Markov Decision Process

### C.1  Proof of Proposition 1

Before we prove this proposition, we present the following fact from (Pishro-Nik, 2016): *a Markov matrix $P$ is irreducible and aperiodic if and only if there exists a positive integer $k$ such that every entry of the matrix $P^k$ is strictly positive, i.e., $P^k_{s,s'} > 0$, for all $s, s' \in \mathcal{S}$.*

For every Markov matrix $P^{(i)}$, we know that there exists such an integer $k_i$ according to the above fact and Assumption 3 in the paper. Then we define a set $J = \{i \in [N] | w_i > 0\}$. Since $\sum_{i=1}^{N} w_i = 1$, and $w_i \geq 0$ holds for all $i \in [N]$, we know that $J$ is a non-empty set. If we define $\bar{k} = \min_{i \in [J]}\{k_i\}$ and $j = \arg\min_{i \in [J]}\{k_i\}$, then we have:

$$\left(\sum_{i \in [N]} w_i P^{(i)}\right)^{\bar{k}} = \underbrace{w_j^{\bar{k}}\left(P^{(j)}\right)^{\bar{k}}}_{\text{positive}} + \underbrace{\cdots\cdots}_{\text{nonnegative}}, \tag{8}$$

where each entry of $w_j^{\bar{k}}\left(P^{(j)}\right)^{\bar{k}}$ is strictly positive while the other matrices in the summation are non-negative. Thus, we can conclude that the Markov chain associated with the Markov matrix $\sum_{i \in [N]} w_i P^{(i)}$ is also irreducible and aperiodic.

### C.2  Proof of Proposition 2

Following similar arguments as in Theorem 1, we bound $\|\bar{A}_i - \bar{A}\|$:

$$\begin{aligned}
\|\bar{A}_i - \bar{A}\| &= \|\Phi^\top D^{(i)}(\Phi - \gamma P^{(i)}\Phi) - \Phi^\top \bar{D}(\Phi - \gamma \bar{P}\Phi)\| \\
&\overset{(a)}{\leq} \gamma \|\Phi\|^2 \|D^{(i)}\| \|P^{(i)} - \bar{P}\| + \|\Phi\|^2 \|D^{(i)} - \bar{D}\| \|(I - \gamma \bar{P})\| \\
&\overset{(b)}{\leq} \gamma\sqrt{n}\epsilon + (1+\gamma)[2(n-1)\epsilon + \mathcal{O}(\epsilon^2)] = A(\epsilon),
\end{aligned} \tag{9}$$

where inequality (a) follows the same reasoning as (a) in Eq.(5), (b) uses the same fact as (b) in Eq.(5), and $\|P^{(i)} - \bar{P}\| \leq \frac{1}{N}\sum_{j=1}^{N}\|P^{(i)} - P^{(j)}\| \leq \epsilon\sqrt{n}$ and $\|D^{(i)} - \bar{D}\| \leq 2(n-1)\epsilon + \mathcal{O}(\epsilon^2)$.

Based on the above facts: (i) $\|\bar{R}\| \leq \frac{1}{N}\sum_{i=1}^{N}\|R^{(i)}\| \leq R_{\max}$, (ii) $\|R^{(i)} - \bar{R}\| \leq \frac{1}{N}\sum_{j=1}^{N}\|R^{(i)} - R^{(j)}\| \leq \epsilon_1$ and (iii) $\|D^{(i)} - \bar{D}\| \leq 2(n-1)\epsilon + \mathcal{O}(\epsilon^2)$, we finish the proof by showing that $\|\bar{b}_i - \bar{b}\| \leq b(\epsilon, \epsilon_1)$. To do so, we follow the same steps as Eq. (6), and prove the bound on $\|\theta_i^* - \theta^*\|$ by following the same analysis as Eq. (7).

### C.3  Proof of Proposition 3

Since the virtual MDP is an average of the agents' MDPs, i.e., $\bar{P} = \frac{1}{N}\sum_{i=1}^{N} P^{(i)}$, the virtual Markov chain is irreducible and aperiodic from Proposition 1. The maximum eigenvalue of a symmetric positive-semidefinite matrix is a convex function. Then we have $\lambda_{\max}(\Phi^\top \bar{D}\Phi) \leq \sum_{s \in \mathcal{S}} \bar{\pi}(s)\lambda_{\max}\left(\phi(s)\phi(s)^\top\right) \leq \sum_{s \in \mathcal{S}} \bar{\pi}(s) = 1$.

To show that there exists $\omega > 0$ such that $\lambda_{\min}(\Phi^\top \bar{D}\Phi) \geq \omega > 0$, we will establish that $\Phi^\top \bar{D}\Phi$ is a positive-definite matrix. Since $\Phi$ is full-column rank, this amounts to showing that $\bar{D}$ is a positive definite matrix. From the definition of $\bar{D}$, establishing positive-definiteness of $\bar{D}$ is equivalent to arguing that every element of the stationary distribution vector $\bar{\pi}$ is strictly positive; here, $\bar{\pi}^\top \bar{P} = \bar{\pi}$. To that end, from Proposition 1, we know that the Markov chain associated with $\bar{P}$ is aperiodic and irreducible. From the Perron-Frobenius theorem (Frobenius et al., 1912), we conclude that indeed every entry of $\bar{\pi}$ is strictly positive. If we choose $\omega = \min_{s \in \mathcal{S}}\{\bar{\pi}(s)\} > 0$, we have $\lambda_{\min}(\Phi^\top \bar{D}\Phi) \geq \omega > 0$.

# D    Pseudo-gradient heterogeneity: Proof of Lemma 2

For each $\theta \in \mathcal{H}$, we have:

$$\left\| \bar{g}(\theta) - \frac{1}{N} \sum_{i=1}^{N} \bar{g}_i(\theta) \right\| = \left\| \Phi^T \bar{D}(\bar{T}_\mu \Phi\theta - \Phi\theta) - \frac{1}{N} \Big( \sum_{i=1}^{N} \Phi^T D^{(i)}(T_\mu^{(i)} \Phi\theta - \Phi\theta) \Big) \right\|$$

$$\overset{(a)}{\leq} \frac{1}{N} \sum_{i=1}^{N} \left\| \Phi^T \bar{D}(\bar{T}_\mu \Phi\theta - \Phi\theta) - \Phi^T D^{(i)}(T_\mu^{(i)} \Phi\theta - \Phi\theta) \right\|$$

$$\overset{(b)}{\leq} \frac{1}{N} \sum_{i=1}^{N} \left\| \bar{D}\Big[\frac{1}{N} \sum_{j=1}^{N} R^{(j)} + \gamma \frac{1}{N} \sum_{j=1}^{N} P^{(j)} \Phi\theta - \Phi\theta\Big] - D^{(i)}(T_\mu^{(i)} \Phi\theta - \Phi\theta) \right\|$$

$$\leq \frac{1}{N} \sum_{i=1}^{N} \left\| \bar{D}\Big[\frac{1}{N} \sum_{j=1}^{N} R^{(j)} + \gamma \frac{1}{N} \sum_{j=1}^{N} P^{(j)} \Phi\theta - \Phi\theta\Big] - \bar{D}(T_\mu^{(i)} \Phi\theta - \Phi\theta) \right.$$

$$\left. + \bar{D}(T_\mu^{(i)} \Phi\theta - \Phi\theta) - D^{(i)}(T_\mu^{(i)} \Phi\theta - \Phi\theta) \right\|$$

$$\overset{(c)}{\leq} \frac{1}{N} \sum_{i=1}^{N} \left\| \bar{D}\Big[\frac{1}{N} \sum_{j=1}^{N} R^{(j)} + \gamma \frac{1}{N} \sum_{j=1}^{N} P^{(j)} \Phi\theta - \Phi\theta\Big] - \bar{D}(T_\mu^{(i)} \Phi\theta - \Phi\theta) \right\|$$

$$+ \frac{1}{N} \sum_{i=1}^{N} \left\| \bar{D}(T_\mu^{(i)} \Phi\theta - \Phi\theta) - D^{(i)}(T_\mu^{(i)} \Phi\theta - \Phi\theta) \right\|$$

$$\leq \frac{1}{N} \sum_{i=1}^{N} \left\| \bar{D} \right\| \left\| \frac{1}{N} \sum_{j=1}^{N} R^{(j)} - R^{(i)} \right\| + \gamma \left\| \frac{1}{N} \sum_{j=1}^{N} P^{(j)} - P^{(i)} \right\| \left\| \Phi\theta \right\|$$

$$+ \frac{1}{N} \sum_{i=1}^{N} \left\| \bar{D} - D^{(i)} \right\| \left\| T_\mu^{(i)} \Phi\theta - \Phi\theta \right\|$$

$$\overset{(d)}{\leq} \frac{1}{N} \sum_{i=1}^{N} \left\| \frac{1}{N} \sum_{j=1}^{N} R^{(j)} - R^{(i)} \right\|_2 + \gamma \left\| \frac{1}{N} \sum_{j=1}^{N} P^{(j)} - P^{(i)} \right\| \left\| \Phi\theta \right\|$$

$$+ \frac{1}{N} \sum_{i=1}^{N} \left\| \bar{D} - D^{(i)} \right\| \left\| T_\mu^{(i)} \Phi\theta - \Phi\theta \right\|$$

$$\overset{(e)}{\leq} \Big[\epsilon_1 + \gamma \sqrt{n}\epsilon \|\Phi\theta\| + \Big[2(n-1)\epsilon + \mathcal{O}(\epsilon^2)\Big] \|\Phi\theta\|$$

$$\leq H\Big[\mathcal{O}(\epsilon_1) + \gamma \sqrt{n}\epsilon + 2(n-1)\epsilon + \mathcal{O}(\epsilon^2)\Big] = B(\epsilon, \epsilon_1). \tag{10}$$

Inequalities (a) and (c) follow from the triangle inequality, (b) is due to $\|\Phi\| \leq 1$; (d) is due to the fact that $\|\bar{D}\| \leq 1$; and (e) uses the following facts: (i) $\|R^{(i)} - \bar{R}\| \leq \epsilon_1$; (ii) $\|P^{(i)} - P^{(j)}\| \leq \sqrt{n}\|P^{(i)} - P^{(j)}\|_\infty \leq \epsilon\sqrt{n}\|P^{(i)}\|_\infty = \epsilon\sqrt{n}$, which, in turn, follows from the proof of Theorem 1; (iii) $\|D^{(i)} - \bar{D}\| \leq 2(n-1)\epsilon + \mathcal{O}(\epsilon^2)$, which, in turn, follows from the proof of Theorem 1 or Eq. (5); and (iv) $\|\theta\| \leq H$ for any $\theta \in \mathcal{H}$.

# E    Auxiliary results used in the I.I.D. and Markovian settings

We make repeated use throughout the appendix (often without explicitly stating so) of the following inequalities:

- Given any two vectors $x, y \in \mathbb{R}^d$, for any $\beta > 0$, we have

$$\|x + y\|^2 \leq (1 + \beta)\|x\|^2 + \left(1 + \frac{1}{\beta}\right) \|y\|^2. \tag{11}$$

- Given any two vectors $x, y \in \mathbb{R}^d$, for any $\beta > 0$, we have

$$\langle x, y \rangle \leq \frac{\beta}{2} \|x\|^2 + \frac{1}{2\beta} \|y\|^2. \tag{12}$$

  This inequality goes by the name of Young's inequality.

- Given $m$ vectors $x_1, \ldots, x_m \in \mathbb{R}^d$, the following is a simple application of Jensen's inequality:

$$\left\| \sum_{i=1}^m x_i \right\|^2 \leq m \sum_{i=1}^m \|x_i\|^2. \tag{13}$$

We prove the following result for the virtual MDP.

**Lemma 3.** *For any $\theta_1, \theta_2 \in \mathbb{R}^d$,*

$$(\theta_2 - \theta_1)^\top [\bar{g}(\theta_1) - \bar{g}(\theta_2)] \geq (1 - \gamma) \left\| \hat{V}_{\theta_1} - \hat{V}_{\theta_2} \right\|_{\bar{D}}^2. \tag{14}$$

*Proof.* Consider a stationary sequence of states with random initial state $s \sim \bar{\pi}$ and subsequent state $s'$, which, conditioned on $s$, is drawn from $\bar{P}(\cdot \mid s)$. Define $\phi \triangleq \phi(s)$ and $\phi' \triangleq \phi(s')$. Define $\chi_1 \triangleq \hat{V}_{\theta_2}(s) - \hat{V}_{\theta_1}(s) = (\theta_2 - \theta_1)^\top \phi$ and $\chi_2 \triangleq \hat{V}_{\theta_2}(s') - \hat{V}_{\theta_1}(s') = (\theta_2 - \theta_1)^\top \phi'$. By stationarity, $\chi_1$ and $\chi_2$ are two correlated random variables with the same same marginal distribution. By definition, $\mathbb{E}[\chi_1^2] = \mathbb{E}[\chi_2^2] = \left\| \hat{V}_{\theta_2} - \hat{V}_{\theta_2} \right\|_{\bar{D}}^2$ since $s, s'$ are drawn from $\bar{\pi}$. And we have,

$$\bar{g}(\theta_1) - \bar{g}(\theta_2) = \mathbb{E}\left[\phi (\gamma\phi' - \phi)^\top (\theta_1 - \theta_2)\right] = \mathbb{E}\left[\phi (\chi_1 - \gamma\chi_2)\right].$$

Therefore,

$$\begin{aligned}
(\theta_2 - \theta_1)^\top [\bar{g}(\theta_1) - \bar{g}(\theta_2)] &= \mathbb{E}\left[\chi_1 (\chi_1 - \gamma\chi_2)\right] \\
&= \mathbb{E}[\chi_1^2] - \gamma\mathbb{E}[\chi_1\chi_2] \\
&\geq (1 - \gamma)\mathbb{E}[\chi_1^2] \\
&= (1 - \gamma) \left\| \hat{V}_{\theta_2} - \hat{V}_{\theta_2} \right\|_{\bar{D}}^2,
\end{aligned}$$

where we use the Cauchy-Schwartz inequality to conclude $\mathbb{E}[\chi_1\chi_2] \leq \sqrt{\mathbb{E}[\chi_1^2]}\sqrt{\mathbb{E}[\chi_2^2]} = \mathbb{E}[\chi_1^2]$. $\qquad\square$

**Lemma 4.** *For any $\theta_1, \theta_2 \in \mathbb{R}^d$, we have*

$$\|\bar{g}(\theta_1) - \bar{g}(\theta_2)\| \leq 2 \left\| \hat{V}_{\theta_1} - \hat{V}_{\theta_2} \right\|_{\bar{D}}. \tag{15}$$

*Proof.* Following the analysis of Lemma 3, we have

$$\begin{aligned}
\|\bar{g}(\theta_1) - \bar{g}(\theta_2)\| &= \|\mathbb{E}\left[\phi (\chi_1 - \gamma\chi_2)\right]\| \\
&\leq \sqrt{\mathbb{E}[\|\phi\|^2]}\sqrt{\mathbb{E}\left[(\chi_1 - \gamma\chi_2)^2\right]} \\
&\leq \sqrt{\mathbb{E}[\chi_1^2]} + \gamma\sqrt{\mathbb{E}[\chi_2^2]} \\
&= (1 + \gamma)\sqrt{\mathbb{E}[\chi_1^2]},
\end{aligned}$$

where the second inequality is due to $\|\phi\| \leq 1$ and the final equality is due to $\mathbb{E}\left[\chi_1^2\right] = \mathbb{E}\left[\chi_2^2\right]$. We finish the proof by using the fact that $\mathbb{E}\left[\chi_1^2\right] = \left\|\hat{V}_{\theta_2} - \hat{V}_{\theta_2}\right\|_{\bar{D}}^2$ and $1 + \gamma \leq 2$. $\qquad\square$

With this Lemma, we next show that the steady-state TD(0) update direction $\bar{g}$ and $\bar{g}_i$ are 2-Lipschitz.

**Lemma 5.** *(2-Lipschitzness of steady-state TD(0) update direction) For any $\theta_1, \theta_2 \in \mathbb{R}^d$, we have*

$$\|\bar{g}(\theta_1) - \bar{g}(\theta_2)\| \leq 2\|\theta_1 - \theta_2\|. \tag{16}$$

*And for each agent $i \in [N]$, we have*

$$\|\bar{g}_i(\theta_1) - \bar{g}_i(\theta_2)\| \leq 2\|\theta_1 - \theta_2\|. \tag{17}$$

*Proof.* From Lemma 4, we can easily conclude that the steady-state TD(0) update direction $\bar{g}$ for the vitual MDP is 2-Lipschitz, i.e.,

$$\|\bar{g}(\theta_1) - \bar{g}(\theta_2)\| \leq 2\|\theta_1 - \theta_2\|, \tag{18}$$

based on the fact that $\lambda_{\max}(\Phi^\top \bar{D}\Phi) \leq 1$. We can follow the same reasoning to prove Eq. (17) since $\|\bar{g}_i(\theta_1) - \bar{g}_i(\theta_2)\| \leq 2\left\|\hat{V}_{\theta_1} - \hat{V}_{\theta_2}\right\|_{D_i}$ holds for each $i \in [N]$ from Bhandari et al. (2018). $\qquad\square$

Next, we prove an analog of the Lipschitz property in Lemma 5 for the random TD(0) update direction of each agent $i$.

**Lemma 6.** *(2-Lipschitzness of random TD(0) update direction) For any $\theta_1, \theta_2 \in \mathbb{R}^d$ and $i \in [N]$, we have*

$$\|g_i(\theta_1) - g_i(\theta_2)\| \leq 2\|\theta_1 - \theta_2\|.$$

*Proof.* In this proof, we will use the fact that the random TD(0) update direction of agent $i$ at the $t$-th communication round and $k$-th local update is an affine function of the parameter $\theta$. In particular, we have $g_i(\theta) = b_i(O_{t,k}^{(i)}) - A_i(O_{t,k}^{(i)})\theta$, where $A_i(O_{t,k}^{(i)}) = \phi(s_{t,k}^{(i)})(\phi^\top(s_{t,k}^{(i)}) - \gamma\phi^\top(s_{t,k+1}^{(i)}))$ and $b_i(O_{t,k}^{(i)}) = r(s_{t,k}^{(i)})\phi(s_{t,k}^{(i)})$. Thus, we have

$$\begin{aligned}
\|g_i(\theta_1) - g_i(\theta_2)\| &= \left\|A_i(O_{t,k}^{(i)})(\theta_1 - \theta_2)\right\| \\
&\leq \left\|A_i(O_{t,k}^{(i)})\right\| \|\theta_1 - \theta_2\| \\
&\leq \left(\left\|\phi\left(s_{t,k}^i\right)\right\|^2 + \gamma\left\|\phi\left(s_{t,k}^i\right)\right\|\left\|\phi\left(s_{t,k+1}^i\right)\right\|\right)\|\theta_1 - \theta_2\| \\
&\leq 2\|\theta_1 - \theta_2\|,
\end{aligned}$$

where we used that $\|\phi(s)\| \leq 1, \forall s \in \mathcal{S}$ in the last step. $\qquad\square$

## F    Notation

For our subsequent analysis, we will use $\mathcal{F}_k^t$ to denote the filtration that captures all the randomness up to the $k$-th local step in round $t$. We will also use $\mathcal{F}^t$ to represent the filtration capturing all the randomness up to the end of round $t-1$. With a slight abuse of notation, $\mathcal{F}_{-1}^t$ is to be interpreted as $\mathcal{F}^t$. Based on the description of FedTD(0), it should be apparent that for each $i \in [N]$, $\theta_{t,k}^{(i)}$ is $\mathcal{F}_{k-1}^t$-measurable and $\bar{\theta}_t$ is $\mathcal{F}^t$-measurable. Furthermore, we use $\mathbb{E}_t$ to represent the expectation conditioned on all the randomness up to the end of round $t-1$.

For simplicity, we define $\delta_t = \frac{1}{NK} \sum_{i=1}^{N} \sum_{k=0}^{K-1} \left\| \theta_{t,k}^{(i)} - \bar{\theta}_t \right\|$ and $\Delta_t = \frac{1}{NK} \sum_{i=1}^{N} \sum_{k=0}^{K-1} \left\| \theta_{t,k}^{(i)} - \bar{\theta}_t \right\|^2$. The latter term is referred to as the *drift term*. Note that $(\delta_t)^2 \leq \Delta_t$ holds for all $t$ via Jensen's inequality. Unless specified otherwise, $\|\cdot\|$ denotes the Euclidean norm.

**Step-size:**    Throughout the paper, we encounter three kinds of step-sizes: local step-size $\alpha_l$, global step-size $\alpha_g$, and the effective step-size $\alpha$. Some of our results will rely on effective step-sizes that decay as a function of the communication round $t$; we will use $\{\alpha_t\}$ to represent such a decaying effective step-size sequence. While the local step-size $\alpha_\ell$ will always be held constant, the decay in the effective step-size will be achieved by making the global step-size at the server decay with the communication round. Accordingly, we will use $\{\alpha_g^{(t)}\}$ to represent the decaying global step-size sequence at the server. In what follows, unless specified in the subscript, all the step-sizes appearing in the proofs refer to the effective step-size.

# G   Warm-up: Analysis of `FedTD` under i.i.d. sampling

To isolate the effect of heterogeneity and provide key insights regarding our main proof ideas, we will analyze a simpler i.i.d. setting in this section. Specifically, we assume that for each agent $i \in [N]$, the data tuples $\{O_{t,k}^{(i)}\}$ are sampled i.i.d. from the stationary distribution $\pi^{(i)}$ of the Markov matrix $P^{(i)}$. Such an i.i.d assumption is common in the finite-time analysis of RL algorithms (Dalal et al., 2018; Bhandari et al., 2018; Doan et al., 2019). To proceed, for a fixed $\theta$ and for each $i \in [N]$, let us define $\bar{g}_i(\theta) \triangleq \mathbb{E}_{O_{t,k}^{(i)} \sim \pi^{(i)}} [g_i(\theta)]$ as the

expected `TD`(0) update direction at iterate $\theta$ when the Markov tuple $O_{t,k}^{(i)}$ hits its stationary distribution $\pi^{(i)}$. We make the following standard bounded variance assumption (Bhandari et al., 2018); similar assumptions are also made in FL analyses.

**Assumption 4.** $\mathbb{E}\|g_i(\theta) - \bar{g}_i(\theta)\|^2 \leq \sigma^2$ *holds for all agents $i \in [N]$, in each round $t$ and local update $k$, and $\forall \theta$.*

Let $H$ denote the radius of the set $\mathcal{H}$. Also, define $G \triangleq R_{\max} + 2H$ and $\nu = (1-\gamma)\bar{\omega}$, where $\bar{\omega}$ is as in Proposition 3. Our convergence result for `FedTD`(0) in the i.i.d. setting is as follows.

**Theorem 4.** *(I.I.D. Setting) There exists a decreasing global step-size sequence $\{\alpha_g^{(t)}\}$, a fixed local step-size $\alpha_l$, and a set of convex weights, such that a convex combination $\tilde{\theta}_T$ of the global models $\{\bar{\theta}_t\}$ satisfies the following for each $i \in [N]$ after $T$ rounds:*

$$\mathbb{E}\left\|V_{\tilde{\theta}_T} - V_{\theta_i^*}\right\|_{\bar{D}}^2 \leq \tilde{\mathcal{O}}\Big(\frac{G^2}{K^2 T^2} + \frac{\sigma^2}{\nu^2 NKT} + \frac{\sigma^2}{\nu^4 KT^2} + Q(\epsilon, \epsilon_1)\Big), \tag{19}$$

*where $Q(\epsilon, \epsilon_1) = \tilde{\mathcal{O}}(\frac{B(\epsilon,\epsilon_1)G}{\nu} + \Gamma^2(\epsilon, \epsilon_1))$, $B(\epsilon, \epsilon_1) = H\left(\sqrt{n}\epsilon + 2(n-1)\epsilon + \mathcal{O}(\epsilon^2) + \mathcal{O}(\epsilon_1)\right)$, and $\Gamma(\epsilon, \epsilon_1)$ is as defined in Theorem 1.*

In what follows, we provide a detailed convergence analysis of the above result.

## G.1   Auxiliary lemmas for Theorem 4

### G.1.1   Variance reduction

**Lemma 7.** *(Variance reduction in the i.i.d. setting). In the i.i.d. setting, under Assumption 4, at each round $t$, we have $\mathbb{E}\left\|\frac{1}{NK} \sum_{i=1}^{N} \sum_{k=0}^{K-1} \left[g_i(\theta_{t,k}^{(i)}) - \bar{g}_i(\theta_{t,k}^{(i)})\right]\right\|^2 \leq \frac{\sigma^2}{NK}$.*

*Proof.* Define $Y_{t,k}^{(i)} \triangleq g_i(O_{t,k}^{(i)}, \theta_{t,k}^{(i)}) - \bar{g}_i(\theta_{t,k}^{(i)})$. Since $\{O_{t,k}^{(i)}\}$ is drawn i.i.d. over time from its stationary distribution $\pi^{(i)}$, we have $\mathbb{E}[Y_{t,k}^{(i)}] = \mathbb{E}\left[\mathbb{E}[Y_{t,k}^{(i)} \mid \theta_{t,k}^{(i)}]\right] = 0$. As we mentioned before, for each $i \in [N]$, $\theta_{t,k}^{(i)}$ is $\mathcal{F}_{k-1}^t$-measurable. If we condition on $\mathcal{F}_{k-1}^t$, we know that $\theta_{t,k}^{(i)}$ and $\theta_{t,k}^{(j)}$ are deterministic and the only randomness in $Y_{t,k}^{(i)}$ and $Y_{t,k}^{(j)}$ come from $O_{t,k}^{(i)}$ and $O_{t,k}^{(j)}$, which are independent. Therefore, $Y_{t,k}^{(i)}$ and $Y_{t,k}^{(j)}$ are independent conditioned on $\mathcal{F}_{k-1}^t$.

For every $i \neq j \in [N]$, we have

$$\mathbb{E}\left[\left\langle Y_{t,k}^{(i)}, Y_{t,k}^{(j)}\right\rangle\right] = \mathbb{E}\left[\mathbb{E}\left[\left\langle Y_{t,k}^{(i)}, Y_{t,k}^{(j)}\right\rangle \mid \mathcal{F}_{k-1}^t\right]\right] \stackrel{(a)}{=} \mathbb{E}\left[\left\langle \mathbb{E}[Y_{t,k}^{(i)} \mid \mathcal{F}_{k-1}^t], \mathbb{E}[Y_{t,k}^{(j)} \mid \mathcal{F}_{k-1}^t]\right\rangle\right] = 0, \tag{20}$$

where (a) follows from the fact that $Y_{t,k}^{(i)}$ and $Y_{t,k}^{(j)}$ are independent conditioned on $\mathcal{F}_{k-1}^t$. For every $k < l$ and $i, j \in [N]$,

$$\mathbb{E}\left[\left\langle Y_{t,k}^{(i)}, Y_{t,l}^{(j)}\right\rangle\right] = \mathbb{E}\left[\mathbb{E}\left[\left\langle Y_{t,k}^{(i)}, Y_{t,l}^{(j)}\right\rangle \,\middle|\, \mathcal{F}_{l-1}^t\right]\right] = \mathbb{E}\left[\left\langle Y_{t,k}^{(i)}, \mathbb{E}[Y_{t,l}^{(j)} \mid \mathcal{F}_{l-1}^t]\right\rangle\right] = 0. \tag{21}$$

Then,

$$\mathbb{E}\left\|\frac{1}{NK} \sum_{i=1}^{N} \sum_{k=0}^{K-1} \left[g_i(\theta_{t,k}^{(i)}) - \bar{g}_i(\theta_{t,k}^{(i)})\right]\right\|^2$$

$$= \mathbb{E}\left\|\frac{1}{NK}\sum_{i=1}^{N}\sum_{k=0}^{K-1}Y_{t,k}^{(i)}\right\|^2$$

$$= \frac{1}{N^2K^2}\sum_{i=1}^{N}\sum_{k=0}^{K-1}\mathbb{E}\|Y_{t,k}^{(i)}\|^2 + \frac{2}{N^2K^2}\underbrace{\sum_{i<j}\sum_{k=0}^{K-1}\mathbb{E}[\langle Y_{t,k}^{(i)}, Y_{t,k}^{(j)}\rangle]}_{0}$$

$$+ \frac{2}{N^2K^2}\sum_{i,j=1}^{N}\sum_{k<l}\underbrace{\mathbb{E}[\langle Y_{t,k}^{(i)}, Y_{t,l}^{(j)}\rangle]}_{0}$$

$$\leq \frac{\sigma^2}{NK},$$

where the second equality is due to Eq. (20)) and Eq. (21) and the last inequality is due to Assumption 4. $\quad\square$

### G.1.2 Per Round Progress

First, we characterize the error decrease at each iteration in the following lemma.

**Lemma 8.** *(Per Round Progress). If the local step-size $\alpha_l$ satisfies $\alpha_l \leq \frac{(1-\gamma)\bar{\omega}}{48K}$, then the updates of* `FedTD(0)` *with any global step-size $\alpha_g$ satisfy*

$$\mathbb{E}\|\bar{\theta}_{t+1} - \theta^*\|^2 \leq (1+\zeta_1)\mathbb{E}\left\|\bar{\theta}_t - \theta^*\right\|^2 + 2\alpha\mathbb{E}\langle\bar{g}(\bar{\theta}_t),\bar{\theta}_t - \theta^*\rangle + 6\alpha^2\mathbb{E}\left\|\bar{g}(\bar{\theta}_t)\right\|^2$$

$$+ 4\alpha^2\left(\frac{1}{\zeta_1}+6\right)\mathbb{E}[\Delta_t] + \frac{2\alpha^2\sigma^2}{NK} + 2\alpha B(\epsilon,\epsilon_1)G + 6\alpha^2 B^2(\epsilon,\epsilon_1), \tag{22}$$

*where $\zeta_1$ is any positive constant, and $\alpha$ is the effective step-size, i.e., $\alpha = K\alpha_l\alpha_g$.*

*Proof.*

$$\mathbb{E}\|\bar{\theta}_{t+1} - \theta^*\|^2$$

$$= \mathbb{E}\left\|\Pi_{2,\mathcal{H}}\left(\bar{\theta}_t + \frac{\alpha}{NK}\sum_{i=1}^{N}\sum_{k=0}^{K-1}g_i(\theta_{t,k}^{(i)}) - \theta^*\right)\right\|^2 \quad \text{(updating rule)}$$

$$\leq \mathbb{E}\left\|\bar{\theta}_t + \frac{\alpha}{NK}\sum_{i=1}^{N}\sum_{k=0}^{K-1}g_i(\theta_{t,k}^{(i)}) - \theta^*\right\|^2 \quad \text{(projection is non-expansive)}$$

$$= \mathbb{E}\left\|\bar{\theta}_t - \theta^*\right\|^2 + 2\mathbb{E}\langle\frac{\alpha}{NK}\sum_{i=1}^{N}\sum_{k=0}^{K-1}g_i(\theta_{t,k}^{(i)}),\bar{\theta}_t - \theta^*\rangle + \mathbb{E}\left\|\frac{\alpha}{NK}\sum_{i=1}^{N}\sum_{k=0}^{K-1}g_i(\theta_{t,k}^{(i)})\right\|^2$$

$$= \mathbb{E}\left\|\bar{\theta}_t - \theta^*\right\|^2 + \frac{2\alpha}{NK}\underbrace{\sum_{i=1}^{N}\sum_{k=0}^{K-1}\mathbb{E}\langle g_i(\theta_{t,k}^{(i)}) - \bar{g}_i(\theta_{t,k}^{(i)}),\bar{\theta}_t - \theta^*\rangle}_{\mathcal{C}_1=0}$$

$$+ \frac{2\alpha}{NK}\sum_{i=1}^{N}\sum_{k=0}^{K-1}\mathbb{E}\langle\bar{g}_i(\theta_{t,k}^{(i)}),\bar{\theta}_t - \theta^*\rangle + \mathbb{E}\left\|\frac{\alpha}{NK}\sum_{i=1}^{N}\sum_{k=0}^{K-1}g_i(\theta_{t,k}^{(i)})\right\|^2$$

$$= \mathbb{E}\left\|\bar{\theta}_t - \theta^*\right\|^2 + \frac{2\alpha}{NK}\sum_{i=1}^{N}\sum_{k=0}^{K-1}\mathbb{E}\langle\bar{g}_i(\theta_{t,k}^{(i)}),\bar{\theta}_t - \theta^*\rangle + \mathbb{E}\left\|\frac{\alpha}{NK}\sum_{i=1}^{N}\sum_{k=0}^{K-1}g_i(\theta_{t,k}^{(i)})\right\|^2$$

$$\leq \mathbb{E}\left\|\bar{\theta}_t - \theta^*\right\|^2 + \frac{2\alpha}{NK}\sum_{i=1}^{N}\sum_{k=0}^{K-1}\mathbb{E}\langle\bar{g}_i(\theta_{t,k}^{(i)}),\bar{\theta}_t - \theta^*\rangle$$

$$+ 2\mathbb{E}\left\|\frac{\alpha}{NK}\sum_{i=1}^{N}\sum_{k=0}^{K-1}\left[g_i(\theta_{t,k}^{(i)}) - \bar{g}_i(\theta_{t,k}^{(i)})\right]\right\|^2 + 2\mathbb{E}\left\|\frac{\alpha}{NK}\sum_{i=1}^{N}\bar{g}_i(\theta_{t,k}^{(i)})\right\|^2 \quad \text{(Young's inequality (12))}$$

$$\overset{(a)}{\leq} \mathbb{E}\left\|\bar{\theta}_t - \theta^*\right\|^2 + \frac{2\alpha}{NK} \sum_{i=1}^{N} \sum_{k=0}^{K-1} \mathbb{E}\langle \bar{g}_i(\theta_{t,k}^{(i)}), \bar{\theta}_t - \theta^* \rangle + \frac{2\sigma^2}{NK} + 2\mathbb{E}\left\|\frac{\alpha}{NK} \sum_{i=1}^{N} \bar{g}_i(\theta_{t,k}^{(i)})\right\|^2$$

$$= \mathbb{E}\left\|\bar{\theta}_t - \theta^*\right\|^2 + \frac{2\alpha}{NK} \sum_{i=1}^{N} \sum_{k=0}^{K-1} \mathbb{E}\langle \bar{g}_i(\theta_{t,k}^{(i)}) - \bar{g}_i(\bar{\theta}_t) + \bar{g}_i(\bar{\theta}_t) - \bar{g}(\bar{\theta}_t) + \bar{g}(\bar{\theta}_t), \bar{\theta}_t - \theta^* \rangle$$

$$+ 2\mathbb{E}\left\|\frac{\alpha}{NK} \sum_{i=1}^{N} \sum_{k=0}^{K-1} \bar{g}_i(\theta_{t,k}^{(i)})\right\|^2 + \frac{2\alpha^2\sigma^2}{NK}$$

$$\leq \mathbb{E}\left\|\bar{\theta}_t - \theta^*\right\|^2 + \frac{2\alpha}{NK} \sum_{i=1}^{N} \sum_{k=0}^{K-1} \mathbb{E}\langle \bar{g}_i(\theta_{t,k}^{(i)}) - \bar{g}_i(\bar{\theta}_t), \bar{\theta}_t - \theta^* \rangle + \frac{2\alpha}{N} \sum_{i=1}^{N} \mathbb{E}\langle \bar{g}_i(\bar{\theta}_t) - \bar{g}(\bar{\theta}_t), \bar{\theta}_t - \theta^* \rangle$$

$$+ 2\alpha\mathbb{E}\langle \bar{g}(\bar{\theta}_t), \bar{\theta}_t - \theta^* \rangle + 2\mathbb{E}\left\|\frac{\alpha}{NK} \sum_{i=1}^{N} \sum_{k=0}^{K-1} \bar{g}_i(\theta_{t,k}^{(i)})\right\|^2 + \frac{2\alpha^2\sigma^2}{NK}$$

$$\leq (1+\zeta_1)\mathbb{E}\left\|\bar{\theta}_t - \theta^*\right\|^2 + \frac{1}{\zeta_1}\mathbb{E}\left\|\frac{\alpha}{NK} \sum_{i=1}^{N} \sum_{k=0}^{K-1} \left[\bar{g}_i(\theta_{t,k}^{(i)}) - \bar{g}_i(\bar{\theta}_t)\right]\right\|^2 + 2\alpha B(\epsilon, \epsilon_1) G$$

$$+ 2\alpha\mathbb{E}\langle \bar{g}(\bar{\theta}_t), \bar{\theta}_t - \theta^* \rangle + 2\mathbb{E}\left\|\frac{\alpha}{NK} \sum_{i=1}^{N} \sum_{k=0}^{K-1} \bar{g}_i(\theta_{t,k}^{(i)})\right\|^2 + \frac{2\alpha^2\sigma^2}{NK} \quad \text{(Eq (12) and Lemma 2)}$$

$$\leq (1+\zeta_1)\mathbb{E}\left\|\bar{\theta}_t - \theta^*\right\|^2 + \frac{4\alpha^2}{\zeta_1 NK} \sum_{i=1}^{N} \sum_{k=0}^{K-1} \mathbb{E}\left\|\theta_{t,k}^{(i)} - \bar{\theta}_t\right\|^2 + 2\alpha B(\epsilon, \epsilon_1) G$$

$$+ 2\alpha\mathbb{E}\langle \bar{g}(\bar{\theta}_t), \bar{\theta}_t - \theta^* \rangle + 2\mathbb{E}\left\|\frac{\alpha}{NK} \sum_{i=1}^{N} \sum_{k=0}^{K-1} \bar{g}_i(\theta_{t,k}^{(i)})\right\|^2 + \frac{2\alpha^2\sigma^2}{NK} \quad \text{(2-Lipschitz of } \bar{g}_i \text{ in Lemma 5)}$$

$$\leq (1+\zeta_1)\mathbb{E}\left\|\bar{\theta}_t - \theta^*\right\|^2 + \frac{4\alpha^2}{\zeta_1}\mathbb{E}[\Delta_t] + \frac{2\alpha^2\sigma^2}{NK} + 2\alpha B(\epsilon, \epsilon_1) G$$

$$+ 2\alpha\mathbb{E}\langle \bar{g}(\bar{\theta}_t), \bar{\theta}_t - \theta^* \rangle + 2\mathbb{E}\left\|\frac{\alpha}{NK} \sum_{i=1}^{N} \sum_{k=0}^{K-1} \left[\bar{g}_i(\theta_{t,k}^{(i)}) - \bar{g}_i(\bar{\theta}_t) + \bar{g}_i(\bar{\theta}_t) - \bar{g}(\bar{\theta}_t) + \bar{g}(\bar{\theta}_t)\right]\right\|^2$$

$$\leq (1+\zeta_1)\mathbb{E}\left\|\bar{\theta}_t - \theta^*\right\|^2 + \frac{4\alpha^2}{\zeta_1}\mathbb{E}[\Delta_t] + \frac{2\alpha^2\sigma^2}{NK} + 2\alpha B(\epsilon, \epsilon_1) G$$

$$+ 2\alpha\mathbb{E}\langle \bar{g}(\bar{\theta}_t), \bar{\theta}_t - \theta^* \rangle + 6\mathbb{E}\left\|\frac{\alpha}{NK} \sum_{i=1}^{N} \sum_{k=0}^{K-1} \bar{g}_i(\theta_{t,k}^{(i)}) - \bar{g}_i(\bar{\theta}_t)\right\|^2$$

$$+ 6\mathbb{E}\left\|\frac{\alpha}{N} \sum_{i=1}^{N} \left[\bar{g}_i(\bar{\theta}_t) - \bar{g}(\bar{\theta}_t)\right]\right\|^2 + 6\mathbb{E}\left\|\alpha \bar{g}(\bar{\theta}_t)\right\|^2 \quad \text{(Eq (12) and Lemma 2)}$$

$$\leq (1+\zeta_1)\mathbb{E}\left\|\bar{\theta}_t - \theta^*\right\|^2 + \frac{4\alpha^2}{\zeta_1}\mathbb{E}[\Delta_t] + \frac{2\alpha^2\sigma^2}{NK} + 2\alpha B(\epsilon, \epsilon_1) G$$

$$+ 2\alpha\mathbb{E}\langle \bar{g}(\bar{\theta}_t), \bar{\theta}_t - \theta^* \rangle + 24\alpha^2\mathbb{E}[\Delta_t] \quad \text{(2-Lipschitz of } \bar{g}_i)$$

$$+ 6\alpha^2 B^2(\epsilon, \epsilon_1) + 6\alpha^2\mathbb{E}\left\|\bar{g}(\bar{\theta}_t)\right\|^2 \quad \text{(Eq (12))}$$

$$= (1+\zeta_1)\mathbb{E}\left\|\bar{\theta}_t - \theta^*\right\|^2 + 2\alpha\mathbb{E}\langle \bar{g}(\bar{\theta}_t), \bar{\theta}_t - \theta^* \rangle + 6\alpha^2\mathbb{E}\left\|\bar{g}(\bar{\theta}_t)\right\|^2$$

$$+ 4\alpha^2\left(\frac{1}{\zeta_1} + 6\right)\mathbb{E}[\Delta_t] + \frac{2\alpha^2\sigma^2}{NK} + 2\alpha B(\epsilon, \epsilon_1) G + 6\alpha^2 B^2(\epsilon, \epsilon_1), \tag{23}$$

where (a) is due to Lemma 7. Furthermore, the reason why $\mathcal{C}_1 = 0$ is as follows:

$$\mathcal{C}_1 = \sum_{i=1}^{N} \sum_{k=0}^{K-1} \mathbb{E}\langle g_i(\theta_{t,k}^{(i)}) - \bar{g}_i(\theta_{t,k}^{(i)}), \bar{\theta}_t - \theta^* \rangle$$

$$
\begin{aligned}
&= \sum_{i=1}^{N} \sum_{k=0}^{K-2} \mathbb{E}\langle g_i(\theta_{t,k}^{(i)}) - \bar{g}_i(\theta_{t,k}^{(i)}), \bar{\theta}_t - \theta^* \rangle + \\
&\qquad \sum_{i=1}^{N} \mathbb{E}\langle g_i(\theta_{t,K-1}^{(i)}) - \bar{g}_i(\theta_{t,K-1}^{(i)}), \bar{\theta}_t - \theta^* \rangle \\
&= \sum_{i=1}^{N} \sum_{k=0}^{K-2} \mathbb{E}\langle g_i(\theta_{t,k}^{(i)}) - \bar{g}_i(\theta_{t,k}^{(i)}), \bar{\theta}_t - \theta^* \rangle + \\
&\qquad \sum_{i=1}^{N} \mathbb{E}\left[ \mathbb{E}\left[ \langle g_i(\theta_{t,K-1}^{(i)}) - \bar{g}_i(\theta_{t,K-1}^{(i)}), \bar{\theta}_t - \theta^* \rangle \mid \mathcal{F}_{K-1}^t \right] \right] \\
&= \sum_{i=1}^{N} \sum_{k=0}^{K-2} \mathbb{E}\langle g_i(\theta_{t,k}^{(i)}) - \bar{g}_i(\theta_{t,k}^{(i)}), \bar{\theta}_t - \theta^* \rangle + \\
&\qquad \sum_{i=1}^{N} \mathbb{E}\left[ \left\langle \bar{\theta}_t - \theta^*, \underbrace{\mathbb{E}\left[ g_i(\theta_{t,k}^{(i)}) - \bar{g}_i(\theta_{t,k}^{(i)}) \mid \mathcal{F}_{K-1}^t \right]}_{0} \right\rangle \right] \\
&= \sum_{i=1}^{N} \sum_{k=0}^{K-2} \mathbb{E}\langle g_i(\theta_{t,k}^{(i)}) - \bar{g}_i(\theta_{t,k}^{(i)}), \bar{\theta}_t - \theta^* \rangle.
\end{aligned}
$$

We can keep repeating this procedure by iteratively conditioning on $\mathcal{F}_{K-2}^t, \cdots, \mathcal{F}_1^t, \mathcal{F}_0^t$. $\qquad\square$

### G.1.3 Drift Term Analysis

We now turn to bounding the drift term $\Delta_t$.

**Lemma 9.** *(Bounded Client Drift) The drift term $\Delta_t$ at the $t$-th round can be bounded as*

$$
\mathbb{E}[\Delta_t] = \frac{1}{NK} \sum_{i=1}^{N} \sum_{k=0}^{K-1} \mathbb{E}\left\| \theta_{t,k}^{(i)} - \bar{\theta}_t \right\|^2 \leq 27(\sigma^2 + 3KB^2(\epsilon, \epsilon_1) + 2KG^2) \frac{\alpha^2}{K\alpha_g^2}, \tag{24}
$$

*provided the fixed local step-size $\alpha_l$ satisfies $\alpha_l \leq \min \frac{(1-\gamma)\bar{\omega}}{48K}$.*

*Proof.*

$$
\begin{aligned}
&\mathbb{E}\left\| \theta_{t,k}^{(i)} - \bar{\theta}_t \right\|^2 \\
&= \mathbb{E}\left\| \theta_{t,k-1}^{(i)} + \alpha_l g_i(\theta_{t,k-1}^{(i)}) - \bar{\theta}_t \right\|^2 \quad \text{(updating rule)} \\
&= \mathbb{E}\left\| \theta_{t,k-1}^{(i)} + \alpha_l \bar{g}_i(\theta_{t,k-1}^{(i)}) - \bar{\theta}_t + \alpha_l \left( g_i(\theta_{t,k-1}^{(i)}) - \bar{g}_i(\theta_{t,k-1}^{(i)}) \right) \right\|^2 \\
&= \mathbb{E}\left\| \theta_{t,k-1}^{(i)} + \alpha_l \bar{g}_i(\theta_{t,k-1}^{(i)}) - \bar{\theta}_t \right\|^2 + \alpha_l^2 \mathbb{E}\left\| g_i(\theta_{t,k-1}^{(i)}) - \bar{g}_i(\theta_{t,k-1}^{(i)}) \right\|^2 \\
&\quad + 2\alpha_l \underbrace{\mathbb{E}\left[ \mathbb{E}\left\langle g_i(\theta_{t,k-1}^{(i)}) - \bar{g}_i(\theta_{t,k-1}^{(i)}), \theta_{t,k-1}^{(i)} + \alpha_l \bar{g}_i(\theta_{t,k-1}^{(i)}) - \bar{\theta}_t \mid \mathcal{F}_{k-1}^t \right\rangle \right]}_{\mathcal{C}_2 = 0} \\
&\overset{(a)}{\leq} (1 + \zeta_2)\mathbb{E}\left\| \theta_{t,k-1}^{(i)} + \alpha_l \bar{g}_i(\theta_{t,k-1}^{(i)}) - \bar{\theta}_t \right\|^2 + (1 + \frac{1}{\zeta_2})\alpha_l^2 \mathbb{E}\left\| \bar{g}(\theta_{t,k-1}^{(i)}) - \bar{g}_i(\theta_{t,k-1}^{(i)}) \right\|^2 \\
&\quad + \alpha_l^2 \mathbb{E}\left\| g_i(\theta_{t,k-1}^{(i)}) - \bar{g}_i(\theta_{t,k-1}^{(i)}) \right\|^2 \\
&\overset{(b)}{\leq} (1 + \zeta_2)(1 + \zeta_3)\mathbb{E}\left\| \theta_{t,k-1}^{(i)} + \alpha_l \bar{g}_i(\theta_{t,k-1}^{(i)}) - \bar{\theta}_t - \alpha_l \bar{g}(\bar{\theta}_t) \right\|^2
\end{aligned}
$$

$$+ (1 + \zeta_2)(1 + \frac{1}{\zeta_3})\alpha_l^2 \mathbb{E}\left\| \bar{g}(\bar{\theta}_t) \right\|^2$$

$$+ (1 + \frac{1}{\zeta_2})\alpha_l^2 \mathbb{E}\left\| \bar{g}(\theta_{t,k-1}^{(i)}) - \bar{g}(\bar{\theta}_t) + \bar{g}(\bar{\theta}_t) - \bar{g}_i(\bar{\theta}_t) + \bar{g}_i(\bar{\theta}_t) - \bar{g}_i(\theta_{t,k-1}^{(i)}) \right\|^2 + \alpha_l^2 \sigma^2$$

$$\overset{(c)}{\leq} (1 + \zeta_2)(1 + \zeta_3)\mathbb{E}\left\| \theta_{t,k-1}^{(i)} + \alpha_l \bar{g}(\theta_{t,k-1}^{(i)}) - \bar{\theta}_t - \alpha_l \bar{g}(\bar{\theta}_t) \right\|^2 + (1 + \zeta_2)(1 + \frac{1}{\zeta_3})\alpha_l^2 \mathbb{E}\left\| \bar{g}(\bar{\theta}_t) \right\|^2$$

$$+ 3(1 + \frac{1}{\zeta_2})\alpha_l^2 \mathbb{E}\left\| \bar{g}(\theta_{t,k-1}^{(i)}) - \bar{g}(\bar{\theta}_t) \right\|^2 + 3(1 + \frac{1}{\zeta_2})\alpha_l^2 \mathbb{E}\left\| \bar{g}(\bar{\theta}_t) - \bar{g}_i(\bar{\theta}_t) \right\|^2$$

$$+ 3(1 + \frac{1}{\zeta_2})\alpha_l^2 \mathbb{E}\left\| \bar{g}_i(\bar{\theta}_t) - \bar{g}_i(\theta_{t,k-1}^{(i)}) \right\|^2 + \alpha_l^2 \sigma^2$$

$$\overset{(d)}{\leq} (1 + \zeta_2)(1 + \zeta_3)\left[ 1 - (2\alpha_l(1 - \gamma) - 4\alpha_l^2)\bar{\omega} \right] \mathbb{E}\left\| \theta_{t,k-1}^{(i)} - \bar{\theta}_t \right\|^2$$

$$+ (1 + \zeta_2)(1 + \frac{1}{\zeta_3})\alpha_l^2 \mathbb{E}\left\| \bar{g}(\bar{\theta}_t) \right\|^2$$

$$+ 12(1 + \frac{1}{\zeta_2})\alpha_l^2 \mathbb{E}\left\| \theta_{t,k-1}^{(i)} - \bar{\theta}_t \right\|^2 + 3(1 + \frac{1}{\zeta_3})\alpha_l^2 B^2(\epsilon, \epsilon_1)$$

$$+ 12(1 + \frac{1}{\zeta_3})\alpha_l^2 \mathbb{E}\left\| \theta_{t,k-1}^{(i)} - \bar{\theta}_t \right\|^2 + \alpha_l^2 \sigma^2$$

$$= (1 + \zeta_2)(1 + \zeta_3)\left[ 1 - (2\alpha_l(1 - \gamma) - 4\alpha_l^2)\bar{\omega} + \frac{24(1 + \frac{1}{\zeta_3})\alpha_l^2}{(1 + \zeta_2)(1 + \zeta_3)} \right] \mathbb{E}\left\| \theta_{t,k-1}^{(i)} - \bar{\theta}_t \right\|^2$$

$$+ \underbrace{(1 + \zeta_2)(1 + \frac{1}{\zeta_3})\alpha_l^2 \mathbb{E}\left\| \bar{g}(\bar{\theta}_t) \right\|^2 + 3(1 + \frac{1}{\zeta_3})\alpha_l^2 B^2(\epsilon, \epsilon_1) + \alpha_l^2 \sigma^2}_{\mathcal{D}_1},$$

where we used the inequality in Eq (11) with any positive constant $\zeta_2$ for (a); for (b), we used Assumption 4 and the same reasoning as Eq (11) with any positive constant $\zeta_3$; for (c), we used the inequality in Eq (13) to bound the third term; and for (d), we used Lemma 3 and Lemma 4 to bound the first term, the 2-Lipschitz property of $\bar{g}$, $\bar{g}_i$ (i.e., Lemma 5) in the third term and the fifth term, and the gradient heterogeneity bound from Lemma 2 in the fourth term. If we define $\zeta_4 \triangleq (1 + \zeta_2)(1 + \zeta_3)\left[ 1 - (2\alpha_l(1 - \gamma) - 4\alpha_l^2)\bar{\omega} + \frac{24(1 + \frac{1}{\zeta_3})\alpha_l^2}{(1 + \zeta_2)(1 + \zeta_3)} \right]$ and define $\mathcal{D}_1$ as above, we have that

$$\mathbb{E}\left\| \theta_{t,k}^{(i)} - \bar{\theta}_t \right\|^2 \leq \zeta_4 \mathbb{E}\left\| \theta_{t,k-1}^{(i)} - \bar{\theta}_t \right\|^2 + \mathcal{D}_1. \tag{25}$$

Next, we set $\zeta_2 = \zeta_3 = \frac{1}{K-1}, K \geq 2$, and choose the local step-size $\alpha_l$ to satisfy

$$\frac{\alpha_l(1 - \gamma)\bar{\omega}}{2} \geq 4\alpha_l^2 \bar{\omega} \quad \& \quad \frac{\alpha_l(1 - \gamma)\bar{\omega}}{2} \geq \frac{24(1 + \frac{1}{\zeta_3})\alpha_l^2}{(1 + \zeta_2)(1 + \zeta_3)},$$

so that $\left[ 1 - (2\alpha_l(1 - \gamma) - 4\alpha_l^2)\bar{\omega} + \frac{24(1 + \frac{1}{\zeta_3})\alpha_l^2}{(1 + \zeta_2)(1 + \zeta_3)} \right] \leq 1 - \alpha_l(1 - \gamma)\bar{\omega}$. These inequalities hold when $\alpha_l \leq$ min $\frac{(1 - \gamma)\bar{\omega}}{48K}$. Then, Eq (25) becomes

$$\mathbb{E}\left\| \theta_{t,k}^{(i)} - \bar{\theta}_t \right\|^2 \leq (1 + \frac{3}{K-1})\left[ 1 - \alpha_l(1 - \gamma)\bar{\omega} \right] \mathbb{E}\left\| \theta_{t,k-1}^{(i)} - \bar{\theta}_t \right\|^2 + \mathcal{D}_1.$$

If we unroll this recurrence above, using $\theta_{r,0}^{(i)} = \bar{\theta}_t$, we have that

$$\mathbb{E}\left\| \theta_{t,k}^{(i)} - \bar{\theta}_t \right\|^2 \leq \sum_{s=0}^{k-1} \mathcal{D}_1 \left\{ \Pi_{j=s+1}^{k-1}(1 + \frac{3}{K-1})\left[ 1 - \alpha(1 - \gamma)\bar{\omega} \right] \right\}$$

$$\overset{(e)}{\leq} \sum_{s=0}^{k-1} \left[ \alpha_l^2 \sigma^2 + 3K\alpha_l^2 B^2(\epsilon, \epsilon_1), + 2\alpha_l^2 K \mathbb{E}\left\| \bar{g}(\bar{\theta}_t) \right\|^2 \right]$$

$$\times \, \Pi_{j=s+1}^{k-1}(1 + \frac{3}{K-1})[1 - \alpha_l(1-\gamma)\bar{\omega}]$$

$$\leq \sum_{s=0}^{k-1} \left[ \alpha_l^2 \sigma^2 + 3\alpha_l^2 K B^2(\epsilon, \epsilon_1) + 2\alpha_l^2 K \mathbb{E}\left\|\bar{g}(\bar{\theta}_t)\right\|^2 \right]$$

$$\times \, (1 + \frac{3}{K-1})^{K-1}\Pi_{j=s+1}^{k-1}[1 - \alpha_l(1-\gamma)\bar{\omega}]$$

$$\stackrel{(f)}{\leq} 27(\sigma^2 + 3KB^2(\epsilon, \epsilon_1) + 2K\mathbb{E}\left\|\bar{g}(\bar{\theta}_t)\right\|^2) \sum_{s=0}^{k-1} \alpha_l^2 \times \underbrace{\Pi_{j=s+1}^{k-1}[1 - \alpha(1-\gamma)\bar{\omega}]}_{\leq 1}$$

$$\leq 27(\sigma^2 + 3KB^2(\epsilon, \epsilon_1) + 2KG^2)K\alpha_l^2 \quad \text{(constant local step-size)}$$

where we used the fact that $(1 + \zeta_2)(1 + \frac{1}{\zeta_3}) \leq 2K$ for $(e)$ and $(1 + \frac{3}{K-1})^{K-1} \leq 27$ for $(f)$. we finish the proof by substituting $\alpha_l = \frac{\alpha}{K\alpha_g}$. $\qquad \square$

If we incorporate Eq (24) into Eq (22), we have that

$$\mathbb{E}\left\|\bar{\theta}_{t+1} - \theta^*\right\|^2 \tag{26}$$

$$\leq (1 + \zeta_1)\mathbb{E}\left\|\bar{\theta}_r - \theta^*\right\|^2 + 2\alpha\mathbb{E}\langle \bar{g}(\bar{\theta}_r), \bar{\theta}_r - \theta^* \rangle + 6\alpha^2 \mathbb{E}\left\|\bar{g}(\bar{\theta}_r)\right\|^2$$

$$+ 108\frac{\alpha^4}{K\alpha_g^2}(6 + \frac{1}{\zeta_1})(\sigma^2 + 3KB^2(\epsilon, \epsilon_1) + 2KG^2) + \frac{2\alpha^2\sigma^2}{NK} + 2\alpha B(\epsilon, \epsilon_1)G + 6\alpha^2 B^2(\epsilon, \epsilon_1). \tag{27}$$

### G.1.4 Parameter Selection

**Lemma 10.** *Define $\nu \triangleq (1-\gamma)\bar{\omega}$. If we choose any effective step-size $\alpha = K\alpha_g\alpha_l < \frac{(1-\gamma)\bar{\omega}}{96}$, any local step-size $\alpha_l \leq \min \frac{(1-\gamma)\bar{\omega}}{48K}$, and choose the constant $\zeta_1 = \alpha\nu$, the updates of FedTD(0) satisfy*

$$\nu_1 \mathbb{E}\left\|V_{\bar{\theta}_t} - V_{\theta^*}\right\|_{\bar{D}}^2 \leq (\frac{1}{\alpha} - \nu_1)\mathbb{E}\left\|\bar{\theta}_t - \theta^*\right\|^2 - \frac{1}{\alpha}\mathbb{E}\left\|\bar{\theta}_{t+1} - \theta^*\right\|^2 + \underbrace{\frac{2\alpha\sigma^2}{NK}}_{O(\alpha^1)}$$

$$+ \underbrace{\frac{1080\alpha^2}{K\alpha_g^2\nu}(\sigma^2 + 3KB^2(\epsilon, \epsilon_1) + 2KG^2)}_{O(\alpha^2)} + \underbrace{2B(\epsilon, \epsilon_1)G + 6\alpha B^2(\epsilon, \epsilon_1)}_{heterogeneity\ term}, \tag{28}$$

*where $\nu_1 = \frac{\nu}{4} = \frac{(1-\gamma)\bar{\omega}}{4}$.*

*Proof.* From Eq (26) and $\zeta_1 = \alpha\nu$, we know

$$\mathbb{E}\left\|\bar{\theta}_{t+1} - \theta^*\right\|^2$$

$$\leq (1 + \zeta_1)\mathbb{E}\left\|\bar{\theta}_t - \theta^*\right\|^2 + 2\alpha\mathbb{E}\langle \bar{g}(\bar{\theta}_t), \bar{\theta}_t - \theta^* \rangle + 6\alpha^2 \mathbb{E}\left\|\bar{g}(\bar{\theta}_r)\right\|^2$$

$$+ 108\frac{\alpha^4}{K\alpha_g^2}(6 + \frac{1}{\zeta_1})(\sigma^2 + 3KB^2(\epsilon, \epsilon_1) + 2KG^2) + \frac{2\alpha^2\sigma^2}{NK} + 2\alpha B(\epsilon, \epsilon_1)G + 6\alpha^2 B^2(\epsilon, \epsilon_1)$$

$$\leq (1 + \alpha\nu - 2\alpha\nu)\mathbb{E}\left\|\bar{\theta}_t - \theta^*\right\|^2 + 24\alpha^2 \mathbb{E}\left\|V_{\bar{\theta}_t} - V_{\theta^*}\right\|_{\bar{D}}^2 + \frac{2\alpha^2\sigma^2}{NK} \quad \text{(Lemma 3 and 4)}$$

$$+ 108\frac{\alpha^4}{K\alpha_g^2}(6 + \frac{1}{\alpha\nu})(\sigma^2 + 3KB^2(\epsilon, \epsilon_1) + 2KG^2) + 2\alpha B(\epsilon, \epsilon_1)G + 6\alpha^2 B^2(\epsilon, \epsilon_1)$$

$$\leq (1 - \frac{\alpha\nu}{2})\mathbb{E}\left\|\bar{\theta}_t - \theta^*\right\|^2 - \frac{\alpha\nu}{2}\mathbb{E}\left\|\bar{\theta}_t - \theta^*\right\|^2 + 24\alpha^2 \mathbb{E}\left\|V_{\bar{\theta}_t} - V_{\theta^*}\right\|_{\bar{D}}^2 + \frac{2\alpha^2\sigma^2}{NK}$$

$$+ 108 \frac{\alpha^4}{K\alpha_g^2} (6 + \frac{1}{\alpha\nu})(\sigma^2 + 3KB^2(\epsilon, \epsilon_1) + 2KG^2) + 2\alpha B(\epsilon, \epsilon_1)G + 6\alpha^2 B^2(\epsilon, \epsilon_1)$$

$$\overset{(a)}{\leq} (1 - \frac{\alpha\nu}{2})\mathbb{E}\left\|\bar{\theta}_t - \theta^*\right\|^2 - \frac{\alpha\nu}{2}\mathbb{E}\left\|V_{\bar{\theta}_t} - V_{\theta^*}\right\|_{\bar{D}}^2 + \frac{\alpha\nu}{4}\mathbb{E}\left\|V_{\bar{\theta}_t} - V_{\theta^*}\right\|_{\bar{D}}^2 + \frac{2\alpha^2\sigma^2}{NK}$$

$$+ 108 \frac{\alpha^4}{K\alpha_g^2} (6 + \frac{1}{\alpha\nu})(\sigma^2 + 3KB^2(\epsilon, \epsilon_1) + 2KG^2) + 2\alpha B(\epsilon, \epsilon_1)G + 6\alpha^2 B^2(\epsilon, \epsilon_1),$$

where $(a)$ comes from $\lambda_{\max}(\Phi^T \bar{D}\Phi) \leq 1$ and $24\alpha^2 \leq 24\alpha\frac{(1-\gamma)\bar{w}}{96} = \frac{\alpha\nu}{4}$. Moving $\mathbb{E}\left\|V_{\bar{\theta}_t} - V_{\theta^*}\right\|_{\bar{D}}^2$ (on the right-hand side of $(a)$) to the left hand side of the above inequality yields:

$$\frac{\alpha\nu}{4}\mathbb{E}\left\|V_{\bar{\theta}_t} - V_{\theta^*}\right\|_{\bar{D}}^2 \leq (1 - \frac{\alpha\nu}{2})\mathbb{E}\left\|\bar{\theta}_t - \theta^*\right\|^2 - \mathbb{E}\left\|\bar{\theta}_{t+1} - \theta^*\right\|^2 + \frac{2\alpha^2\sigma^2}{NK}$$

$$+ 108(\frac{6\alpha^4}{K\alpha_g^2} + \frac{\alpha^3}{K\alpha_g^2\nu})(\sigma^2 + 3KB^2(\epsilon, \epsilon_1) + 2KG^2) + 2\alpha B(\epsilon, \epsilon_1)G + 6\alpha^2 B^2(\epsilon, \epsilon_1).$$

Dividing by $\alpha$ on both sides of the inequality above and changing $\nu$ into $\nu_1$, we have:

$$\nu_1\mathbb{E}\left\|V_{\bar{\theta}_t} - V_{\theta^*}\right\|_{\bar{D}}^2$$

$$\leq (\frac{1}{\alpha} - \nu_1)\mathbb{E}\left\|\bar{\theta}_t - \theta^*\right\|^2 - \frac{1}{\alpha}\mathbb{E}\left\|\bar{\theta}_{t+1} - \theta^*\right\|^2 + \frac{2\alpha\sigma^2}{NK}$$

$$+ 108(\frac{6\alpha^3}{K\alpha_g^2} + \frac{4\alpha^2}{K\alpha_g^2\nu_1})(\sigma^2 + 3KB^2(\epsilon, \epsilon_1) + 2KG^2) + 2B(\epsilon, \epsilon_1)G + 6\alpha B^2(\epsilon, \epsilon_1)$$

$$\leq (\frac{1}{\alpha} - \nu_1)\mathbb{E}\left\|\bar{\theta}_t - \theta^*\right\|^2 - \frac{1}{\alpha}\mathbb{E}\left\|\bar{\theta}_{t+1} - \theta^*\right\|^2 + \underbrace{\frac{2\alpha\sigma^2}{NK}}_{O(\alpha^1)}$$

$$+ \underbrace{\frac{1080\alpha^2}{K\alpha_g^2\nu_1}(\sigma^2 + 3KB^2(\epsilon, \epsilon_1) + 2KG^2)}_{O(\alpha^2)} + \underbrace{2B(\epsilon, \epsilon_1)G + 6\alpha B^2(\epsilon, \epsilon_1)}_{\text{heterogeneity term}},$$

where we used the fact that $\alpha \leq 1$ in the last inequality. $\qquad\square$

With these lemmas, we are now ready to prove Theorem 4, which we restate for clarity.

### G.2  Proof of Theorem 4

Given a fixed local step-size $\alpha_l = \frac{1}{2}\frac{(1-\gamma)\bar{\omega}}{48K}$, decreasing effective step-sizes $\alpha_t = \frac{8}{\nu(a+t+1)} = \frac{8}{(1-\gamma)\bar{\omega}(a+t+1)}$, decreasing global step-sizes $\alpha_g^{(t)} = \frac{\alpha_t}{K\alpha_l}$, and weights $w_t = (a+t)$, we have that

$$\mathbb{E}\left\|V_{\tilde{\theta}_T} - V_{\theta_i^*}\right\|_{\bar{D}}^2 \leq \tilde{\mathcal{O}}\left(\frac{G^2}{K^2T^2} + \frac{\sigma^2}{\nu^4KT^2} + \frac{\sigma^2}{\nu^2NKT} + \frac{B(\epsilon, \epsilon_1)G}{\nu} + \Gamma^2(\epsilon, \epsilon_1)\right) \qquad (29)$$

holds for any agent $i \in [N]$.

*Proof.* We take the effective step-size $\alpha_t = \frac{8}{\nu(a+t+1)} = \frac{2}{\nu_1(a+t+1)}$ for $a > 0$. In addition, we define weights $w_t = (a+t)$ and define

$$\tilde{\theta}_T = \frac{1}{W}\sum_{t=1}^{T} w_t\bar{\theta}_t,$$

where $W = \sum_{t=1}^{T} w_t \geq \frac{1}{2}T(a+T)$. By convexity of positive definite quadratic forms ($\lambda_{\min}(\Phi^T\bar{D}\Phi) \geq \bar{\omega} > 0$), we have that

$$\nu_1\mathbb{E}\left\|V_{\tilde{\theta}_T} - V_{\theta^*}\right\|_{\bar{D}}^2$$

$$\leq \frac{\nu_1}{W} \sum_{t=1}^{T} (a+t) \mathbb{E} \left\| V_{\bar{\theta}_t} - V_{\theta^*} \right\|_{\bar{D}}^2$$

$$\overset{(28)}{\leq} \frac{\nu_1 (a+1)(a+2) G^2}{2W} + \frac{1}{W} \sum_{t=1}^{T} \left[ \frac{2(a+t)\alpha_t}{NK} \sigma^2 \right]$$

$$+ \frac{1}{W} \sum_{t=1}^{T} \left[ \frac{1080(a+t)\alpha_t^2}{K\alpha_g^2 \nu_1} (\sigma^2 + 3KB^2(\epsilon, \epsilon_1) + 2KG^2) \right]$$

$$+ \frac{1}{W} \sum_{t=1}^{T} (a+t) \left[ 2B(\epsilon, \epsilon_1) G + 6\alpha_t B^2(\epsilon, \epsilon_1) \right]$$

$$\leq \frac{\nu_1 (a+1)(a+2) G^2}{2W} + \frac{2\sigma^2}{NKW} \sum_{t=1}^{T} (a+t)\alpha_t$$

$$+ \frac{1080(\sigma^2 + 3KB^2(\epsilon, \epsilon_1) + 2KG^2)}{K\alpha_g^2 \nu_1 W} \sum_{t=1}^{T} (a+t)\alpha_t^2 + 2B(\epsilon, \epsilon_1) G + \frac{6B^2(\epsilon, \epsilon_1)}{W} \sum_{t=1}^{T} (a+t)\alpha_t$$

$$\leq \frac{\nu_1 (a+1)(a+2) G^2}{2W} + \frac{4\sigma^2}{\nu_1 NKW} \cdot T$$

$$+ \frac{4320(\sigma^2 + 3KB^2(\epsilon, \epsilon_1) + 2KG^2)}{K\alpha_g^2 \nu_1^3 W} \cdot (1 + \log(a+T)) + 2B(\epsilon, \epsilon_1) G + \frac{12B^2(\epsilon, \epsilon_1)}{\nu_1 W} \cdot T,$$

where we used $\left\| V_{\bar{\theta}_0} - V_{\theta^*} \right\|_{\bar{D}}^2 \leq G^2$. Dividing by $\nu_1$ on both sides, changing $\nu_1$ into $\nu$, and using $W \geq \frac{T(a+T)}{2}$, we have:

$$\mathbb{E} \left\| V_{\bar{\theta}_T} - V_{\theta^*} \right\|_{\bar{D}}^2 \leq \tilde{\mathcal{O}} \left( \frac{G^2}{K^2 T^2} + \frac{\sigma^2}{\nu^4 K T^2} + \frac{\sigma^2}{\nu^2 NKT} + \frac{B(\epsilon, \epsilon_1) G}{\nu} \right).$$

We finish the proof by using the following inequality: $\mathbb{E} \left\| V_{\bar{\theta}_T} - V_{\theta_i^*} \right\|_{\bar{D}}^2 \leq 2\mathbb{E} \left\| V_{\bar{\theta}_T} - V_{\theta^*} \right\|_{\bar{D}}^2 + 2\mathbb{E} \left\| V_{\theta_i^*} - V_{\theta^*} \right\|_{\bar{D}}^2$, in tandem with the third point in Theorem 1. $\qquad \square$

## H   Heterogeneity bias: Proof of Theorem 3

In this section, we prove Theorem 3.

**Proof of Theorem 3.**   As $\theta_1^*$ and $\theta_2^*$ are the TD(0) fixed points of agents 1 and 2, respectively, we have $\theta_1^* = \bar{A}_1^{-1}\bar{b}_1$ and $\theta_2^* = \bar{A}_2^{-1}\bar{b}_2$. The output of mean-path `FedTD(0)` with $k = 1$ and $\alpha = \alpha_g\alpha_l$ satisfies:

$$\bar{\theta}_{t+1} = \bar{\theta}_t + \alpha(-\hat{A}\bar{\theta}_t + \hat{b})$$

$$\Longrightarrow \bar{\theta}_{t+1} - \theta_1^* = \bar{\theta}_t - \theta_1^* + \alpha(-\hat{A}(\bar{\theta}_t - \theta_1^* + \theta_1^*) + \hat{b})$$

$$\Longrightarrow e_{1,t+1} = (I - \alpha\hat{A})e_{1,t} - \alpha\hat{A}\theta_1^* + \alpha\hat{b}$$

$$\Longrightarrow e_{1,t+1} = (I - \alpha\hat{A})e_{1,t} - \alpha\left(\frac{\bar{A}_1 + \bar{A}_2}{2}\right)\bar{A}_1^{-1}\bar{b}_1 + \alpha\frac{\bar{b}_1 + \bar{b}_2}{2}$$

$$\Longrightarrow e_{1,t+1} = (I - \alpha\hat{A})e_{1,t} - \alpha\frac{\bar{A}_2\bar{A}_1^{-1}\bar{b}_1}{2} + \alpha\frac{\bar{b}_2}{2}$$

$$\Longrightarrow e_{1,t+1} = (I - \alpha\hat{A})e_{1,t} - \frac{\alpha\bar{A}_2}{2}\left(\bar{A}_1^{-1}\bar{b}_1 - \bar{A}_2^{-1}\bar{b}_2\right)$$

$$\Longrightarrow e_{1,t+1} = \underbrace{(I - \alpha\hat{A})}_{\tilde{\mathcal{A}}}e_{1,t} + \underbrace{\frac{\alpha\bar{A}_2}{2}\left(\theta_2^* - \theta_1^*\right)}_{\tilde{\mathcal{Y}}}. \tag{30}$$

Let us now note that $e_{1,t+1} = \tilde{\mathcal{A}}e_{1,t} + \tilde{\mathcal{Y}}$ can be viewed as a discrete-time linear time-invariant (LTI) system where $\alpha$ is chosen s.t. $\tilde{\mathcal{A}}$ is Schur stable, i.e., $|\lambda_{\max}(\tilde{\mathcal{A}})| < 1$. At the $t$-th iteration, we have:

$$e_{1,t} = \tilde{\mathcal{A}}^t e_{1,0} + \sum_{k=0}^{t-1}\tilde{\mathcal{A}}^k\tilde{\mathcal{Y}}.$$

As $t \to \infty$, the small gain theorem tells us that because $\rho(\tilde{\mathcal{A}}) < 1$ (where $\rho(\cdot)$ denotes the spectral radius), $\sum_{k=0}^{t-1}\tilde{\mathcal{A}}^k$ exists and is given by $(I - \tilde{\mathcal{A}})^{-1}$. We can then conclude that

$$\lim_{t\to\infty} e_{1,t} = (I - \tilde{\mathcal{A}})^{-1}\tilde{\mathcal{Y}}$$

$$= \left(\alpha\hat{A}\right)^{-1}\frac{\alpha\bar{A}_2}{2}\left(\theta_1^* - \theta_2^*\right)$$

$$= \frac{1}{2}\hat{A}^{-1}\bar{A}_2\left(\theta_1^* - \theta_2^*\right). \tag{31}$$

The limiting expression for $e_{2,t}$ follows the same analysis.

# I Proof of the Markovian setting

We now turn our attention to proving the main result of the paper, namely, Theorem 2.

## I.1 Outline

As mentioned in the main body, one of the main obstacles to overcome in the analysis is that in general, $\mathbb{E}[(1/N)\sum_{i=1}^{N}\left(g_i(\theta_{t,k}^{(i)}, O_{t,k}^{(i)}) - \bar{g}_i(\theta_{t,k}^{(i)})\right)] \neq 0$. In order to show that a linear speedup is achievable, we first decompose the random TD direction of each agent $i$ as $g_i(\theta_{t,k}^{(i)}) = b_i(O_{t,k}^{(i)}) - A_i(O_{t,k}^{(i)})\theta_{t,k}^{(i)}$ in subsection I.2.1 and show that the variances of $(1/NK)\sum_{i=1}^{N}\sum_{k=0}^{K-1}A_i(O_{t,k}^{(i)})$ and $(1/NK)\sum_{i=1}^{N}\sum_{k=0}^{K-1}b_i(O_{t,k}^{(i)})$ get scaled down by $NK$ in subsection I.2.2. To decouple the randomness between the parameter $\theta_{t,k}^{(i)}$ and the observations $O_{t,k}^{(i)}$ using the method called *information theoretic control of coupling* in Bhandari et al. (2018), we need to bound $\mathbb{E}\left[\left\|\bar{\theta}_t - \bar{\theta}_{t-\tau}\right\|^2\right]$ in subsection I.2.3. As the analysis in the i.i.d. setting and traditional FL, we characterize the drift term, per-iteration error decrease, and parameter selection in subsections I.2.4, I.2.5 and I.2.6, respectively. Finally, we prove Theorem 2 in subsection I.3.

**Additional Notation:** Under Assumption 3, for each MDP $i$, there exists some $m_i \geq 1$ and some $\rho_i \in (0,1)$, such that for all $t \geq 0$ and $0 \leq k \leq K-1$, it holds that

$$d_{TV}\left(\mathbb{P}\left(s_{t,k}^{(i)} = \cdot \mid s_{0,0}^{(i)} = s\right), \pi^{(i)}\right) \leq m_i \rho_i^{tK+k}, \forall s \in \mathcal{S}.$$

Furthermore, we define $\rho = \max_{i\in[N]}\{\rho_i\}$, $m = \max_{i\in[N]}\{m_i\}$.

## I.2 Auxiliary lemmas for Theorem 2

### I.2.1 Decomposition Form

The first step in our proof of Theorem 2 is to rewrite agent $i$'s update direction of FedTD(0) as:

$$g_i(\theta_{t,k}^{(i)}) = -A_i(O_{t,k}^{(i)})\theta_{t,k}^{(i)} + b_i(O_{t,k}^{(i)})$$

where $A_i(O_{t,k}^{(i)}) = \phi(s_{t,k}^{(i)})(\phi^\top(s_{t,k}^{(i)}) - \gamma\phi^\top(s_{t,k+1}^{(i)}))$ and $b_i(O_{t,k}^{(i)}) = r(s_{t,k}^{(i)})\phi(s_{t,k}^{(i)})$. Note that the steady-state value of $\mathbb{E}[b_i(O_{t,k}^{(i)})]$ is not equal to 0. For convenience, we apply appropriate centering to rewrite $g_i$ as:

$$g_i(\theta_{t,k}^{(i)}) = -A_i(O_{t,k}^{(i)})(\theta_{t,k}^{(i)} - \theta_i^*) + \underbrace{b_i(O_{t,k}^{(i)}) - A_i(O_{t,k}^{(i)})\theta_i^*}_{Z_i(O_{t,k}^{(i)})}. \tag{32}$$

Define $Z_i(O_{t,k}^{(i)}) \triangleq b_i(O_{t,k}^{(i)}) - A_i(O_{t,k}^{(i)})\theta_i^*$. As $\bar{g}_i(\theta) \triangleq \mathbb{E}_{O_{t,k}^{(i)}\sim\pi^{(i)}}[g_i(\theta)]$, we have:

$$\bar{g}_i(\theta_{t,k}^{(i)}) = -\bar{A}_i(\theta_{t,k}^{(i)} - \theta_i^*). \tag{33}$$

where $\bar{A}_i = \Phi^\top D^{(i)}(\Phi - \gamma P^{(i)}\Phi)$. Note that $\mathbb{E}_{O_{t,k}^{(i)}\sim\pi^{(i)}}\left[Z_i(O_{t,k}^{(i)})\right]$ equals to 0. Taking into account the definitions above, we establish the following lemmas:

**Lemma 11.** *(Uniform norm bound) There exist some constants $c_1, c_2, c_3 \geq 0$ such that $\left\|A_i\left(O_{t,k}^{(i)}\right)\right\| \leq c_1 := 1 + \gamma$, $\left\|\bar{A}_i\right\| \leq c_2 := 1 + \gamma$ and $\left\|Z_i\left(O_{t,k}^{(i)}\right)\right\| \leq c_3 := R_{\max} + c_1 H$ holds for all $i \in [N]$.*

*Proof.* Based on the definition and the fact that $\|\phi(s)\| \leq 1$, we have

$$\left\| A_i\left(O_{t,k}^{(i)}\right) \right\| = \left\| \phi(s_{t,k}^{(i)})(\phi^\top(s_{t,k}^{(i)}) - \gamma\phi^\top(s_{t,k+1}^{(i)})) \right\| \leq \left\| \phi(s_{t,k}^{(i)}) \right\| \left\| \phi^\top(s_{t,k}^{(i)}) - \gamma\phi^\top(s_{t,k+1}^{(i)}) \right\| \leq 1 + \gamma.$$

Similarly, making use of the fact that $r(s) \leq R_{\max}$ for any $s \in \mathcal{S}$, we apply the same reasoning to conclude that

$$\left\| \bar{A}_i \right\| \leq 1 + \gamma \ \& \ \left\| Z_i\left(O_{t,k}^{(i)}\right) \right\| \leq R_{\max} + c_1 H$$

. $\qquad\qquad\square$

**Lemma 12.** *There exist some constants $L_1, L_2 \geq 0$ such that*

$$\left\| \bar{A}_i - \mathbb{E}\left[ A_i\left(O_{t_2,k_2}^{(i)}\right) \mid \mathcal{F}_{k_1}^{t_1} \right] \right\| \leq L_1 \rho^{(t_2-t_1)K+k_2-k_1},$$

$$\left\| \bar{A}_i - \mathbb{E}_{t_1}\left[ A_i\left(O_{t_2,k_2}^{(i)}\right) \right] \right\| \leq L_1 \rho^{(t_2-t_1)K+k_2},$$

$$\left\| \mathbb{E}\left[ Z_i\left(O_{t_2,k_2}^{(i)}\right) \mid \mathcal{F}_{k_1}^{t_1} \right] \right\| \leq L_2 \rho^{(t_2-t_1)K+k_2-k_1},$$

$$\left\| \mathbb{E}_{t_1}\left[ Z_i\left(O_{t_2,k_2}^{(i)}\right) \right] \right\| \leq L_2 \rho^{(t_2-t_1)K+k_2}$$

*hold for any $i \in [N]$, $0 \leq k_1, k_2 \leq K-1$ and $t_2 \geq t_1 \geq 0$.*

*Proof.* We have:

$$\left\| \mathbb{E}\left[ Z_i\left(O_{t_2,k_2}^{(i)}\right) \mid \mathcal{F}_{k_1}^{t_1} \right] \right\| = \left\| \mathbb{E}\left[ Z_i\left(O_{t_2,k_2}^{(i)}\right) \mid \mathcal{F}_{k_1}^{t_1} \right] - \mathbb{E}_{O_{t_2,k_2}^{(i)} \sim \pi^{(i)}}\left[ Z_i\left(O_{t_2,k_2}^{(i)}\right) \mid \mathcal{F}_{k_1}^{t_1} \right] \right\|$$

$$= \left\| \sum_{s_{t_2,k_2}^{(i)}, s_{t_2+1,k_2+1}^{(i)}} \left( \pi^{(i)}(s_{t_2,k_2}^{(i)})P(s_{t_2+1,k_2+1}^{(i)} \mid s_{t_2,k_2}^{(i)}) \right. \right.$$

$$\left. \left. -P(s_{t_2,k_2}^{(i)} = \cdot \mid s_{t_1,k_1}^{(i)})P(s_{t_2+1,k_2+1}^{(i)} \mid s_{t_2,k_2}^{(i)}) \right) Z_i(O_{t_2,k_2}^{(i)}) \right\|$$

$$\leq \sum_{s_{t_2,k_2}^{(i)}} \left| \pi^{(i)}(s_{t_2,k_2}^{(i)}) - P(s_{t_2,k_2}^{(i)} = \cdot \mid s_{t_1,k_1}^{(i)}) \right| \left\| Z_i(O_{t_2,k_2}^{(i)}) \right\|$$

$$\overset{(a)}{\leq} \sum_{s_{t_2,k_2}^{(i)}} \left| \pi^{(i)}(s_{t_2,k_2}^{(i)}) - P(s_{t_2,k_2}^{(i)} = \cdot \mid s_{t_1,k_1}^{(i)}) \right| (R_{\max} + c_1 H)$$

$$= 2(R_{\max} + c_1 H)d_{TV}\left( \mathbb{P}\left( s_{t_2,k_2}^{(i)} = \cdot \mid s_{t_1,k_1}^{(i)} = s \right), \pi^{(i)} \right)$$

$$\leq 2(R_{\max} + c_1 H)m_i \rho_i^{(t_2-t_1)K+k_2-k_1}$$

where $(a)$ is due to Lemma 11 and the last step follows from Assumption 3. We finish the proof by choosing $L_2 \triangleq \max_{i \in [N]}\{2(R_{\max} + c_1 H)m_i\} = 2c_3 m$. And we follow the same analysis to bound:

$$\left\| \bar{A}_i - \mathbb{E}\left[ A_i\left(O_{t_2,k_2}^{(i)}\right) \mid \mathcal{F}_{k_1}^{t_1} \right] \right\| = \left\| \|\mathbb{E}\left[ A_i\left(O_{t_2,k_2}^{(i)}\right) \mid \mathcal{F}_{k_1}^{t_1} \right] - \mathbb{E}_{O_{t_2,k_2}^{(i)} \sim \pi^{(i)}}\left[ A_i\left(O_{t_2,k_2}^{(i)}\right) \mid \mathcal{F}_{k_1}^{t_1} \right] \right\|$$

$$= \left\| \sum_{s_{t_2,k_2}^{(i)}, s_{t_2+1,k_2+1}^{(i)}} \left( \pi^{(i)}(s_{t_2,k_2}^{(i)})P(s_{t_2+1,k_2+1}^{(i)} \mid s_{t_2,k_2}^{(i)}) \right. \right.$$

$$\left. \left. -P(s_{t_2,k_2}^{(i)} = \cdot \mid s_{t_1,k_1}^{(i)})P(s_{t_2+1,k_2+1}^{(i)} \mid s_{t_2,k_2}^{(i)}) \right) A_i(O_{t_2,k_2}^{(i)}) \right\|$$

$$\leq \sum_{s_{t_2,k_2}^{(i)}} \left| \pi^{(i)}(s_{t_2,k_2}^{(i)}) - P(s_{t_2,k_2}^{(i)} = \cdot \mid s_{t_1,k_1}^{(i)}) \right| \left\| A_i(O_{t_2,k_2}^{(i)}) \right\|$$

$$\overset{(b)}{\leq} 2c_1 d_{TV}\left(\mathbb{P}\left(s_{t_2,k_2}^{(i)} = \cdot \mid s_{t_1,k_1}^{(i)} = s\right), \pi^{(i)}\right)$$

$$\leq 2c_1 m_i \rho_i^{(t_2-t_1)K + k_2 - k_1}$$

We finish the proof by choosing $L_1 \triangleq \max_{i \in [N]}\{2c_1 m_i\} = 2c_1 m$. We employ the same reasoning to prove the remaining three inequalities. $\qquad \square$

### I.2.2 Variance Reduction

We are now ready to present the variance reduction Lemma in the Markov setting. The following Lemma establishes an analog of the variance reduction Lemma 7 in the i.i.d. setting. Based on the assumption that trajectories are independent across agents, it is easy to understand that the variance of $(1/NK)\sum_{i=1}^N \sum_{k=0}^{K-1} A_i(O_{t,k}^{(i)})$ and $(1/NK)\sum_{i=1}^N \sum_{k=0}^{K-1} b_i(O_{t,k}^{(i)})$ can be scaled by the number of agents $N$. However, it is not obvious that the variances can be scaled by $K$ (the number of local iterations), since the observations of each agent $O_{t,k_1}^{(i)}$ and $O_{t,k_2}^{(i)}$ are correlated at different local steps $k_1, k_2$. Due to the geometric mixing property of the Markov chain, the correlation between $O_{t,k_1}^{(i)}$ and $O_{t,k_2}^{(i)}$ will geometrically decay after the mixing time. Based on this fact, we show that the variances of $(1/NK)\sum_{i=1}^N \sum_{k=0}^{K-1} A_i(O_{t,k}^{(i)})$ and $(1/NK)\sum_{i=1}^N \sum_{k=0}^{K-1} b_i(O_{t,k}^{(i)})$ get scaled down by $NK$ with an additional additive, higher order term dependent on the mixing time $\tau$, which is formally stated as follows:

**Lemma 13.** *(Variance reduction in the Markovian setting) For any $0 < \tau < t$, there exists $d_1, d_2 > 0$ such that:*

$$\mathbb{E}_{t-\tau}\left[\left\|\frac{1}{NK}\sum_{i=1}^N \sum_{k=0}^{K-1}\left[A_i(O_{t,k}^{(i)}) - \bar{A}_i\right]\right\|\right] \leq \frac{d_1}{\sqrt{NK}} + 2L_1 \rho^{\tau K}, \tag{34}$$

$$\mathbb{E}_{t-\tau}\left[\left\|\frac{1}{NK}\sum_{i=1}^N \sum_{k=0}^{K-1}\left[A_i(O_{t,k}^{(i)}) - \bar{A}_i\right]\right\|^2\right] \leq \frac{d_1^2}{NK} + 4L_1^2 \rho^{2\tau K}, \quad and \tag{35}$$

$$\mathbb{E}_{t-\tau}\left[\left\|\frac{1}{NK}\sum_{i=1}^N \sum_{k=0}^{K-1} Z_i(O_{t,k}^{(i)})\right\|\right] \leq \frac{d_2}{\sqrt{NK}} + 2L_2 \rho^{\tau K}, \tag{36}$$

$$\mathbb{E}_{t-\tau}\left[\left\|\frac{1}{NK}\sum_{i=1}^N \sum_{k=0}^{K-1} Z_i(O_{t,k}^{(i)})\right\|^2\right] \leq \frac{d_2^2}{NK} + 4L_2^2 \rho^{2\tau K}, \tag{37}$$

*where $d_1 \triangleq \sqrt{(c_1 + c_2)^2 + \frac{2(c_1+c_2)L_1\rho}{1-\rho}}$ and $d_2 \triangleq \sqrt{c_3^2 + \frac{2c_3 L_2 \rho}{1-\rho}}$.*

*Proof.*

$$\mathbb{E}_{t-\tau}\left[\left\|\frac{1}{NK}\sum_{i=1}^N \sum_{k=0}^{K-1} Z_i(O_{t,k}^{(i)})\right\|\right]$$

$$= \mathbb{E}_{t-\tau}\left[\sqrt{\left(\frac{1}{NK}\sum_{i=1}^N \sum_{k=0}^{K-1} Z_i(O_{t,k}^{(i)})\right)^\top \left(\frac{1}{NK}\sum_{i=1}^N \sum_{k=0}^{K-1} Z_i(O_{t,k}^{(i)})\right)}\right]$$

$$\leq \sqrt{\mathbb{E}_{t-\tau}\left[\left(\frac{1}{NK}\sum_{i=1}^N \sum_{k=0}^{K-1} Z_i(O_{t,k}^{(i)})\right)^\top \left(\frac{1}{NK}\sum_{i=1}^N \sum_{k=0}^{K-1} Z_i(O_{t,k}^{(i)})\right)\right]}$$

(concavity of square root and Jensen's inequality)

$$
= \left\{ \mathbb{E}_{t-\tau} \left[ \frac{1}{N^2 K^2} \sum_{i=1}^{N} \sum_{k=0}^{K-1} Z_i(O_{t,k}^{(i)})^\top Z_i(O_{t,k}^{(i)}) + \underbrace{\frac{2}{N^2 K^2} \sum_{i=1}^{N} \sum_{k<l} Z_i(O_{t,k}^{(i)})^\top Z_i(O_{t,l}^{(i)})}_{T_1} \right. \right.
$$

$$
\left. \left. + \underbrace{\frac{2}{N^2 K^2} \sum_{i<j} \sum_{k=0}^{K-1} Z_i(O_{t,k}^{(i)})^\top Z_j(O_{t,k}^{(j)})}_{T_2} + \underbrace{\frac{2}{N^2 K^2} \sum_{i<j} \sum_{k<l} Z_i(O_{t,k}^{(i)})^\top Z_j(O_{t,l}^{(j)})}_{T_3} \right] \right\}^{\frac{1}{2}} \tag{38}
$$

where $T_1$ can be further bounded by:

$$
\mathbb{E}_{t-\tau}[T_1] = \mathbb{E}_{t-\tau} \left[ \frac{2}{N^2 K^2} \sum_{i=1}^{N} \sum_{k<l} Z_i(O_{t,k}^{(i)})^\top Z_i(O_{t,l}^{(i)}) \right]
$$

$$
= \mathbb{E}_{t-\tau} \left[ \frac{2}{N^2 K^2} \sum_{i=1}^{N} \sum_{k<l} Z_i(O_{t,k}^{(i)})^\top \mathbb{E} \left[ Z_i(O_{t,l}^{(i)}) \mid \mathcal{F}_k^t \right] \right]
$$

$$
\leq \mathbb{E}_{t-\tau} \left[ \frac{2}{N^2 K^2} \sum_{i=1}^{N} \sum_{k<l} \left\| Z_i(O_{t,k}^{(i)}) \right\| \left\| \mathbb{E} \left[ Z_i(O_{t,l}^{(i)}) \mid \mathcal{F}_k^t \right] \right\| \right]
$$

(Cauchy–Schwarz inequality)

$$
\leq \mathbb{E}_{t-\tau} \left[ \frac{2}{N^2 K^2} \sum_{i=1}^{N} \sum_{k<l} c_3 L_2 \rho^{(l-k)} \right] \qquad (\text{ Lemma 11 and 12})
$$

$$
\leq \mathbb{E}_{t-\tau} \left[ \frac{2}{N^2 K^2} \sum_{i=1}^{N} \sum_{k=0}^{K-1} \sum_{m=1}^{\infty} c_3 L_2 \rho^m \right]
$$

$$
= \frac{2 c_3 L_2 N K}{N^2 K^2} \frac{\rho}{1 - \rho} = \frac{2 c_3 L_2 \rho}{N K (1 - \rho)}.
$$

And $T_2$ can be bounded by:

$$
\mathbb{E}_{t-\tau}[T_2] = \frac{2}{N^2 K^2} \sum_{i<j} \sum_{k=0}^{K-1} \mathbb{E}_{t-\tau} \left[ Z_i(O_{t,k}^{(i)}) \right]^\top \mathbb{E}_{t-\tau} \left[ Z_j(O_{t,k}^{(j)}) \right] \quad (O_{t,k}^{(i)} \text{ and } O_{t,k}^{(j)} \text{ are independent})
$$

$$
\leq \frac{2}{N^2 K^2} \sum_{i<j} \sum_{k=0}^{K-1} L_2^2 \rho^{2\tau K + 2k} \quad (\text{Lemma 12})
$$

$$
\leq \frac{2}{K} L_2^2 \rho^{2\tau K}.
$$

Meanwhile, $T_3$ can be bounded by:

$$
\mathbb{E}_{t-\tau}[T_3] = \frac{2}{N^2 K^2} \sum_{i<j} \sum_{k<l} \mathbb{E}_{t-\tau} \left[ Z_i(O_{t,k}^{(i)}) \right]^\top \mathbb{E}_{t-\tau} \left[ Z_j(O_{t,l}^{(j)}) \right] \quad (O_{t,k}^{(i)} \text{ and } O_{t,l}^{(j)} \text{ are independent})
$$

$$
\leq \frac{2}{N^2 K^2} \sum_{i<j} \sum_{k<l} L_2^2 \rho^{2\tau K + k + l} \quad (\text{Lemma 12})
$$

$$
\leq 2 L_2^2 \rho^{2\tau K}
$$

Substituting the upper bound of $T_1$, $T_2$ and $T_3$ into Eq (38), we have:

$$
\mathbb{E}_{t-\tau} \left[ \left\| \frac{1}{NK} \sum_{i=1}^{N} \sum_{k=0}^{K-1} Z_i(O_{t,k}^{(i)}) \right\| \right]
$$

$$\leq \sqrt{\frac{1}{N^2K^2} \sum_{i=1}^{N} \sum_{k=0}^{K-1} \mathbb{E}_{t-\tau} \left[ Z_i(O_{t,k}^{(i)})^\top Z_i(O_{t,k}^{(i)}) \right] + \frac{2c_3 L_2 \rho}{NK(1-\rho)} + \frac{2}{K} L_2^2 \rho^{2\tau K} + 2L_2^2 \rho^{2\tau K}}$$

$$\overset{(a)}{\leq} \sqrt{\frac{NK}{N^2K^2} c_3^2 + \frac{2c_3 L \rho}{NK(1-\rho)} + \frac{2}{K} L_2^2 \rho^{2\tau K} + 2L_2^2 \rho^{2\tau K}}$$

$$\leq \sqrt{\frac{1}{NK} \left( c_3^2 + \frac{2c_3 L_2 \rho}{1-\rho} \right) + 4L_2^2 \rho^{2\tau K}} \quad (K \geq 1)$$

$$\leq \sqrt{\frac{1}{NK} \left( c_3^2 + \frac{2c_3 L_2 \rho}{1-\rho} \right)} + \sqrt{4L_2^2 \rho^{2\tau K}} = \sqrt{\frac{1}{NK} \left( c_3^2 + \frac{2c_3 L_2 \rho}{1-\rho} \right)} + 2L_2 \rho^{\tau K}.$$

where $(a)$ used the fact that $\left\| Z_i \left( O_{t,k}^{(i)} \right) \right\| \leq c_3$ mentioned in Lemma 11. The proof of other inequalities follows the same reasoning. $\qquad \square$

### I.2.3 Bounding $\mathbb{E} \left[ \left\| \bar{\theta}_t - \bar{\theta}_{t-\tau} \right\|^2 \right]$

**Lemma 14.** *(Bounding $\|\theta_t - \theta_{t-\tau}\|^2$) Consider $\tau = \lceil \frac{\tau^{\mathrm{mix}}(\alpha_T^2)}{K} \rceil$ and choose the effective step-size*

$$\alpha \leq \min \left\{ \frac{1}{30c_4(\tau+1)}, \frac{1}{96c_4^2\tau}, 1 \right\}$$

*where $c_4 = 3c_1$. For any $t \geq 2\tau$, we have the following bound:*

$$\mathbb{E}_{t-2\tau} \left[ \left\| \bar{\theta}_t - \bar{\theta}_{t-\tau} \right\|^2 \right] \leq 8\alpha^2 \tau^2 c_4^2 \mathbb{E}_{t-2\tau} \left[ \left\| \bar{\theta}_t - \theta^* \right\|^2 \right] + 14\alpha^2 \tau^2 \frac{d_2^2}{NK} + \frac{52 L_2^2 \alpha^4 \tau}{1-\rho^2}$$

$$+ 4\alpha^2 c_4^2 \tau \sum_{s=0}^{\tau} E_{t-2\tau}[\Delta_{t-s}] + 3200\alpha^2 c_4^2 c_1^2 \tau^3 \Gamma^2(\epsilon, \epsilon_1) + 4\alpha^2 c_1^2 \tau^2 \Gamma^2(\epsilon, \epsilon_1). \qquad (39)$$

*Proof.* For any $l \geq 2\tau$, we have

$$\left\| \bar{\theta}_{l+1} - \bar{\theta}_l \right\|^2$$

$$= \left\| \Pi_{2,\mathcal{H}} \left( \bar{\theta}_l + \frac{\alpha}{NK} \sum_{i=1}^{N} \sum_{k=0}^{K-1} g_i(\theta_{l,k}^{(i)}) \right) - \bar{\theta}_l \right\|^2$$

$$\leq \left\| \bar{\theta}_l + \frac{\alpha}{NK} \sum_{i=1}^{N} \sum_{k=0}^{K-1} g_i(\theta_{l,k}^{(i)}) - \bar{\theta}_l \right\|^2$$

$$= \alpha^2 \left\| \frac{1}{NK} \sum_{i=1}^{N} \sum_{k=0}^{K-1} \left[ -A_i(O_{l,k}^{(i)}) \left( \theta_{l,k}^{(i)} - \theta_i^* \right) + Z_i(O_{l,k}^{(i)}) \right] \right\|^2$$

$$\leq 2\alpha^2 \left\| \frac{1}{NK} \sum_{i=1}^{N} \sum_{k=0}^{K-1} \left[ -A_i(O_{l,k}^{(i)}) \left( \theta_{l,k}^{(i)} - \theta^* \right) + Z_i(O_{l,k}^{(i)}) \right] \right\|^2$$

$$+ 2\alpha^2 \left\| \frac{1}{NK} \sum_{i=1}^{N} \sum_{k=0}^{K-1} \left[ -A_i(O_{l,k}^{(i)}) \left( \theta^* - \theta_i^* \right) \right] \right\|^2$$

$$\overset{(a)}{\leq} 2\alpha^2 \left\| \frac{1}{NK} \sum_{i=1}^{N} \sum_{k=0}^{K-1} \left[ -A_i(O_{l,k}^{(i)}) \left( \theta_{l,k}^{(i)} - \theta^* \right) + Z_i(O_{l,k}^{(i)}) \right] \right\|^2 + 2\alpha^2 c_1^2 \Gamma^2(\epsilon, \epsilon_1)$$

$$= 6\alpha^2 \left\| \frac{1}{NK} \sum_{i=1}^{N} \sum_{k=0}^{K-1} A_i(O_{l,k}^{(i)})\left(\theta_{l,k}^{(i)} - \bar{\theta}_l\right) \right\|^2 + 6\alpha^2 \left\| \frac{1}{NK} \sum_{i=1}^{N} \sum_{k=0}^{K-1} A_i(O_{l,k}^{(i)})\left(\bar{\theta}_l - \theta^*\right) \right\|^2$$

$$+ 6\alpha^2 \left\| \frac{1}{NK} \sum_{i=1}^{N} \sum_{k=0}^{K-1} Z_i(O_{l,k}^{(i)}) \right\|^2 + 2\alpha^2 c_1^2 \Gamma^2(\epsilon, \epsilon_1)$$

$$\leq 6\alpha^2 \left( \frac{c_1}{NK} \sum_{i=1}^{N} \sum_{k=0}^{K-1} \left\| \theta_{l,k}^{(i)} - \bar{\theta}_l \right\| \right)^2$$

$$+ 6\alpha^2 c_1^2 \left\| \bar{\theta}_l - \theta^* \right\|^2 + 6\alpha^2 \left\| \frac{1}{NK} \sum_{i=1}^{N} \sum_{k=0}^{K-1} Z_i(O_{l,k}^{(i)}) \right\|^2 + 2\alpha^2 c_1^2 \Gamma^2(\epsilon, \epsilon_1), \tag{40}$$

where $(a)$ comes from the upper bound of fixed points distance in Theorem 1 and the fact that $\left\| A_i\left(O_{t,k}^{(i)}\right) \right\| \leq c_1$ in Lemma 11. Taking square root on both sides of the inequality above, we get:

$$\left\| \bar{\theta}_{l+1} - \bar{\theta}_l \right\|$$

$$\leq 3\sqrt{\alpha^2 \left( \frac{c_1}{NK} \sum_{i=1}^{N} \sum_{k=0}^{K-1} \left\| \theta_{l,k}^{(i)} - \bar{\theta}_l \right\| \right)^2} + 3\sqrt{\alpha^2 c_1^2 \left\| \bar{\theta}_l - \theta^* \right\|^2}$$

$$+ 3\sqrt{\alpha^2 \left\| \frac{1}{NK} \sum_{i=1}^{N} \sum_{k=0}^{K-1} Z_i(O_{l,k}^{(i)}) \right\|^2} + \sqrt{2\alpha^2 c_1^2 \Gamma^2(\epsilon, \epsilon_1)}$$

$$\leq \frac{3\alpha c_1}{NK} \sum_{i=1}^{N} \sum_{k=0}^{K-1} \left\| \theta_{l,k}^{(i)} - \bar{\theta}_l \right\| + 3\alpha c_1 \left\| \bar{\theta}_l - \theta^* \right\| + 3\alpha \left\| \frac{1}{NK} \sum_{i=1}^{N} \sum_{k=0}^{K-1} Z_i(O_{l,k}^{(i)}) \right\| + 2\alpha c_1 \Gamma(\epsilon, \epsilon_1). \tag{41}$$

By using the fact that $\left\| \bar{\theta}_{l+1} - \theta^* \right\| \leq \left\| \bar{\theta}_l - \theta^* \right\| + \left\| \bar{\theta}_{l+1} - \bar{\theta}_l \right\|$, we have:

$$\left\| \bar{\theta}_{l+1} - \theta^* \right\| \leq (1 + 3\alpha c_1)\left\| \bar{\theta}_l - \theta^* \right\| + \frac{3\alpha c_1}{NK} \sum_{i=1}^{N} \sum_{k=0}^{K-1} \left\| \theta_{l,k}^{(i)} - \bar{\theta}_l \right\|$$
$$+ 3\alpha \left\| \frac{1}{NK} \sum_{i=1}^{N} \sum_{k=0}^{K-1} Z_i(O_{l,k}^{(i)}) \right\| + 2\alpha c_1 \Gamma(\epsilon, \epsilon_1). \tag{42}$$

For simplicity, we define $c_4 \triangleq 3c_1$ and $\delta_l \triangleq \frac{1}{NK} \sum_{i=1}^{N} \sum_{k=0}^{K-1} \left\| \theta_{l,k}^{(i)} - \bar{\theta}_l \right\|$. Taking the square on both sides of Eq (42), we have:

$$\left\| \bar{\theta}_{l+1} - \theta^* \right\|^2$$

$$\leq (1 + \alpha c_4)^2 \left\| \bar{\theta}_l - \theta^* \right\|^2 + \alpha^2 c_4^2 \delta_l^2 + 9\alpha^2 \left\| \frac{1}{NK} \sum_{i=1}^{N} \sum_{k=0}^{K-1} Z_i(O_{l,k}^{(i)}) \right\|^2 + 4\alpha^2 c_1^2 \Gamma^2(\epsilon, \epsilon_1)$$

$$+ \underbrace{6\alpha(1 + \alpha c_4)\left\| \bar{\theta}_l - \theta^* \right\| \left\| \frac{1}{NK} \sum_{i=1}^{N} \sum_{k=0}^{K-1} Z_i(O_{l,k}^{(i)}) \right\|}_{H_1} + \underbrace{2\alpha c_4(1 + \alpha c_4)\left\| \bar{\theta}_l - \theta^* \right\| \delta_l}_{H_2}$$

$$+ \underbrace{6\alpha^2 c_4 \delta_l \left\| \frac{1}{NK} \sum_{i=1}^{N} \sum_{k=0}^{K-1} Z_i(O_{l,k}^{(i)}) \right\|}_{H_3} + \underbrace{4\alpha^2 c_1 c_4 \delta_l \Gamma(\epsilon, \epsilon_1)}_{H_4}$$

$$+ \underbrace{4\alpha c_1(1+\alpha c_4)\left\|\bar{\theta}_l - \theta^*\right\|\Gamma(\epsilon,\epsilon_1)}_{H_5} + \underbrace{12\alpha^2 c_1\left\|\frac{1}{NK}\sum_{i=1}^{N}\sum_{k=0}^{K-1}Z_i(O_{l,k}^{(i)})\right\|\Gamma(\epsilon,\epsilon_1)}_{H_6}. \tag{43}$$

We can further bound $H_1$ as:

$$
\begin{aligned}
H_1 &= 6\alpha(1+\alpha c_4)\left\|\bar{\theta}_l - \theta^*\right\|\left\|\frac{1}{NK}\sum_{i=1}^{N}\sum_{k=0}^{K-1}Z_i(O_{l,k}^{(i)})\right\| \\
&= 2\sqrt{3\alpha(1+\alpha c_4)}\left\|\bar{\theta}_l - \theta^*\right\| \cdot \sqrt{3\alpha(1+\alpha c_4)}\left\|\frac{1}{NK}\sum_{i=1}^{N}\sum_{k=0}^{K-1}Z_i(O_{l,k}^{(i)})\right\| \\
&\leq 3\alpha(1+\alpha c_4)\left\|\bar{\theta}_l - \theta^*\right\|^2 + 3\alpha(1+\alpha c_4)\left\|\frac{1}{NK}\sum_{i=1}^{N}\sum_{k=0}^{K-1}Z_i(O_{l,k}^{(i)})\right\|^2 \\
&\leq 6\alpha\left\|\bar{\theta}_l - \theta^*\right\|^2 + 6\alpha\left\|\frac{1}{NK}\sum_{i=1}^{N}\sum_{k=0}^{K-1}Z_i(O_{l,k}^{(i)})\right\|^2. \tag{44}
\end{aligned}
$$

where we use the fact $1 + \alpha c_4 \leq 2$ in the last inequality. Similary, we can bound $H_2$ as:

$$H_2 = 2\alpha c_4(1+\alpha c_4)\left\|\bar{\theta}_l - \theta^*\right\|\delta_l \leq 2\alpha\left\|\bar{\theta}_l - \theta^*\right\|^2 + 2\alpha c_4^2\delta_l^2. \tag{45}$$

And we bound $H_3$ as:

$$H_3 = 6\alpha^2 c_4\delta_l\left\|\frac{1}{NK}\sum_{i=1}^{N}\sum_{k=0}^{K-1}Z_i(O_{l,k}^{(i)})\right\| \leq 3\alpha^2\left\|\frac{1}{NK}\sum_{i=1}^{N}\sum_{k=0}^{K-1}Z_i(O_{l,k}^{(i)})\right\|^2 + 3\alpha^2 c_4^2\delta_l^2. \tag{46}$$

For $H_4, H_5, H_6$, we have:

$$H_4 = 4\alpha^2 c_1 c_4\delta_l\Gamma(\epsilon,\epsilon_1) \leq 2\alpha^2 c_4^2\delta_l^2 + 2\alpha^2 c_1^2\Gamma^2(\epsilon,\epsilon_1),$$

$$H_5 = 4\alpha c_1(1+\alpha c_4)\left\|\bar{\theta}_l - \theta^*\right\|\Gamma(\epsilon,\epsilon_1) \leq 4\alpha\left\|\bar{\theta}_l - \theta^*\right\|^2 + 4\alpha c_1^2\Gamma^2(\epsilon,\epsilon_1),$$

$$H_6 = 12\alpha^2 c_1\left\|\frac{1}{NK}\sum_{i=1}^{N}\sum_{k=0}^{K-1}Z_i(O_{l,k}^{(i)})\right\|\Gamma(\epsilon,\epsilon_1) \leq 6\alpha^2\left\|\frac{1}{NK}\sum_{i=1}^{N}\sum_{k=0}^{K-1}Z_i(O_{l,k}^{(i)})\right\|^2 + 6\alpha^2 c_1^2\Gamma^2(\epsilon,\epsilon_1),$$

Substituting the upper bound of $H_1, H_2, \ldots, H_6$ into Eq (43) and noting that $(1+\alpha c_4)^2 \leq 1 + 3\alpha c_4$ because $\alpha c_4 \leq 1$, we have:

$$
\begin{aligned}
&\left\|\bar{\theta}_{l+1} - \theta^*\right\|^2 \\
&\leq (1+\alpha(3c_4+12))\left\|\bar{\theta}_l - \theta^*\right\|^2 + (6\alpha^2 + 2\alpha)c_4^2\delta_l^2 \\
&\quad + (18\alpha^2 + 6\alpha)\left\|\frac{1}{NK}\sum_{i=1}^{N}\sum_{k=0}^{K-1}Z_i(O_{l,k}^{(i)})\right\|^2 + (12\alpha^2 + 4\alpha)c_1^2\Gamma^2(\epsilon,\epsilon_1) \\
&\leq (1+\alpha h_1)\left\|\bar{\theta}_l - \theta^*\right\|^2 + 8\alpha c_4^2\delta_l^2 + 24\alpha\left\|\frac{1}{NK}\sum_{i=1}^{N}\sum_{k=0}^{K-1}Z_i(O_{l,k}^{(i)})\right\|^2 + 16\alpha c_1^2\Gamma^2(\epsilon,\epsilon_1), \tag{47}
\end{aligned}
$$

where we denote $h_1 \triangleq 3c_4 + 12$ for simplicity. For any $t - \tau \leq l \leq t$, conditioning on $\mathcal{F}_{t-2\tau}$ on both sides of the above inequality, we have:

$$\mathbb{E}_{t-2\tau}\left\|\bar{\theta}_{l+1} - \theta^*\right\|^2$$

$$\leq (1 + \alpha h_1) \, \mathbb{E}_{t-2\tau} \left\| \bar{\theta}_l - \theta^* \right\|^2 + 24\alpha \mathbb{E}_{t-2\tau} \left\| \frac{1}{NK} \sum_{i=1}^{N} \sum_{k=0}^{K-1} Z_i(O_{l,k}^{(i)}) \right\|^2$$

$$+ 8\alpha c_4^2 \mathbb{E}_{t-2\tau} \left[ \delta_l^2 \right] + \alpha M_3(\epsilon, \epsilon_1)$$

$$\leq (1 + \alpha h_1) \, \mathbb{E}_{t-2\tau} \left\| \bar{\theta}_l - \theta^* \right\|^2 + 24\alpha \left[ \frac{d_2^2}{NK} + 4L_2^2 \rho^{2(l-t+2\tau)K} \right] \quad \text{(Lemma 13)}$$

$$+ 8\alpha c_4^2 \mathbb{E}_{t-2\tau} \left[ \delta_l^2 \right] + \alpha M_3(\epsilon, \epsilon_1)$$

$$\overset{(a)}{\leq} (1 + \alpha h_1) \, \mathbb{E}_{t-2\tau} \left\| \bar{\theta}_l - \theta^* \right\|^2 + 24\alpha \left[ \frac{d_2^2}{NK} + 4L_2^2 \alpha^2 \rho^{2(l-t+\tau)K} \right]$$

$$+ 8\alpha c_4^2 \mathbb{E}_{t-2\tau} \left[ \delta_l^2 \right] + \alpha M_3(\epsilon, \epsilon_1)$$

$$\leq (1 + \alpha h_1) \, \mathbb{E}_{t-2\tau} \left\| \bar{\theta}_l - \theta^* \right\|^2 + \alpha c_t(l) + 8\alpha c_4^2 \mathbb{E}_{t-2\tau} \left[ \delta_l^2 \right] + \alpha M_3(\epsilon, \epsilon_1), \tag{48}$$

where we denote $M_3(\epsilon, \epsilon_1) \triangleq 16c_1^2 \Gamma^2(\epsilon, \epsilon_1)$ and $c_t(l) = 24 \left[ \frac{d_2^2}{NK} + 4L_2^2 \alpha^2 \rho^{2(l-t+\tau)K} \right]$ for simplicity. Inequality $(a)$ is due to $\rho^{2\tau K} \leq \alpha_T^4 \leq \alpha_t^2$. In the following steps, we try to map $\mathbb{E}_{t-2\tau} \left\| \bar{\theta}_{l+1} - \theta^* \right\|^2$ to $\mathbb{E}_{t-2\tau} \left\| \bar{\theta}_{t-\tau} - \theta^* \right\|^2$ for any $t - \tau \leq l \leq t$. By applying Eq (48) recursively, we have:

$$\mathbb{E}_{t-2\tau} \left\| \bar{\theta}_{l+1} - \theta^* \right\|^2$$

$$\leq (1 + \alpha h_1)^{l+1-t+\tau} \mathbb{E}_{t-2\tau} \left\| \bar{\theta}_{t-\tau} - \theta^* \right\|^2 + \alpha \sum_{k=t-\tau}^{l} (1 + \alpha h_1)^{l-k} \left( c_t(k) + M_3(\epsilon, \epsilon_1) \right)$$

$$+ 8\alpha c_4^2 \mathbb{E}_{t-2\tau} \left[ \sum_{k=t-\tau}^{l} (1 + \alpha h_1)^{l-k} \delta_k^2 \right]$$

$$\overset{(b)}{\leq} (1 + \alpha h_1)^{\tau+1} \mathbb{E}_{t-2\tau} \left\| \bar{\theta}_{t-\tau} - \theta^* \right\|^2 + \alpha \underbrace{\sum_{k=t-\tau}^{t} (1 + \alpha h_1)^{l-k} \left( c_t(k) + M_3(\epsilon, \epsilon_1) \right)}_{H_7}$$

$$+ 8\alpha c_4^2 \mathbb{E}_{t-2\tau} \underbrace{\left[ \sum_{k=t-\tau}^{t} (1 + \alpha h_1)^{l-k} \delta_k^2 \right]}_{H_8} \tag{49}$$

where $(b)$ is due to $l \leq t$. For $H_7$, we have:

$$H_7 \leq \sum_{k=t-\tau}^{t} (1 + \alpha h_1)^{t-k} \left( c_t(k) + M_3(\epsilon, \epsilon_1) \right) \quad (l \leq t)$$

$$= \sum_{k'=0}^{\tau} (1 + \alpha h_1)^{\tau-k'} \left( c_t(k' + t - \tau) + M_3(\epsilon, \epsilon_1) \right)$$

$$\text{( changing index } k \text{ into } k' \text{ with } k' = k + \tau - t)$$

$$\leq 24 \sum_{k'=0}^{\tau} (1 + \alpha h_1)^{\tau-k'} \left[ \frac{d_2^2}{NK} + 4L_2^2 \alpha^2 \rho^{2k'K} + M_3(\epsilon, \epsilon_1) \right]$$

$$\text{(Substituting the definition of } c_t(k') \text{ inside)}$$

$$= 24 \left[ \left( \frac{d_2^2}{NK} + M_3(\epsilon, \epsilon_1) \right) \frac{(1 + \alpha h_1)^{\tau+1} - 1}{\alpha h_1} + 4L_2^2 \alpha^2 (1 + \alpha h_1)^{\tau} \sum_{k'=0}^{\tau} \left( \frac{\rho^{2K}}{1 + \alpha h_1} \right)^{k'} \right]$$

$$\leq 24 \left[ \left( \frac{d_2^2}{NK} + M_3(\epsilon, \epsilon_1) \right) \frac{(1 + \alpha h_1)^{\tau+1} - 1}{\alpha h_1} + 4L_2^2 \alpha^2 (1 + \alpha h_1)^{\tau} \sum_{k'=0}^{\tau} \rho^{2k'K} \right] (1 + \alpha h_1 \geq 1)$$

$$\leq 24 \left[ \left( \frac{d_2^2}{NK} + M_3(\epsilon, \epsilon_1) \right) \frac{(1 + \alpha h_1)^{\tau+1} - 1}{\alpha h_1} + 4 L_2^2 \alpha^2 (1 + \alpha h_1)^{\tau} \frac{1}{1 - \rho^2} \right].$$

Here we follow the analysis in (Khodadadian et al., 2022). Notice that for $x \leq \frac{\log 2}{\tau}$, we have $(1 + x)^{\tau+1} \leq 1 + 2x(\tau + 1)$. If $\alpha \leq \frac{1}{4 h_1 \tau} \leq \frac{\log 2}{h_1 \tau}$ and $\alpha \leq \frac{1}{2 h_1 (\tau+1)}$, we have $(1 + \alpha h_1)^{\tau+1} \leq 1 + 2\alpha h_1(\tau + 1) \leq 2$ and $(1 + \alpha h_1)^{\tau} \leq 1 + 2\alpha h_1 \tau \leq 1 + 1/2 \leq 2$. Hence, we have

$$H_7 \leq 24 \left[ \left( \frac{d_2^2}{NK} + M_3(\epsilon, \epsilon_1) \right) 2(\tau + 1) + \frac{8 L_2^2 \alpha^2}{1 - \rho^2} \right].$$

We apply the similar analysis to bound $H_8$ as:

$$H_8 = \sum_{k=0}^{\tau} (1 + \alpha h_1)^{\tau-k} \delta_{t-\tau+k}^2 \leq \sum_{k=0}^{\tau} (1 + \alpha h_1)^{\tau} \delta_{t-\tau+k}^2 \leq \sum_{k=0}^{\tau} (1 + 2\alpha h_1 \tau) \delta_{t-\tau+k}^2 \leq 2 \sum_{k=0}^{\tau} \delta_{t-k}^2.$$

Substituting the upper bound of $H_7$ and $H_8$ into Eq (49), we have:

$$\mathbb{E}_{t-2\tau} \left\| \bar{\theta}_{l+1} - \theta^* \right\|^2 \leq 2 \mathbb{E}_{t-2\tau} \left\| \bar{\theta}_{t-\tau} - \theta^* \right\|^2 + 24\alpha \left[ \left( \frac{d_2^2}{NK} + M_3(\epsilon, \epsilon_1) \right) 2(\tau + 1) + \frac{8 L_2^2 \alpha^2}{1 - \rho^2} \right]$$
$$+ 16\alpha c_4^2 \sum_{k=0}^{\tau} \mathbb{E}_{t-2\tau}[\delta_{t-k}^2].$$

Then it is straightforward to bound $\mathbb{E}_{t-2\tau} \left\| \bar{\theta}_l - \theta^* \right\|^2$ as:

$$\mathbb{E}_{t-2\tau} \left\| \bar{\theta}_l - \theta^* \right\|^2 \leq 2 \mathbb{E}_{t-2\tau} \left\| \bar{\theta}_{t-\tau} - \theta^* \right\|^2 + 24\alpha \left[ \left( \frac{d_2^2}{NK} + M_3(\epsilon, \epsilon_1) \right) 4\tau + \frac{8 L_2^2 \alpha^2}{1 - \rho^2} \right]$$
$$+ 16\alpha c_4^2 \sum_{k=0}^{\tau} \mathbb{E}_{t-2\tau}[\delta_{t-k}^2]. \tag{50}$$

Furthermore, based on the triangle inequality, we have:

$$\left\| \bar{\theta}_t - \bar{\theta}_{t-\tau} \right\|^2$$
$$\leq \left( \sum_{s=t-\tau}^{t-1} \left\| \bar{\theta}_{s+1} - \bar{\theta}_s \right\| \right)^2 \leq \tau \sum_{s=t-\tau}^{t-1} \left\| \bar{\theta}_{s+1} - \bar{\theta}_s \right\|^2$$
$$\leq \tau \sum_{s=t-\tau}^{t-1} \left[ \alpha^2 c_4^2 \left\| \bar{\theta}_s - \theta^* \right\|^2 + \alpha^2 c_4^2 \delta_s^2 + 6\alpha^2 \left\| \frac{1}{NK} \sum_{i=1}^{N} \sum_{k=0}^{K-1} Z_i(O_{s,k}^{(i)}) \right\|^2 + 2\alpha^2 c_1^2 \Gamma^2(\epsilon, \epsilon_1) \right]$$

where the last inequality is due to Eq (40) with $c_4 = 3 c_1$. If we take the expectation on both sides, we have:

$$\mathbb{E}_{t-2\tau} \left\| \bar{\theta}_t - \bar{\theta}_{t-\tau} \right\|^2$$
$$\leq \tau \sum_{s=t-\tau}^{t-1} \left[ \alpha^2 c_4^2 \mathbb{E}_{t-2\tau} \left\| \bar{\theta}_s - \theta^* \right\|^2 + \alpha^2 c_4^2 \delta_s^2 \right.$$
$$+ 6\alpha^2 \mathbb{E}_{t-2\tau} \left\| \frac{1}{NK} \sum_{i=1}^{N} \sum_{k=0}^{K-1} Z_i(O_{s,k}^{(i)}) \right\|^2 + 2\alpha^2 c_1^2 \Gamma^2(\epsilon, \epsilon_1) \right]$$
$$\leq \tau \alpha^2 c_4^2 \sum_{s=t-\tau}^{t-1} \left[ 2 \mathbb{E}_{t-2\tau} \left\| \bar{\theta}_{t-\tau} - \theta^* \right\|^2 \right]$$

$$+ 24\alpha \left[ \left( \frac{d_2^2}{NK} + M_3(\epsilon, \epsilon_1) \right) 4\tau + \frac{8L_2^2\alpha^2}{1 - \rho^2} \right] + 16\alpha c_4^2 \sum_{k=0}^{\tau} \mathbb{E}_{t-2\tau}[\delta_{t-k}^2] \right] \quad (\text{Eq } (50))$$

$$+ 6\alpha^2\tau \sum_{s=t-\tau}^{t-1} \left( \frac{d_2^2}{NK} + 4L_2^2\rho^{2(s-t+2\tau)K} \right)$$

$$+ \alpha^2 c_4^2 \tau \sum_{s=t-\tau}^{t-1} \mathbb{E}_{t-2\tau}[\delta_s^2] + 2\alpha^2 c_1^2 \tau^2 \Gamma^2(\epsilon, \epsilon_1) \quad (\text{Lemma } 13)$$

$$\overset{(a)}{\leq} \tau^2\alpha^2 c_4^2 \left[ 2\mathbb{E}_{t-2\tau} \left\| \bar{\theta}_{t-\tau} - \theta^* \right\|^2 + 96 \left( \frac{d_2^2}{NK}\alpha\tau + \frac{2L_2^2\alpha^3}{1 - \rho^2} \right) \right] + 6\alpha^2\tau \left[ \frac{d_2^2}{NK}\tau + \frac{4L_2^2\alpha^2}{1 - \rho^{2K}} \right]$$

$$+ \alpha^2 c_4^2 \tau (1 + 16\alpha\tau c_4^2) \sum_{s=0}^{\tau} \mathbb{E}_{t-2\tau}[\delta_{t-s}^2] + 96\alpha^2 c_4^2 \tau^3 M_3(\epsilon, \epsilon_1) + 2\alpha^2 c_1^2 \tau^2 \Gamma^2(\epsilon, \epsilon_1)$$

$$\overset{(b)}{\leq} 2\tau^2\alpha^2 c_4^2 \mathbb{E}_{t-2\tau} \left\| \bar{\theta}_{t-\tau} - \theta^* \right\|^2 + \frac{d_2^2}{NK}\alpha^2\tau^2 (96\alpha\tau c_4^2 + 6) + \frac{12L_2^2\alpha^4\tau}{1 - \rho^2} (16\alpha c_4^2\tau + 2)$$

$$+ \alpha^2 c_4^2 \tau (1 + 16\alpha\tau c_4^2) \sum_{s=0}^{\tau} E_{t-2\tau}[\Delta_{t-s}] + 96\alpha^2 c_4^2 \tau^3 M_3(\epsilon, \epsilon_1) + 2\alpha^2 c_1^2 \tau^2 \Gamma^2(\epsilon, \epsilon_1) \tag{51}$$

where we used the fact that $\rho^{2\tau K} \leq \alpha^2$ for $(a)$ and $(b)$, and that $\delta_t^2 \leq \Delta_t$ (via Jensens' inequality) for all $t \geq 0$ in the last inequality. Let us choose $\alpha$ such that $96\alpha\tau c_4^2 + 6 \leq 7$, $16\alpha c_4^2\tau + 2 \leq \frac{13}{6}$ and $1 + 16\alpha\tau c_4^2 \leq 2$, this holds when

$$\alpha \leq \min \left\{ \frac{1}{96\tau c_4^2}, \frac{1}{96 c_4^2\tau}, \frac{1}{16\tau c_4^2}, 1 \right\}.$$

Based on the fact that $\|\bar{\theta}_{t-\tau} - \theta^*\|^2 \leq 2\|\bar{\theta}_t - \bar{\theta}_{t-\tau}\|^2 + 2\|\bar{\theta}_t - \theta^*\|^2$ and the requirement on $\alpha$, we have

$$2\alpha^2\tau^2 c_4^2 \mathbb{E}_{t-2\tau}\|\bar{\theta}_{t-\tau} - \theta^*\|^2 \leq 4\alpha^2\tau^2 c_4^2 \mathbb{E}_{t-2\tau}\|\bar{\theta}_t - \bar{\theta}_{t-\tau}\|^2 + 4\alpha^2\tau^2 c_4^2 \mathbb{E}_{t-2\tau}\|\bar{\theta}_t - \theta^*\|^2$$

$$\leq 0.5\mathbb{E}_{t-2\tau}\|\bar{\theta}_t - \bar{\theta}_{t-\tau}\|^2 + 4\alpha^2\tau^2 c_4^2 \mathbb{E}_{t-2\tau}\|\bar{\theta}_t - \theta^*\|^2 \quad (4\alpha^2\tau^2 c_4^2 \leq 0.5)$$

$$\overset{(a)}{\leq} \tau^2\alpha^2 c_4^2 \mathbb{E}_{t-2\tau} \left\| \bar{\theta}_{t-\tau} - \theta^* \right\|^2 + \frac{7d_2^2}{2NK}\alpha^2\tau^2 + \frac{13L_2^2\alpha^4\tau}{(1 - \rho^2)}$$

$$+ \alpha^2 c_4^2 \tau \sum_{s=0}^{\tau} E_{t-2\tau}[\Delta_{t-s}] + 48\alpha^2 c_4^2 \tau^3 M_3(\epsilon, \epsilon_1) + \alpha^2 c_1^2 \tau^2 \Gamma^2(\epsilon, \epsilon_1)$$

$$+ 4\alpha^2\tau^2 c_4^2 \mathbb{E}_{t-2\tau}\|\bar{\theta}_t - \theta^*\|^2 \tag{52}$$

where (a) is due to Eq (51) and the choice of $\alpha$. Putting the term $\tau^2\alpha^2 c_4^2 \mathbb{E}_{t-2\tau} \left\| \bar{\theta}_{t-\tau} - \theta^* \right\|^2$ together by rearranging the terms, we have:

$$\alpha^2\tau^2 c_4^2 \mathbb{E}_{t-2\tau}\|\bar{\theta}_{t-\tau} - \theta^*\|^2 \leq \frac{7d_2^2}{2NK}\alpha^2\tau^2 + \frac{13L_2^2\alpha^4\tau}{(1 - \rho^2)}$$

$$+ \alpha^2 c_4^2 \tau \sum_{s=0}^{\tau} E_{t-2\tau}[\Delta_{t-s}] + 48\alpha^2 c_4^2 \tau^3 M_3(\epsilon, \epsilon_1) + \alpha^2 c_1^2 \tau^2 \Gamma^2(\epsilon, \epsilon_1)$$

$$+ 4\alpha^2\tau^2 c_4^2 \mathbb{E}_{t-2\tau}\|\bar{\theta}_t - \theta^*\|^2 \tag{53}$$

The proof is completed by substituting this inequality into Eq (51) and the definition of $M_3(\epsilon, \epsilon_1)$. Note that we require the effective step-size

$$\alpha \leq \min \left\{ \frac{1}{4h_1\tau}, \frac{1}{2h_1(\tau + 1)}, \frac{1}{96 c_4^2\tau}, 1 \right\}$$

in this proof, which holds when $\alpha \leq \min \left\{ \frac{1}{30 c_4(\tau + 1)}, \frac{1}{96 c_4^2\tau}, 1 \right\}$ since $c_4 = 3c_1 \geq 1$.

$\square$

### I.2.4 Drift Term Analysis.

Now we bound the drift term as follows:

**Lemma 15.** *(Bounded Client Drift) If $\alpha_l \leq \frac{1}{2\sqrt{2}c_1(K-1)}$, the drift term satisfies*

$$\mathbb{E}[\Delta_t] = \frac{1}{NK}\sum_{i=1}^{N}\sum_{k=0}^{K-1}\mathbb{E}\left\|\theta_{t,k}^{(i)} - \bar{\theta}_t\right\|^2 \leq \frac{4\alpha^2}{K\alpha_g^2}\left[c_3^2 + \frac{2c_3L_2\rho}{1-\rho} + 8c_1^2(K-1)H^2\right]. \tag{54}$$

*Proof.*

$$\frac{1}{NK}\sum_{i=1}^{N}\sum_{k=0}^{K-1}\mathbb{E}\left\|\theta_{t,k}^{(i)} - \bar{\theta}_t\right\|^2 = \frac{1}{NK}\sum_{i=1}^{N}\sum_{k=0}^{K-1}\mathbb{E}\left\|\bar{\theta}_t + \alpha_l\sum_{s=0}^{k-1}g_i(\theta_{t,s}^{(i)}) - \bar{\theta}_t\right\|^2$$

$$= \alpha_l^2\frac{1}{NK}\sum_{i=1}^{N}\sum_{k=0}^{K-1}\mathbb{E}\left\|\sum_{s=0}^{k-1}-A_i(O_{t,s}^{(i)})\left(\theta_{t,s}^{(i)} - \theta_i^*\right) + Z_i(O_{t,s}^{(i)})\right\|^2$$

$$\leq 2\alpha_l^2\frac{1}{NK}\sum_{i=1}^{N}\sum_{k=0}^{K-1}\mathbb{E}\left\|\sum_{s=0}^{k-1}-A_i(O_{t,s}^{(i)})\left(\theta_{t,s}^{(i)} - \theta_i^*\right)\right\|^2 + 2\alpha_l^2\frac{1}{NK}\sum_{i=1}^{N}\sum_{k=0}^{K-1}\mathbb{E}\left\|\sum_{s=0}^{k-1}Z_i(O_{t,s}^{(i)})\right\|^2$$

$$\leq 2\alpha_l^2\frac{1}{NK}\sum_{i=1}^{N}\sum_{k=0}^{K-1}k\sum_{s=0}^{k-1}\mathbb{E}\left\|A_i(O_{t,s}^{(i)})\left(\theta_{t,s}^{(i)} - \theta_i^*\right)\right\|^2 + 2\alpha_l^2\frac{1}{NK}\sum_{i=1}^{N}\sum_{k=0}^{K-1}\sum_{s=0}^{k-1}\mathbb{E}\left\|Z_i(O_{t,s}^{(i)})\right\|^2$$

$$+ 2\alpha_l^2\frac{1}{NK}\sum_{i=1}^{N}\sum_{k=0}^{K-1}\sum_{\substack{s,s'=0\\s\neq s'}}^{k-1}\mathbb{E}\left\langle Z_i(O_{t,s}^{(i)}), Z_i(O_{t,s'}^{(i)})\right\rangle$$

$$\leq 2\alpha_l^2\frac{1}{NK}\sum_{i=1}^{N}\sum_{k=0}^{K-1}kc_1^2\sum_{s=0}^{k-1}\mathbb{E}\left\|\theta_{t,s}^{(i)} - \theta_i^*\right\|^2 + 2\alpha_l^2\frac{1}{NK}\sum_{i=1}^{N}\sum_{k=0}^{K-1}kc_3^2 \quad \text{(Lemma 11)}$$

$$+ 2\alpha_l^2\frac{1}{NK}\sum_{i=1}^{N}\sum_{k=0}^{K-1}\sum_{\substack{s,s'=0\\s\neq s'}}^{k-1}\mathbb{E}\left[\mathbb{E}\left\langle Z_i(O_{t,s}^{(i)}), Z_i(O_{t,s'}^{(i)})\right\rangle \mid \mathcal{F}_s^t\right]$$

$$\leq 2\alpha_l^2\frac{1}{NK}\sum_{i=1}^{N}\sum_{k=0}^{K-1}kc_1^2\sum_{s=0}^{k-1}\mathbb{E}\left\|\theta_{t,s}^{(i)} - \theta_i^*\right\|^2 + 2\alpha_l^2\frac{1}{NK}\sum_{i=1}^{N}\sum_{k=0}^{K-1}kc_3^2$$

$$+ 2\alpha_l^2\frac{1}{NK}\sum_{i=1}^{N}\sum_{k=0}^{K-1}\sum_{\substack{s,s'=0\\s\neq s'}}^{k-1}\mathbb{E}\left[\left\langle Z_i(O_{t,s}^{(i)}), \mathbb{E}\left[Z_i(O_{t,s'}^{(i)}) \mid \mathcal{F}_s^t\right]\right\rangle\right]$$

$$\leq 2\alpha_l^2\frac{1}{NK}\sum_{i=1}^{N}\sum_{k=0}^{K-1}kc_1^2\sum_{s=0}^{k-1}\mathbb{E}\left\|\theta_{t,s}^{(i)} - \theta_i^*\right\|^2 + 2\alpha_l^2\frac{1}{NK}\sum_{i=1}^{N}\sum_{k=0}^{K-1}kc_3^2$$

$$+ 4\alpha_l^2\frac{1}{NK}\sum_{i=1}^{N}\sum_{k=0}^{K-1}\sum_{\substack{s,s'=0\\s<s'}}^{k-1}\mathbb{E}\left[\left\|Z_i(O_{t,s}^{(i)})\right\|\left\|\mathbb{E}\left[Z_i(O_{t,s'}^{(i)}) \mid \mathcal{F}_s^t\right]\right\|\right]$$

$$\leq 4\alpha_l^2\frac{1}{NK}\sum_{i=1}^{N}\sum_{k=0}^{K-1}kc_1^2\sum_{s=0}^{k-1}\mathbb{E}\left\|\theta_{t,s}^{(i)} - \bar{\theta}_t\right\|^2 + 4\alpha_l^2\frac{1}{NK}\sum_{i=1}^{N}\sum_{k=0}^{K-1}kc_1^2\sum_{s=0}^{k-1}\mathbb{E}\left\|\bar{\theta}_t - \theta_i^*\right\|^2 \quad \text{(Eq (11))}$$

$$+ 2\alpha_l^2 \frac{1}{NK} \sum_{i=1}^{N} \sum_{k=0}^{K-1} kc_3^2 + 4\alpha_l^2 \frac{1}{NK} \sum_{i=1}^{N} \sum_{k=0}^{K-1} \sum_{\substack{s,s'=0 \\ s<s'}}^{k-1} c_3 L_2 \rho^{s'-s} \quad \text{(Lemma 12)}$$

$$\leq 4\alpha_l^2 \frac{1}{NK} \sum_{i=1}^{N} \sum_{k=0}^{K-1} kc_1^2 \sum_{s=0}^{k-1} \mathbb{E}\left\|\theta_{t,s}^{(i)} - \bar{\theta}_t\right\|^2 + 4\alpha_l^2 \frac{1}{NK} \sum_{i=1}^{N} \sum_{k=0}^{K-1} 4kc_1^2(K-1)H^2$$

$$+ 2\alpha_l^2 \frac{1}{NK} \sum_{i=1}^{N} \sum_{k=0}^{K-1} kc_3^2 + 4\alpha_l^2 \frac{1}{NK} \sum_{i=1}^{N} \sum_{k=0}^{K-1} \underbrace{\sum_{\substack{s,s'=0 \\ s<s'}}^{k-1} c_3 L_2 \rho^{s'-s}}_{\mathcal{M}_1} \tag{55}$$

where we used the property that $\bar{\theta}_t, \theta_i^* \in \mathcal{H}$ in the last inequality, i.e., $\|\bar{\theta}_t\| \leq H^2$ and $\|\theta_i^*\| \leq H^2$. We now bound $\mathcal{M}_1$ as:

$$\sum_{\substack{s,s'=0 \\ s<s'}}^{k-1} c_3 L_2 \rho^{s'-s} = c_3 L_2 \sum_{s=0}^{k-1} \sum_{s'=s+1}^{k-1} \rho^{s'-s} = c_3 L_2 \sum_{s=0}^{k-1} \frac{\rho - \rho^{s-s'}}{1-\rho} \leq c_3 L_2 \frac{\rho k}{1-\rho} \tag{56}$$

Define $\mathcal{R}_K \triangleq \sum_{i=1}^{N} \sum_{k=0}^{K-1} \mathbb{E}\left\|\theta_{t,k}^{(i)} - \bar{\theta}_t\right\|^2$ and note that $\mathcal{R}_K$ is monotonically increasing in $K$. With this definition, if we plug in the upper bound of $\mathcal{M}_1$ into Eq (55), we have:

$$\mathcal{R}_K \leq 4\alpha_l^2 \sum_{i=1}^{N} \sum_{k=0}^{K-1} kc_1^2 \sum_{s=0}^{k-1} \mathbb{E}\left\|\theta_{t,s}^{(i)} - \bar{\theta}_t\right\|^2 + 4\alpha_l^2 \sum_{i=1}^{N} \sum_{k=0}^{K-1} 4kc_1^2(K-1)H^2$$

$$+ 2\alpha_l^2 \sum_{i=1}^{N} \sum_{k=0}^{K-1} kc_3^2 + 4\alpha_l^2 \sum_{i=1}^{N} \sum_{k=0}^{K-1} c_3 L_2 \frac{\rho k}{1-\rho}$$

$$\leq 2\alpha_l^2(K-1)NK \left[c_3^2 + \frac{2c_3 L_2 \rho}{1-\rho} + 8c_1^2(K-1)H^2\right]$$

$$+ 4\alpha_l^2 c_1^2(K-1) \sum_{k=1}^{K-1} \underbrace{\sum_{i=1}^{N} \sum_{s=0}^{k-1} \mathbb{E}\left\|\theta_{t,s}^{(i)} - \bar{\theta}_t\right\|^2}_{\mathcal{R}_k}$$

$$= 2\alpha_l^2(K-1)NK \left[c_3^2 + \frac{2c_3 L_2 \rho}{1-\rho} + 8c_1^2(K-1)H^2\right] + 4\alpha_l^2 c_1^2(K-1) \sum_{k=1}^{K-1} \mathcal{R}_k \tag{57}$$

By the monotonicity of $\mathcal{R}_k$, we have

$$\mathcal{R}_K \leq 2\alpha_l^2(K-1)NK \left[c_3^2 + \frac{2c_3 L_2 \rho}{1-\rho} + 8c_1^2(K-1)H^2\right] + 4\alpha_l^2 c_1^2(K-1)^2 \mathcal{R}_{K-1}$$

Let us choose $\alpha_l$ such that $4\alpha_l^2 c_1^2(K-1)^2 \leq \frac{1}{2}$, i.e., $\alpha_l \leq \frac{1}{2\sqrt{2}c_1(K-1)}$, the following recursion holds:

$$\mathcal{R}_K \leq \frac{1}{2}\mathcal{R}_{K-1} + 2\alpha_l^2(K-1)NK \left[c_3^2 + \frac{2c_3 L_2 \rho}{1-\rho} + 8c_1^2(K-1)H^2\right] \tag{58}$$

for all $k \in [K]$. Next, we unroll the recurrence, go back $K-1$ steps and use the fact that $\mathcal{R}_1 = 0$, we have:

$$\mathcal{R}_K \leq \left\{\sum_{l=1}^{\infty} \left(\frac{1}{2}\right)^l\right\} \left(2\alpha_l^2(K-1)NK \left[c_3^2 + \frac{2c_3 L_2 \rho}{1-\rho} + 8c_1^2(K-1)H^2\right]\right)$$

$$= 4\alpha_l^2(K-1)NK \left[c_3^2 + \frac{2c_3 L_2 \rho}{1-\rho} + 8c_1^2(K-1)H^2\right] \tag{59}$$

We finish the proof by dividing $NK$ on both sides and substituting $\alpha_l = \frac{\alpha}{K\alpha_g}$. $\qquad\square$

### I.2.5 Per Round Progress

**Lemma 16.** *(Per Round Progress). If the local step-size $\alpha_l \leq \frac{1}{2\sqrt{2}c_1(K-1)}$, and the effective step-size $\alpha = K\alpha_l\alpha_g$ satisfies:*

$$\alpha \leq \min\{\frac{\xi_1}{24(c_1+c_2)^2 + 24\xi_1^2 + 16}, 1, \frac{\xi_1(c_1+c_2)}{2L_1 + 8\tau^2 c_4^2}, \frac{1}{30c_4(\tau+1)}, \frac{1}{96c_4^2\tau}, \mathcal{X}\},^{[5]}$$

*where*

$$\mathcal{X} = \frac{2B(\epsilon,\epsilon_1)G + 3\xi_1(c_1+c_2)\Gamma^2(\epsilon,\epsilon_1)}{4B^2(\epsilon,\epsilon_1) + 24(c_1+c_2)^2\Gamma^2(\epsilon,\epsilon_1) + 2L_1\Gamma(\epsilon,\epsilon_1)G + 6400c_1^2c_4^2\tau^3\Gamma^2(\epsilon,\epsilon_1) + 8c_1^2\tau^2\Gamma^2(\epsilon,\epsilon_1)},$$

*and choose $\tau = \lceil \frac{\tau^{\mathrm{mix}}(\alpha_T^2)}{K} \rceil$, then we have,*

$$\mathbb{E}_{t-2\tau}\|\bar{\theta}_{t+1} - \theta^*\|^2$$

$$\leq \underbrace{(1 + 32\alpha\xi_1(c_1+c_2))\mathbb{E}_{t-2\tau}\left\|\bar{\theta}_t - \theta^*\right\|^2 + 2\alpha\mathbb{E}_{t-2\tau}\left\langle \bar{g}(\bar{\theta}_t), \bar{\theta}_t - \theta^*\right\rangle + 4\alpha^2\mathbb{E}_{t-2\tau}\left\|\bar{g}(\bar{\theta}_t)\right\|^2}_{\textit{Expected progress for the virtual MDP}}$$

$$+ \underbrace{\frac{9 + 28\tau^2}{NK}\alpha^2 d_2^2}_{\textit{Linear speedup}} + \underbrace{\alpha^3\left(36L_2^2 + \frac{108\tau}{1-\rho^2}L_2^2 + 4L_1G^2 + 2L_2G\right)}_{\textit{High order terms: } O(\alpha^3)}$$

$$+ \underbrace{\frac{4\alpha^3}{K\alpha_g^2}(\frac{14}{\xi_1} + 14\xi_1)(c_1+c_2)\left[c_3^2 + \frac{2c_3L_2\rho}{1-\rho} + 4c_1^2(K-1)H^2\right]}_{\textit{drift term}}$$

$$+ \underbrace{4\alpha B(\epsilon,\epsilon_1)G + 6\alpha\xi_1(c_1+c_2)\Gamma^2(\epsilon,\epsilon_1)}_{\textit{heterogeneity term}}. \tag{60}$$

*where $\xi_1$ is any universal positive constant.*

*Proof.* According to the updating rule and the fact that the projection operator is non-expansive, we have:

$$\mathbb{E}_{t-\tau}\left\|\bar{\theta}_{t+1} - \theta^*\right\|^2$$

$$= \mathbb{E}_{t-\tau}\left\|\Pi_{2,\mathcal{H}}\left(\bar{\theta}_t + \frac{\alpha}{NK}\sum_{i=1}^{N}\sum_{k=0}^{K-1}g_i(\theta_{t,k}^{(i)})\right) - \theta^*\right\|^2$$

$$\leq \mathbb{E}_{t-\tau}\left\|\bar{\theta}_t + \frac{\alpha}{NK}\sum_{i=1}^{N}\sum_{k=0}^{K-1}g_i(\theta_{t,k}^{(i)}) - \theta^*\right\|^2$$

$$= \mathbb{E}_{t-\tau}\left\|\bar{\theta}_t - \theta^*\right\|^2 + 2\mathbb{E}_{t-\tau}\left\langle\frac{\alpha}{NK}\sum_{i=1}^{N}\sum_{k=0}^{K-1}\bar{g}_i(\theta_{t,k}^{(i)}), \bar{\theta}_t - \theta^*\right\rangle$$

$$+ 2\mathbb{E}_{t-\tau}\left\langle\frac{\alpha}{NK}\sum_{i=1}^{N}\sum_{k=0}^{K-1}\left[g_i(\theta_{t,k}^{(i)}) - \bar{g}_i(\theta_{t,k}^{(i)})\right], \bar{\theta}_t - \theta^*\right\rangle$$

---

[5]This requirement is very easy to satisfy since the denominator in $\mathcal{X}$ is composed by the heterogeneity terms, which is quite small and thereby makes $\mathcal{X}$ large. Overall, the feasible set of the step-sizes is not empty.

$$+ \alpha^2 \mathbb{E}_{t-\tau} \Big\| \frac{1}{NK} \sum_{i=1}^{N} \sum_{k=0}^{K-1} g_i(\theta_{t,k}^{(i)}) \Big\|^2$$

$$\leq \mathbb{E}_{t-\tau} \underbrace{\left\{ \big\| \bar{\theta}_t - \theta^* \big\|^2 + 2 \Big\langle \frac{\alpha}{N} \sum_{i=1}^{N} \bar{g}_i(\bar{\theta}_t), \bar{\theta}_t - \theta^* \Big\rangle + 2 \Big\langle \frac{\alpha}{NK} \sum_{i=1}^{N} \sum_{k=0}^{K-1} \bar{g}_i(\theta_{t,k}^{(i)}) - \bar{g}_i(\bar{\theta}_t), \bar{\theta}_t - \theta^* \Big\rangle \right\}}_{\mathcal{B}_1}$$

$$+ 2\alpha \mathbb{E}_{t-\tau} \underbrace{\Big\langle \frac{1}{NK} \sum_{i=1}^{N} \sum_{k=0}^{K-1} \big[ g_i(\theta_{t,k}^{(i)}) - \bar{g}_i(\theta_{t,k}^{(i)}) \big], \bar{\theta}_t - \theta^* \Big\rangle}_{\mathcal{B}_2} + \alpha^2 \mathbb{E}_{t-\tau} \underbrace{\Big\| \frac{1}{NK} \sum_{i=1}^{N} \sum_{k=0}^{K-1} g_i(\theta_{t,k}^{(i)}) \Big\|^2}_{\mathcal{B}_3} \qquad (61)$$

We now begin to bound the gradient bias term $\mathcal{B}_2$ by decomposing this term into three terms:

$$\Big\langle \frac{1}{NK} \sum_{i=1}^{N} \sum_{k=0}^{K-1} \big[ g_i(\theta_{t,k}^{(i)}) - \bar{g}_i(\theta_{t,k}^{(i)}) \big], \bar{\theta}_t - \theta^* \Big\rangle$$

$$= \underbrace{\Big\langle \frac{1}{NK} \sum_{i=1}^{N} \sum_{k=0}^{K-1} \big[ g_i(\theta_{t,k}^{(i)}) - \bar{g}_i(\theta_{t,k}^{(i)}) \big], \bar{\theta}_t - \bar{\theta}_{t-\tau} \Big\rangle}_{\mathcal{B}_{21}}$$

$$+ \underbrace{\Big\langle \frac{1}{NK} \sum_{i=1}^{N} \sum_{k=0}^{K-1} \big[ g_i(\theta_{t,k}^{(i)}) - g_i(\theta_{t-\tau,k}^{(i)}) - \bar{g}_i(\theta_{t,k}^{(i)}) + \bar{g}_i(\theta_{t-\tau,k}^{(i)}) \big], \bar{\theta}_{t-\tau} - \theta^* \Big\rangle}_{\mathcal{B}_{22}}$$

$$+ \underbrace{\Big\langle \frac{1}{NK} \sum_{i=1}^{N} \sum_{k=0}^{K-1} \big[ g_i(\theta_{t-\tau,k}^{(i)}) - \bar{g}_i(\theta_{t-\tau,k}^{(i)}) \big], \bar{\theta}_{t-\tau} - \theta^* \Big\rangle}_{\mathcal{B}_{23}}. \qquad (62)$$

Next, we bound $\mathbb{E}_{t-\tau}[\mathcal{B}_{21}]$ as:

$$\mathbb{E}_{t-\tau} \Big\langle \frac{1}{NK} \sum_{i=1}^{N} \sum_{k=0}^{K-1} \big[ g_i(\theta_{t,k}^{(i)}) - \bar{g}_i(\theta_{t,k}^{(i)}) \big], \bar{\theta}_t - \bar{\theta}_{t-\tau} \Big\rangle$$

$$\leq \mathbb{E}_{t-\tau} \Big\| \frac{1}{NK} \sum_{i=1}^{N} \sum_{k=0}^{K-1} g_i(\theta_{t,k}^{(i)}) - \bar{g}_i(\theta_{t,k}^{(i)}) \Big\| \Big\| \bar{\theta}_t - \bar{\theta}_{t-\tau} \Big\|$$

$$\stackrel{(a)}{=} \mathbb{E}_{t-\tau} \left[ \Big\| \frac{1}{NK} \sum_{i=1}^{N} \sum_{k=0}^{K-1} (-A_i(O_{t,k}^{(i)}) + \bar{A}_i)(\theta_{t,k}^{(i)} - \theta_i^*) + Z_i(O_{t,k}^{(i)}) \Big\| \Big\| \bar{\theta}_t - \bar{\theta}_{t-\tau} \Big\| \right]$$

$$\leq \mathbb{E}_{t-\tau} \left[ \Big\| \frac{1}{NK} \sum_{i=1}^{N} \sum_{k=0}^{K-1} (A_i(O_{t,k}^{(i)}) - \bar{A}_i)(\theta_{t,k}^{(i)} - \theta_i^*) \Big\| \Big\| \bar{\theta}_t - \bar{\theta}_{t-\tau} \Big\| \right]$$

$$+ \mathbb{E}_{t-\tau} \left[ \Big\| \frac{1}{NK} \sum_{i=1}^{N} \sum_{k=0}^{K-1} Z_i(O_{t,k}^{(i)}) \Big\| \Big\| \bar{\theta}_t - \bar{\theta}_{t-\tau} \Big\| \right]$$

$$\leq \frac{1}{NK} \sum_{i=1}^{N} \sum_{k=0}^{K-1} \mathbb{E}_{t-\tau} \left[ \Big\| (A_i(O_{t,k}^{(i)}) - \bar{A}_i)(\theta_{t,k}^{(i)} - \theta_i^*) \Big\| \Big\| \bar{\theta}_t - \bar{\theta}_{t-\tau} \Big\| \right]$$

$$+ \frac{\alpha}{2} \mathbb{E}_{t-\tau} \left[ \Big\| \frac{1}{NK} \sum_{i=1}^{N} \sum_{k=0}^{K-1} Z_i(O_{t,k}^{(i)}) \Big\|^2 \right] + \frac{1}{2\alpha} \mathbb{E}_{t-\tau} \left[ \Big\| \bar{\theta}_t - \bar{\theta}_{t-\tau} \Big\|^2 \right]$$

$$\stackrel{(b)}{\leq} \frac{(c_1 + c_2)}{NK} \sum_{i=1}^{N} \sum_{k=0}^{K-1} \mathbb{E}_{t-\tau} \Big\| \theta_{t,k}^{(i)} - \theta_i^* \Big\| \Big\| \bar{\theta}_t - \bar{\theta}_{t-\tau} \Big\|$$

$$+ \frac{\alpha}{2} \mathbb{E}_{t-\tau} \left\| \frac{1}{NK} \sum_{i=1}^{N} \sum_{k=0}^{K-1} Z_i(O_{t,k}^{(i)}) \right\|^2 + \frac{1}{2\alpha} \mathbb{E}_{t-\tau} \left\| \bar{\theta}_t - \bar{\theta}_{t-\tau} \right\|^2$$

$$\leq \frac{\xi_1(c_1 + c_2)}{2NK} \sum_{i=1}^{N} \sum_{k=0}^{K-1} \mathbb{E}_{t-\tau} \left\| \theta_{t,k}^{(i)} - \theta_i^* \right\|^2 + \frac{(c_1 + c_2)}{2\xi_1 NK} \sum_{i=1}^{N} \sum_{k=0}^{K-1} \mathbb{E}_{t-\tau} \left\| \bar{\theta}_t - \bar{\theta}_{t-\tau} \right\|^2$$

(Young's inequality (12) with constant $\xi_1$)

$$+ \frac{\alpha}{2} \mathbb{E}_{t-\tau} \left\| \frac{1}{NK} \sum_{i=1}^{N} \sum_{k=0}^{K-1} Z_i(O_{t,k}^{(i)}) \right\|^2 + \frac{1}{2\alpha} \mathbb{E}_{t-\tau} \left\| \bar{\theta}_t - \bar{\theta}_{t-\tau} \right\|^2$$

$$\overset{(c)}{\leq} \frac{3\xi_1(c_1 + c_2)}{2NK} \sum_{i=1}^{N} \sum_{k=0}^{K-1} \mathbb{E}_{t-\tau} \left\| \theta_{t,k}^{(i)} - \bar{\theta}_t \right\|^2 + \frac{3\xi_1(c_1 + c_2)}{2NK} \sum_{i=1}^{N} \sum_{k=0}^{K-1} \mathbb{E}_{t-\tau} \left\| \bar{\theta}_t - \theta^* \right\|^2$$

$$+ \frac{3\xi_1(c_1 + c_2)}{2NK} \sum_{i=1}^{N} \sum_{k=0}^{K-1} \mathbb{E}_{t-\tau} \left\| \theta^* - \theta_i^* \right\|^2 + \frac{(c_1 + c_2)}{2\xi_1 NK} \sum_{i=1}^{N} \sum_{k=0}^{K-1} \mathbb{E}_{t-\tau} \left\| \bar{\theta}_t - \bar{\theta}_{t-\tau} \right\|^2$$

$$+ \frac{\alpha}{2} \mathbb{E}_{t-\tau} \left\| \frac{1}{NK} \sum_{i=1}^{N} \sum_{k=0}^{K-1} Z_i(O_{t,k}^{(i)}) \right\|^2 + \frac{1}{2\alpha} \mathbb{E}_{t-\tau} \left\| \bar{\theta}_t - \bar{\theta}_{t-\tau} \right\|^2$$

$$= \frac{3\xi_1(c_1 + c_2)}{2NK} \sum_{i=1}^{N} \sum_{k=0}^{K-1} \mathbb{E}_{t-\tau} \left\| \theta_{t,k}^{(i)} - \bar{\theta}_t \right\|^2 + \frac{3\xi_1(c_1 + c_2)}{2} \mathbb{E}_{t-\tau} \left\| \bar{\theta}_t - \theta^* \right\|^2$$

$$+ \frac{3\xi_1(c_1 + c_2)}{2} \Gamma^2(\epsilon, \epsilon_1) + \frac{(c_1 + c_2)}{2\xi_1} \mathbb{E}_{t-\tau} \left\| \bar{\theta}_t - \bar{\theta}_{t-\tau} \right\|^2$$

$$+ \frac{\alpha}{2} \mathbb{E}_{t-\tau} \left\| \frac{1}{NK} \sum_{i=1}^{N} \sum_{k=0}^{K-1} Z_i(O_{t,k}^{(i)}) \right\|^2 + \frac{1}{2\alpha} \mathbb{E}_{t-\tau} \left\| \bar{\theta}_t - \bar{\theta}_{t-\tau} \right\|^2$$

$$\overset{(d)}{\leq} \frac{3\xi_1(c_1 + c_2)}{2NK} \sum_{i=1}^{N} \sum_{k=0}^{K-1} \mathbb{E}_{t-\tau} \left\| \theta_{t,k}^{(i)} - \bar{\theta}_t \right\|^2 + \frac{3\xi_1(c_1 + c_2)}{2} \mathbb{E}_{t-\tau} \left\| \bar{\theta}_t - \theta^* \right\|^2$$

$$+ \frac{3\xi_1(c_1 + c_2)}{2} \Gamma^2(\epsilon, \epsilon_1) + \frac{c_1 + c_2}{2\xi_1} \mathbb{E}_{t-\tau} \left\| \bar{\theta}_t - \bar{\theta}_{t-\tau} \right\|^2$$

$$+ \frac{\alpha}{2} \left[ \frac{d_2^2}{NK} + 4L_2^2 \rho^{2\tau K} \right] + \frac{1}{2\alpha} \mathbb{E}_{t-\tau} \left\| \bar{\theta}_t - \bar{\theta}_{t-\tau} \right\|^2$$

$$= \frac{3\xi_1(c_1 + c_2)}{2NK} \sum_{i=1}^{N} \sum_{k=0}^{K-1} \mathbb{E}_{t-\tau} \left\| \theta_{t,k}^{(i)} - \bar{\theta}_t \right\|^2 + \frac{3\xi_1(c_1 + c_2)}{2} \mathbb{E}_{t-\tau} \left\| \bar{\theta}_t - \theta^* \right\|^2$$

$$+ \frac{3\xi_1(c_1 + c_2)}{2} \Gamma^2(\epsilon, \epsilon_1) + \left( \frac{c_1 + c_2}{2\xi_1} + \frac{1}{2\alpha} \right) \mathbb{E}_{t-\tau} \left\| \bar{\theta}_t - \bar{\theta}_{t-\tau} \right\|^2 + \frac{\alpha}{2} \left[ \frac{d_2^2}{NK} + \underbrace{4L_2^2 \rho^{2\tau K}}_{\leq 4L_2^2 \alpha^2} \right], \qquad (63)$$

where $(a)$ is due to $g_i(\theta_{t,k}^{(i)}) = -A_i(O_{t,k}^{(i)})(\theta_{t,k}^{(i)} - \theta_i^*) + Z_i(O_{t,k}^{(i)})$, $(b)$ is due to Lemma 12 (the upper bound of $A_i(O_{t,k}^{(i)})$ and $\bar{A}_i$), $(c)$ is due to Eq (13) and $(d)$ is due to Lemma 13.

And we bound $\mathcal{B}_{22}$ as:

$$\mathcal{B}_{22} = \left\langle \frac{1}{NK} \sum_{i=1}^{N} \sum_{k=0}^{K-1} \left[ g_i(\theta_{t,k}^{(i)}) - g_i(\theta_{t-\tau,k}^{(i)}) - \bar{g}_i(\theta_{t,k}^{(i)}) + \bar{g}_i(\theta_{t-\tau,k}^{(i)}) \right], \bar{\theta}_{t-\tau} - \theta^* \right\rangle$$

$$\leq \frac{1}{NK} \sum_{i=1}^{N} \sum_{k=0}^{K-1} \left\| g_i(\theta_{t,k}^{(i)}) - g_i(\theta_{t-\tau,k}^{(i)}) - \bar{g}_i(\theta_{t,k}^{(i)}) + \bar{g}_i(\theta_{t-\tau,k}^{(i)}) \right\| \left\| \bar{\theta}_{t-\tau} - \theta^* \right\|$$

(Cauchy-Schwarz inequality)

$$\leq \frac{1}{NK} \sum_{i=1}^{N} \sum_{k=0}^{K-1} \left[ \left\| g_i(\theta_{t,k}^{(i)}) - g_i(\theta_{t-\tau,k}^{(i)}) \right\| + \left\| \bar{g}_i(\theta_{t,k}^{(i)}) - \bar{g}_i(\theta_{t-\tau,k}^{(i)}) \right\| \right] \left\| \bar{\theta}_{t-\tau} - \theta^* \right\|$$

$$\overset{(a)}{\leq} \frac{1}{NK} \sum_{i=1}^{N} \sum_{k=0}^{K-1} \left[ 2 \left\| \theta_{t,k}^{(i)} - \theta_{t-\tau,k}^{(i)} \right\| + 2 \left\| \theta_{t,k}^{(i)} - \theta_{t-\tau,k}^{(i)} \right\| \right] \left\| \bar{\theta}_{t-\tau} - \theta^* \right\|$$

$$\leq \frac{1}{NK} \sum_{i=1}^{N} \sum_{k=0}^{K-1} \left[ 4 \left\| \theta_{t,k}^{(i)} - \bar{\theta}_t \right\| + 4 \left\| \bar{\theta}_t - \bar{\theta}_{t-\tau} \right\| + 4 \left\| \bar{\theta}_{t-\tau} - \theta_{t-\tau,k}^{(i)} \right\| \right] \left\| \bar{\theta}_{t-\tau} - \theta^* \right\|$$

(Triangle inequality)

$$\leq 4\delta_t \left\| \bar{\theta}_{t-\tau} - \theta^* \right\| + 4 \left\| \bar{\theta}_t - \bar{\theta}_{t-\tau} \right\| \left\| \bar{\theta}_{t-\tau} - \theta^* \right\| + 4\delta_{t-\tau} \left\| \bar{\theta}_{t-\tau} - \theta^* \right\|$$

$$\leq \frac{2}{\xi_2} \Delta_t + \frac{2}{\xi_2} \Delta_{t-\tau} + (2\xi_2 + 4\xi_2) \left\| \bar{\theta}_{t-\tau} - \theta^* \right\|^2 + \frac{2}{\xi_2} \left\| \bar{\theta}_t - \bar{\theta}_{t-\tau} \right\|^2$$

(Young's inequality (12) with constants $\xi_2$ and $\delta_t^2 \leq \Delta_t$)

$$\leq \frac{2}{\xi_2} \Delta_t + \frac{2}{\xi_2} \Delta_{t-\tau} + 12\xi_2 \left\| \bar{\theta}_t - \theta^* \right\|^2 + (12\xi_2 + \frac{2}{\xi_2}) \left\| \bar{\theta}_t - \bar{\theta}_{t-\tau} \right\|^2 \quad \text{(Eq 13)} \tag{64}$$

where (a) is due to the 2-Lipschitz property of steady-state $\bar{g}$ (i.e., Lemma 5) and random direction $g_i$ (i.e., Lemma 6), $\delta_t = \frac{1}{NK} \sum_{i=1}^{N} \sum_{k=0}^{K-1} \left\| \theta_{t,k}^{(i)} - \bar{\theta}_t \right\|$ and $\Delta_t \triangleq \frac{1}{NK} \sum_{i=1}^{N} \sum_{k=0}^{K-1} \mathbb{E} \left\| \theta_{t,k}^{(i)} - \bar{\theta}_t \right\|^2$.

Now, we bound $\mathcal{B}_{23}$ as:

$$\mathbb{E}_{t-\tau}[\mathcal{B}_{23}]$$

$$= \left\langle \frac{1}{NK} \sum_{i=1}^{N} \sum_{k=0}^{K-1} \mathbb{E}_{t-\tau} \left[ g_i(\theta_{t-\tau,k}^{(i)}) - \bar{g}_i(\theta_{t-\tau,k}^{(i)}) \right], \bar{\theta}_{t-\tau} - \theta^* \right\rangle$$

$$\leq \frac{1}{NK} \sum_{i=1}^{N} \sum_{k=0}^{K-1} \left\| \bar{\theta}_{t-\tau} - \theta^* \right\| \left\| \mathbb{E}_{t-\tau} \left[ g_i(\theta_{t-\tau,k}^{(i)}) - \bar{g}_i(\theta_{t-\tau,k}^{(i)}) \right] \right\| \quad \text{(Cauchy-Schwarz inequality)}$$

$$= \frac{1}{NK} \sum_{i=1}^{N} \sum_{k=0}^{K-1} \left\| \bar{\theta}_{t-\tau} - \theta^* \right\| \left\| \mathbb{E}_{t-\tau} \left[ -A_i(O_{t,k}^{(i)})(\theta_{t-\tau,k}^{(i)} - \theta_i^*) + Z_i(O_{t,k}^{(i)}) + \bar{A}_i(\theta_{t-\tau,k}^{(i)} - \theta_i^*) \right] \right\|$$

$$\leq \frac{1}{NK} \sum_{i=1}^{N} \sum_{k=0}^{K-1} \left\| \bar{\theta}_{t-\tau} - \theta^* \right\| \left\{ \left\| \mathbb{E}_{t-\tau} (A_i(O_{t,k}^{(i)}) - \bar{A}_i)(\theta_{t-\tau,k}^{(i)} - \theta_i^*) \right\| + \left\| \mathbb{E}_{t-\tau} \left[ Z_i(O_{t,k}^{(i)}) \right] \right\| \right\}$$

$$\overset{(a)}{\leq} \frac{1}{NK} \sum_{i=1}^{N} \sum_{k=0}^{K-1} \left\| \bar{\theta}_{t-\tau} - \theta^* \right\| \left\{ L_1 \rho^{\tau K + k} \left\| \theta_{t-\tau,k}^{(i)} - \theta_i^* \right\| + L_2 \rho^{\tau K + k} \right\}$$

$$\leq \frac{1}{NK} \sum_{i=1}^{N} \sum_{k=0}^{K-1} \left\| \bar{\theta}_{t-\tau} - \theta^* \right\|$$

$$\times \left\{ L_1 \rho^{\tau K + k} \left[ \left\| \theta_{t-\tau,k}^{(i)} - \bar{\theta}_{t-\tau} \right\| + \left\| \bar{\theta}_{t-\tau} - \theta^* \right\| + \left\| \theta^* - \theta_i^* \right\| \right] + L_2 \rho^{\tau K + k} \right\}$$

$$\overset{(b)}{\leq} \alpha^2 L_1 \left\| \bar{\theta}_{t-\tau} - \theta^* \right\| \delta_{t-\tau} + \alpha^2 L_1 \left\| \bar{\theta}_{t-\tau} - \theta^* \right\|^2 + \alpha^2 L_1 \Gamma(\epsilon, \epsilon_1) G + \alpha^2 L_2 G$$

$$\leq \alpha^2 L_1 \left\| \bar{\theta}_{t-\tau} - \theta^* \right\|^2 + \alpha^2 L_1 \Delta_{t-\tau} + \alpha^2 L_1 \left\| \bar{\theta}_{t-\tau} - \theta^* \right\|^2 + \alpha^2 L_1 \Gamma(\epsilon, \epsilon_1) G + \alpha^2 L_2 G$$

$$\overset{(c)}{\leq} 2\alpha^2 L_1 G^2 + \alpha^2 L_2 G + \alpha^2 L_1 \Delta_{t-\tau} + \alpha^2 L_1 \Gamma(\epsilon, \epsilon_1) G, \tag{65}$$

where (a) is due to Lemma 12, (b) is due to the fact that $\bar{\theta}_{t-\tau}, \theta^* \in \mathcal{H}$, which radius is $H \leq \frac{G}{2}$, and $\tau = \lceil \frac{\log_\rho(\alpha_T^2)}{K} \rceil$ (i.e., $\rho^{\tau K} \leq \alpha^2$) and (c) is due to the fact that $\bar{\theta}_{t-\tau}, \theta^* \in \mathcal{H}$. Then, the term $\mathcal{B}_2$ can be

bounded as:

$$\mathbb{E}_{t-\tau}[\mathcal{B}_2]$$

$$= \mathbb{E}_{t-\tau}[\mathcal{B}_{21} + \mathcal{B}_{22} + \mathcal{B}_{23}]$$

$$\leq \frac{3\xi_1(c_1+c_2)}{2}\mathbb{E}_{t-\tau}[\Delta_t] + \frac{3\xi_1(c_1+c_2)}{2}\mathbb{E}_{t-\tau}\left\|\bar{\theta}_t - \theta^*\right\|^2 + \frac{3\xi_1(c_1+c_2)}{2}\Gamma^2(\epsilon,\epsilon_1)$$

$$+ \left(\frac{c_1+c_2}{2\xi_1} + \frac{1}{2\alpha}\right)\mathbb{E}_{t-\tau}\left\|\bar{\theta}_t - \bar{\theta}_{t-\tau}\right\|^2 + \frac{\alpha}{2}\left[\frac{d_2^2}{NK} + 4L_2^2\rho^{2\tau K}\right]$$

$$+ \frac{2}{\xi_2}\mathbb{E}_{t-\tau}[\Delta_t] + \frac{2}{\xi_2}\mathbb{E}_{t-\tau}[\Delta_{t-\tau}] + 12\xi_2\mathbb{E}_{t-\tau}\left\|\bar{\theta}_t - \theta^*\right\|^2 + (12\xi_2 + \frac{2}{\xi_2})\mathbb{E}_{t-\tau}\left\|\bar{\theta}_t - \bar{\theta}_{t-\tau}\right\|^2$$

$$+ 2\alpha^2 L_1 G^2 + \alpha^2 L_2 G + \alpha^2 L_1 \mathbb{E}_{t-\tau}[\Delta_{t-\tau}] + \alpha^2 L_1 \Gamma(\epsilon,\epsilon_1)G$$

$$\leq \left(\frac{3\xi_1(c_1+c_2)}{2} + 12\xi_2\right)\mathbb{E}_{t-\tau}\left\|\bar{\theta}_t - \theta^*\right\|^2 + \left(\frac{c_1+c_2}{2\xi_1} + \frac{1}{2\alpha} + 12\xi_2 + \frac{2}{\xi_2}\right)\mathbb{E}_{t-\tau}\left\|\bar{\theta}_t - \bar{\theta}_{t-\tau}\right\|^2$$

$$+ \left(\frac{3\xi_1(c_1+c_2)}{2} + \frac{2}{\xi_2}\right)\mathbb{E}_{t-\tau}[\Delta_t] + \left(\frac{2}{\xi_2} + \alpha^2 L_1\right)\Delta_{t-\tau} + \frac{\alpha}{2}\left[\frac{d_2^2}{NK} + 4L_2^2\alpha^2\right]$$

$$+ 2\alpha^2 L_1 G^2 + \alpha^2 L_2 G + \frac{3\xi_1(c_1+c_2)}{2}\Gamma^2(\epsilon,\epsilon_1) + \alpha^2 L_1 \Gamma(\epsilon,\epsilon_1)G \tag{66}$$

Next, we bound $\mathcal{B}_3$ as:

$$\mathbb{E}_{t-\tau}[\mathcal{B}_3]$$

$$= \mathbb{E}_{t-\tau}\left\|\frac{1}{NK}\sum_{i=1}^{N}\sum_{k=0}^{K-1}\left[g_i(\theta_{t,k}^{(i)}) - \bar{g}_i(\theta_{t,k}^{(i)}) + \bar{g}_i(\theta_{t,k}^{(i)}) - \bar{g}_i(\bar{\theta}_t) + \bar{g}_i(\bar{\theta}_t) - \bar{g}(\bar{\theta}_t) + \bar{g}(\bar{\theta}_t)\right]\right\|^2$$

$$\leq 4\mathbb{E}_{t-\tau}\left\|\frac{1}{NK}\sum_{i=1}^{N}\sum_{k=0}^{K-1}\left(g_i(\theta_{t,k}^{(i)}) - \bar{g}_i(\theta_{t,k}^{(i)})\right)\right\|^2 + 4\mathbb{E}_{t-\tau}\left\|\frac{1}{NK}\sum_{i=1}^{N}\sum_{k=0}^{K-1}\left(\bar{g}_i(\theta_{t,k}^{(i)}) - \bar{g}_i(\bar{\theta}_t)\right)\right\|^2$$

$$+ 4\mathbb{E}_{t-\tau}\left\|\frac{1}{NK}\sum_{i=1}^{N}\sum_{k=0}^{K-1}\left(\bar{g}_i(\bar{\theta}_t) - \bar{g}(\bar{\theta}_t)\right)\right\|^2 + 4\mathbb{E}_{t-\tau}\left\|\frac{1}{NK}\sum_{i=1}^{N}\sum_{k=0}^{K-1}\bar{g}(\bar{\theta}_t)\right\|^2 \quad \text{(Eq 13)}$$

$$= 4\mathbb{E}_{t-\tau}\left\|\frac{1}{NK}\sum_{i=1}^{N}\sum_{k=0}^{K-1}\left[\left(\bar{A}_i - A_i(O_{t,k}^{(i)})\right)(\theta_{t,k}^{(i)} - \theta_i^*) + Z_i(O_{t,k}^{(i)})\right]\right\|^2 + 4\mathbb{E}_{t-\tau}\left\|\bar{g}(\bar{\theta}_t)\right\|^2$$

$$+ 4\mathbb{E}_{t-\tau}\left\|\frac{1}{NK}\sum_{i=1}^{N}\sum_{k=0}^{K-1}\left(\bar{g}_i(\theta_{t,k}^{(i)}) - \bar{g}_i(\bar{\theta}_t)\right)\right\|^2 + 4\mathbb{E}_{t-\tau}\underbrace{\left\|\frac{1}{N}\sum_{i=1}^{N}\left(\bar{g}_i(\bar{\theta}_t) - \bar{g}(\bar{\theta}_t)\right)\right\|^2}_{\text{Lemma 2}}$$

$$\overset{(a)}{\leq} 8\mathbb{E}_{t-\tau}\left\|\frac{1}{NK}\sum_{i=1}^{N}\sum_{k=0}^{K-1}\left(\bar{A}_i - A_i(O_{t,k}^{(i)})\right)(\theta_{t,k}^{(i)} - \theta_i^*)\right\|^2 + 8\mathbb{E}_{t-\tau}\left\|\frac{1}{NK}\sum_{i=1}^{N}\sum_{k=0}^{K-1}Z_i(O_{t,k}^{(i)})\right\|^2$$

$$+ 16\frac{1}{NK}\sum_{i=1}^{N}\sum_{k=0}^{K-1}\mathbb{E}_{t-\tau}\left\|\theta_{t,k}^{(i)} - \bar{\theta}_t\right\|^2 + 4B^2(\epsilon,\epsilon_1) + 4\mathbb{E}_{t-\tau}\left\|\bar{g}(\bar{\theta}_t)\right\|^2$$

$$\leq \frac{8}{NK}\sum_{i=1}^{N}\sum_{k=0}^{K-1}\mathbb{E}_{t-\tau}\left\|\bar{A}_i - A_i(O_{t,k}^{(i)})\right\|^2\left\|\theta_{t,k}^{(i)} - \theta_i^*\right\|^2 + 8\mathbb{E}_{t-\tau}\left\|\frac{1}{NK}\sum_{i=1}^{N}\sum_{k=0}^{K-1}Z_i(O_{t,k}^{(i)})\right\|^2$$

$$+ 16\mathbb{E}_{t-\tau}[\Delta_t] + 4B^2(\epsilon,\epsilon_1) + 4\mathbb{E}_{t-\tau}\left\|\bar{g}(\bar{\theta}_t)\right\|^2$$

$$\leq \frac{8(c_1+c_2)^2}{NK}\sum_{i=1}^{N}\sum_{k=0}^{K-1}\mathbb{E}_{t-\tau}\left\|\theta_{t,k}^{(i)} - \theta_i^*\right\|^2 + 8\mathbb{E}_{t-\tau}\left\|\frac{1}{NK}\sum_{i=1}^{N}\sum_{k=0}^{K-1}Z_i(O_{t,k}^{(i)})\right\|^2$$

$$+ 16\mathbb{E}_{t-\tau}[\Delta_t] + 4B^2(\epsilon,\epsilon_1) + 4\mathbb{E}_{t-\tau}\left\|\bar{g}(\bar{\theta}_t)\right\|^2$$

$$\leq \frac{24(c_1+c_2)^2}{NK} \sum_{i=1}^{N} \sum_{k=0}^{K-1} \mathbb{E}_{t-\tau} \left\| \theta_{t,k}^{(i)} - \bar{\theta}_t \right\|^2 + \frac{24(c_1+c_2)^2}{NK} \sum_{i=1}^{N} \sum_{k=0}^{K-1} \mathbb{E}_{t-\tau} \left\| \bar{\theta}_t - \theta^* \right\|^2$$

$$+ \frac{24(c_1+c_2)^2}{NK} \sum_{i=1}^{N} \sum_{k=0}^{K-1} \mathbb{E}_{t-\tau} \left\| \theta_i^* - \theta^* \right\|^2 + 8\mathbb{E}_{t-\tau} \left\| \frac{1}{NK} \sum_{i=1}^{N} \sum_{k=0}^{K-1} Z_i(O_{t,k}^{(i)}) \right\|^2$$

$$+ 16\mathbb{E}_{t-\tau}[\Delta_t] + 4B^2(\epsilon, \epsilon_1) + 4\mathbb{E}_{t-\tau} \left\| \bar{g}(\bar{\theta}_t) \right\|^2$$

$$= 24(c_1+c_2)^2 \mathbb{E}_{t-\tau}[\Delta_t] + 24(c_1+c_2)^2 \mathbb{E}_{t-\tau} \left\| \bar{\theta}_t - \theta^* \right\|^2 + 24(c_1+c_2)^2 \Gamma^2(\epsilon, \epsilon_1)$$

$$+ 8\mathbb{E}_{t-\tau} \left\| \frac{1}{NK} \sum_{i=1}^{N} \sum_{k=0}^{K-1} Z_i(O_{t,k}^{(i)}) \right\|^2 + 16\mathbb{E}_{t-\tau}[\Delta_t] + 4B^2(\epsilon, \epsilon_1) + 4\mathbb{E}_{t-\tau} \left\| \bar{g}(\bar{\theta}_t) \right\|^2$$

$$\overset{(b)}{\leq} (24(c_1+c_2)^2 + 16)\mathbb{E}_{t-\tau}[\Delta_t] + 8(\frac{d_2^2}{NK} + \underbrace{4L_2^2 \rho^{\tau K}}_{\leq 4L_2^2 \alpha^2}) + 24(c_1+c_2)^2 \mathbb{E}_{t-\tau} \left\| \bar{\theta}_t - \theta^* \right\|^2$$

$$+ 4\mathbb{E}_{t-\tau} \left\| \bar{g}(\bar{\theta}_t) \right\|^2 + 4B^2(\epsilon, \epsilon_1) + 24(c_1+c_2)^2 \Gamma^2(\epsilon, \epsilon_1), \tag{67}$$

where (a) is due to 2-Lipschitz of $\bar{g}_i$ (i.e., Lemma 5) and the gradient heterogeneity (i.e., Lemma 2) and (b) is due to Lemma 13.

Next, we bound $\mathcal{B}_1$ as:

$$\mathbb{E}_{t-\tau}[\mathcal{B}_1]$$

$$= \mathbb{E}_{t-\tau} \left\| \bar{\theta}_t - \theta^* \right\|^2 + 2\mathbb{E}_{t-\tau} \left\langle \frac{\alpha}{N} \sum_{i=1}^{N} \bar{g}_i(\bar{\theta}_t), \bar{\theta}_t - \theta^* \right\rangle$$

$$+ 2\mathbb{E}_{t-\tau} \left\langle \frac{\alpha}{NK} \sum_{i=1}^{N} \sum_{k=0}^{K-1} \bar{g}_i(\theta_{t,k}^{(i)}) - \bar{g}_i(\bar{\theta}_t), \bar{\theta}_t - \theta^* \right\rangle$$

$$\leq \mathbb{E}_{t-\tau} \left\| \bar{\theta}_t - \theta^* \right\|^2 + 2\alpha \mathbb{E}_{t-\tau} \left\langle \frac{1}{N} \sum_{i=1}^{N} \bar{g}_i(\bar{\theta}_t) - \bar{g}(\bar{\theta}_t), \bar{\theta}_t - \theta^* \right\rangle + 2\alpha \mathbb{E}_{t-\tau} \left\langle \bar{g}(\bar{\theta}_t), \bar{\theta}_t - \theta^* \right\rangle$$

$$+ 2\alpha \mathbb{E}_{t-\tau} \left\langle \frac{1}{NK} \sum_{i=1}^{N} \sum_{k=0}^{K-1} \bar{g}_i(\theta_{t,k}^{(i)}) - \bar{g}_i(\bar{\theta}_t), \bar{\theta}_t - \theta^* \right\rangle$$

$$\leq \mathbb{E}_{t-\tau} \left\| \bar{\theta}_t - \theta^* \right\|^2 + 2\alpha \mathbb{E}_{t-\tau} \left\| \frac{1}{N} \sum_{i=1}^{N} \bar{g}_i(\bar{\theta}_t) - \bar{g}(\bar{\theta}_t) \right\| \left\| \bar{\theta}_t - \theta^* \right\| + 2\alpha \mathbb{E}_{t-\tau} \left\langle \bar{g}(\bar{\theta}_t), \bar{\theta}_t - \theta^* \right\rangle$$

$$+ \frac{\alpha}{\xi_3} \mathbb{E}_{t-\tau} \left\| \frac{1}{NK} \sum_{i=1}^{N} \sum_{k=0}^{K-1} \left( \bar{g}_i(\theta_{t,k}^{(i)}) - \bar{g}_i(\bar{\theta}_t) \right) \right\|^2 + \alpha\xi_3 \mathbb{E}_{t-\tau} \left\| \bar{\theta}_t - \theta^* \right\|^2$$

$$\text{(Young's inequality Eq (12) with constant } \xi_3\text{)}$$

$$\overset{(a)}{\leq} \mathbb{E}_{t-\tau} \left\| \bar{\theta}_t - \theta^* \right\|^2 + 2\alpha B(\epsilon, \epsilon_1)G + 2\alpha \mathbb{E}_{t-\tau} \left\langle \bar{g}(\bar{\theta}_t), \bar{\theta}_t - \theta^* \right\rangle$$

$$+ \frac{\alpha}{\xi_3} \mathbb{E}_{t-\tau} \left\| \frac{1}{NK} \sum_{i=1}^{N} \sum_{k=0}^{K-1} \left( \bar{g}_i(\theta_{t,k}^{(i)}) - \bar{g}_i(\bar{\theta}_t) \right) \right\|^2 + \alpha\xi_3 \mathbb{E}_{t-\tau} \left\| \bar{\theta}_t - \theta^* \right\|^2$$

$$\overset{(b)}{\leq} \mathbb{E}_{t-\tau} \left\| \bar{\theta}_t - \theta^* \right\|^2 + 2\alpha B(\epsilon, \epsilon_1)G + 2\alpha \mathbb{E}_{t-\tau} \left\langle \bar{g}(\bar{\theta}_t), \bar{\theta}_t - \theta^* \right\rangle$$

$$+ \frac{4\alpha}{\xi_3} \mathbb{E}_{t-\tau}[\Delta_t] + \alpha\xi_3 \mathbb{E}_{t-\tau} \left\| \bar{\theta}_t - \theta^* \right\|^2, \tag{68}$$

where (a) is due to the fact that $\bar{\theta}_t, \theta^* \in \mathcal{H}$ and the gradient heterogeneity; (b) is due to 2-Lipschitz property of function $\bar{g}$ in Lemma 5.

Incorporating the upper of $\mathcal{B}_1$ from Eq (68), $\mathcal{B}_2$ from Eq (66) and $\mathcal{B}_3$ from Eq (67) into Eq (61), we have:

$$
\begin{aligned}
\mathbb{E}_{t-\tau}\left\|\bar{\theta}_{t+1}-\theta^*\right\|^2 \leq\ & \mathbb{E}_{t-\tau}\left\|\bar{\theta}_t-\theta^*\right\|^2 + 2\alpha\mathbb{E}_{t-\tau}\left\langle\bar{g}(\bar{\theta}_t),\bar{\theta}_t-\theta^*\right\rangle + 4\alpha^2\mathbb{E}_{t-\tau}\left\|\bar{g}(\bar{\theta}_t)\right\|^2 \\
& + \left(\alpha\xi_3 + \alpha(3\xi_1(c_1+c_2)+24\xi_2)+24\alpha^2(c_1+c_2)^2\right)\mathbb{E}_{t-\tau}\left\|\bar{\theta}_t-\theta^*\right\|^2 \\
& + \alpha\left(\frac{c_1+c_2}{\xi_1}+\frac{1}{\alpha}+24\xi_2+\frac{4}{\xi_2}\right)\mathbb{E}_{t-\tau}\left\|\bar{\theta}_t-\bar{\theta}_{t-\tau}\right\|^2 \\
& + \frac{9d_2^2}{NK}\alpha^2 + 36L_2^2\alpha^4 + 4\alpha^3L_1G^2 + 2\alpha^3L_2G + \left(\frac{4\alpha}{\xi_2}+2\alpha^3L_1\right)\Delta_{t-\tau} \\
& + \alpha\left(\frac{4}{\xi_3}+3\xi_1(c_1+c_2)+\frac{4}{\xi_2}+\alpha^2\left(24(c_1+c_2)^2+16\right)\right)\mathbb{E}_{t-\tau}[\Delta_t] \\
& + 2\alpha B(\epsilon,\epsilon_1)G + 4\alpha^2B^2(\epsilon,\epsilon_1) + 24\alpha^2(c_1+c_2)^2\Gamma^2(\epsilon,\epsilon_1) \\
& + 3\alpha\xi_1(c_1+c_2)\Gamma^2(\epsilon,\epsilon_1) + 2\alpha^3L_1\Gamma(\epsilon,\epsilon_1)G
\end{aligned}
\tag{69}
$$

Conditioned on $\mathcal{F}_{t-2\tau}$ and using Lemma 14 to give an upper bound of $\mathbb{E}_{t-2\tau}\left\|\bar{\theta}_t-\bar{\theta}_{t-\tau}\right\|^2$, we have:

$$
\begin{aligned}
& \mathbb{E}_{t-2\tau}\left\|\bar{\theta}_{t+1}-\theta^*\right\|^2 \\
\leq\ & \mathbb{E}_{t-2\tau}\left\|\bar{\theta}_t-\theta^*\right\|^2 + 2\alpha\mathbb{E}_{t-2\tau}\left\langle\bar{g}(\bar{\theta}_t),\bar{\theta}_t-\theta^*\right\rangle + 4\alpha^2\mathbb{E}_{t-2\tau}\left\|\bar{g}(\bar{\theta}_t)\right\|^2 \\
& + \underbrace{\left(\alpha\xi_3 + \alpha(3\xi_1(c_1+c_2)+24\xi_2)+24\alpha^2(c_1+c_2)^2\right)}_{\mathcal{E}_1}\mathbb{E}_{t-2\tau}\left\|\bar{\theta}_t-\theta^*\right\|^2 \\
& + \alpha\underbrace{\left(\frac{c_1+c_2}{\xi_1}+\frac{1}{\alpha}+24\xi_2+\frac{4}{\xi_2}\right)}_{\mathcal{E}_2}\left\{8\alpha^2\tau^2c_4^2\mathbb{E}_{t-2\tau}\left[\left\|\bar{\theta}_t-\theta^*\right\|^2\right]+14\alpha^2\tau^2\frac{d_2^2}{NK}+\frac{52L_2^2\alpha^4\tau}{1-\rho^2}\right. \\
& \left. + 4\alpha^2c_4^2\tau\sum_{s=0}^{\tau}E_{t-2\tau}[\Delta_{t-s}]+3200\alpha^2c_1^2c_4^2\tau^3\Gamma^2(\epsilon,\epsilon_1)+4\alpha^2c_1^2\tau^2\Gamma^2(\epsilon,\epsilon_1)\right\} \\
& + \frac{9d_2^2}{NK}\alpha^2 + 36L_2^2\alpha^4 + 4\alpha^3L_1G^2 + 2\alpha^3L_2G + \left(\frac{4\alpha}{\xi_2}+2\alpha^3L_1\right)\mathbb{E}_{t-2\tau}[\Delta_{t-\tau}] \\
& + \alpha\underbrace{\left(\frac{4}{\xi_3}+3\xi_1(c_1+c_2)+\frac{4}{\xi_2}+\alpha^2\left(24(c_1+c_2)^2+16\right)\right)}_{\mathcal{E}_3}\mathbb{E}_{t-2\tau}[\Delta_t] \\
& + 2\alpha B(\epsilon,\epsilon_1)G + 4\alpha^2B^2(\epsilon,\epsilon_1) + 24\alpha^2(c_1+c_2)^2\Gamma^2(\epsilon,\epsilon_1) \\
& + 3\alpha\xi_1(c_1+c_2)\Gamma^2(\epsilon,\epsilon_1) + 2\alpha^3L_1\Gamma(\epsilon,\epsilon_1)G
\end{aligned}
\tag{70}
$$

If we choose step-size $\alpha$ such that $\alpha\mathcal{E}_2 = \alpha\left(\frac{c_1+c_2}{\xi_1}+\frac{1}{\alpha}+24\xi_2+\frac{4}{\xi_2}\right) \leq 2$, $\xi_1 = \xi_2 = \xi_3$, $\mathcal{E}_1 = \alpha\xi_3 + \alpha(3\xi_1(c_1+c_2)+24\xi_2)+24\alpha^2(c_1+c_2)^2 \leq 28\alpha\xi_1(c_1+c_2)+24\alpha^2(c_1+c_2)^2 \leq 30\alpha\xi_1(c_1+c_2)$ $(c_1,c_2>1)$ and $\mathcal{E}_3 = \frac{4}{\xi_3}+3\xi_1(c_1+c_2)+\frac{4}{\xi_2}+\alpha^2\left(24(c_1+c_2)^2+16\right) \leq (\frac{9}{\xi_1}+9\xi_1)(c_1+c_2)$, i.e.,

$$
\alpha \leq \frac{1}{\left(\frac{c_1+c_2}{\xi_1}+24\xi_2+\frac{4}{\xi_2}\right)} = \frac{\xi_1}{(c_1+c_2+24\xi_1^2+4)}
$$

$$
\alpha \leq \min\{\frac{\xi_1}{12(c_1+c_2)}, 1, \frac{(\frac{5}{\xi_1}+5\xi_1)(c_1+c_2)}{24(c_1+c_2)^2+16}\},
$$

which is sufficient to hold when $\alpha \leq \min\{\frac{\xi_1}{24(c_1+c_2)^2+24\xi_1^2+16}, 1\}$, then we have:

$$
\mathbb{E}_{t-2\tau}\left\|\bar{\theta}_{t+1}-\theta^*\right\|^2
$$

$$\leq \mathbb{E}_{t-2\tau}\left\|\bar{\theta}_t - \theta^*\right\|^2 + 2\alpha\mathbb{E}_{t-2\tau}\left\langle \bar{g}(\bar{\theta}_t), \bar{\theta}_t - \theta^*\right\rangle + 4\alpha^2\mathbb{E}_{t-2\tau}\left\|\bar{g}(\bar{\theta}_t)\right\|^2$$

$$+ 30\alpha\xi_1(c_1 + c_2)\mathbb{E}_{t-2\tau}\left\|\bar{\theta}_t - \theta^*\right\|^2$$

$$+ 2\left\{8\alpha^2\tau^2 c_4^2 \mathbb{E}_{t-2\tau}\left[\left\|\bar{\theta}_t - \theta^*\right\|^2\right] + 14\alpha^2\tau^2\frac{d_2^2}{NK} + \frac{52L_2^2\alpha^4\tau}{1-\rho^2}\right.$$

$$\left. + 4\alpha^2 c_4^2\tau\sum_{s=0}^{\tau} E_{t-2\tau}[\Delta_{t-s}] + 3200\alpha^2 c_1^2 c_4^2\tau^3\Gamma^2(\epsilon, \epsilon_1) + 4\alpha^2 c_1^2\tau^2\Gamma^2(\epsilon, \epsilon_1)\right\}$$

$$+ \frac{9d_2^2}{NK}\alpha^2 + 36L_2^2\alpha^4 + 4\alpha^3 L_1 G^2 + 2\alpha^3 L_2 G + \left(\frac{4\alpha}{\xi_2} + 2\alpha^3 L_1\right)\mathbb{E}_{t-2\tau}[\Delta_{t-\tau}]$$

$$+ \alpha\left(\frac{4}{\xi_3} + 3\xi_1(c_1 + c_2) + \frac{4}{\xi_2} + \alpha^2\left(24(c_1 + c_2)^2 + 16\right)\right)\mathbb{E}_{t-2\tau}[\Delta_t]$$

$$+ 2\alpha B(\epsilon, \epsilon_1)G + 4\alpha^2 B^2(\epsilon, \epsilon_1) + 24\alpha^2(c_1 + c_2)^2\Gamma^2(\epsilon, \epsilon_1)$$

$$+ 3\alpha\xi_1(c_1 + c_2)\Gamma^2(\epsilon, \epsilon_1) + 2\alpha^3 L_1\Gamma(\epsilon, \epsilon_1)G$$

$$\leq \mathbb{E}_{t-2\tau}\left\|\bar{\theta}_t - \theta^*\right\|^2 + 2\alpha\mathbb{E}_{t-2\tau}\left\langle \bar{g}(\bar{\theta}_t), \bar{\theta}_t - \theta^*\right\rangle + 4\alpha^2\mathbb{E}_{t-2\tau}\left\|\bar{g}(\bar{\theta}_t)\right\|^2$$

$$+ \left(30\alpha\xi_1(c_1 + c_2) + 16\alpha^2\tau^2 c_4^2\right)\mathbb{E}_{t-2\tau}\left\|\bar{\theta}_t - \theta^*\right\|^2$$

$$+ \frac{9 + 28\tau^2}{NK}\alpha^2 d_2^2 + 36\left(1 + \frac{3\tau}{1-\rho^2}\right)L_2^2\alpha^4 + 4\alpha^3 L_1 G^2 + 2\alpha^3 L_2 G$$

$$+ \left(\frac{4\alpha}{\xi_1} + 2\alpha^3 L_1\right)\mathbb{E}_{t-2\tau}[\Delta_{t-\tau}] + \alpha(\frac{9}{\xi_1} + 9\xi_1)(c_1 + c_2)\mathbb{E}_{t-2\tau}[\Delta_t] + 8\alpha^2 c_4^2\tau\sum_{s=0}^{\tau} E_{t-2\tau}[\Delta_{t-s}]$$

$$+ 2\alpha B(\epsilon, \epsilon_1)G + 4\alpha^2 B^2(\epsilon, \epsilon_1) + 24\alpha^2(c_1 + c_2)^2\Gamma^2(\epsilon, \epsilon_1)$$

$$+ 3\alpha\xi_1(c_1 + c_2)\Gamma^2(\epsilon, \epsilon_1) + 2\alpha^3 L_1\Gamma(\epsilon, \epsilon_1)G$$

$$+ 6400\alpha^2 c_1^2 c_4^2\tau^3\Gamma^2(\epsilon, \epsilon_1) + 8\alpha^2 c_1^2\tau^2\Gamma^2(\epsilon, \epsilon_1) \tag{71}$$

if we choose the step-size $\alpha$ such that the high order $O(\alpha^2)$ terms are dominanted by the first order terms $O(\alpha)$, i.e., $4\alpha^2 B^2(\epsilon, \epsilon_1) + 24\alpha^2(c_1 + c_2)^2\Gamma^2(\epsilon, \epsilon_1) + 2\alpha^3 L_1\Gamma(\epsilon, \epsilon_1)G + 6400\alpha^2 c_1^2 c_4^2\tau^3\Gamma^2(\epsilon, \epsilon_1) + 8\alpha^2 c_1^2\tau^2\Gamma^2(\epsilon, \epsilon_1) \leq 2\alpha B(\epsilon, \epsilon_1)G + 3\alpha\xi_1(c_1 + c_2)\Gamma^2(\epsilon, \epsilon_1)$, i.e.,

$$\alpha \leq \min\{\frac{2B(\epsilon, \epsilon_1)G + 3\xi_1(c_1 + c_2)\Gamma^2(\epsilon, \epsilon_1)}{4B^2(\epsilon, \epsilon_1) + 24(c_1 + c_2)^2\Gamma^2(\epsilon, \epsilon_1) + 2L_1\Gamma(\epsilon, \epsilon_1)G + 6400c_1^2 c_4^2\tau^3\Gamma^2(\epsilon, \epsilon_1) + 8c_1^2\tau^2\Gamma^2(\epsilon, \epsilon_1)}, 1\},$$

we have:

$$\mathbb{E}_{t-2\tau}\left\|\bar{\theta}_{t+1} - \theta^*\right\|^2$$

$$\leq \mathbb{E}_{t-2\tau}\left\|\bar{\theta}_t - \theta^*\right\|^2 + 2\alpha\mathbb{E}_{t-2\tau}\left\langle \bar{g}(\bar{\theta}_t), \bar{\theta}_t - \theta^*\right\rangle + 4\alpha^2\mathbb{E}_{t-2\tau}\left\|\bar{g}(\bar{\theta}_t)\right\|^2$$

$$+ \left(30\alpha\xi_1(c_1 + c_2) + 16\alpha^2\tau^2 c_4^2\right)\mathbb{E}_{t-2\tau}\left\|\bar{\theta}_t - \theta^*\right\|^2$$

$$+ \frac{9 + 28\tau^2}{NK}\alpha^2 d_2^2 + 36\left(1 + \frac{3\tau}{1-\rho^2}\right)L_2^2\alpha^4 + 4\alpha^3 L_1 G^2 + 2\alpha^3 L_2 G$$

$$+ \left(\frac{4\alpha}{\xi_1} + 2\alpha^3 L_1\right)\mathbb{E}_{t-2\tau}[\Delta_{t-\tau}] + \alpha(\frac{9}{\xi_1} + 9\xi_1)(c_1 + c_2)\mathbb{E}_{t-2\tau}[\Delta_t] + 8\alpha^2 c_4^2\tau\sum_{s=0}^{\tau} E_{t-2\tau}[\Delta_{t-s}]$$

$$+ 4\alpha B(\epsilon, \epsilon_1)G + 6\alpha\xi_1(c_1 + c_2)\Gamma^2(\epsilon, \epsilon_1) \tag{72}$$

With Lemma (15), we have the upper bound of $\mathbb{E}_{t-2\tau}[\Delta_t]$, $\mathbb{E}_{t-2\tau}[\Delta_{t-\tau}]$ and $\tau\sum_{s=0}^{\tau} E_{t-2\tau}[\Delta_{t-s}]$. Then we have:

$$E_{t-2\tau}\left\|\bar{\theta}_{t+1} - \theta^*\right\|^2$$

$$
\begin{aligned}
&\le \mathbb{E}_{t-2\tau}\left\|\bar{\theta}_t - \theta^*\right\|^2 + 2\alpha\mathbb{E}_{t-2\tau}\left\langle\bar{g}(\bar{\theta}_t), \bar{\theta}_t - \theta^*\right\rangle + 4\alpha^2\mathbb{E}_{t-2\tau}\left\|\bar{g}(\bar{\theta}_t)\right\|^2 \\
&\quad + \underbrace{\left(30\alpha\xi_1(c_1 + c_2) + 16\alpha^2\tau^2 c_4^2\right)}_{\mathcal{E}_4}\mathbb{E}_{t-2\tau}\left\|\bar{\theta}_t - \theta^*\right\|^2 \\
&\quad + \frac{9 + 28\tau^2}{NK}\alpha^2 d_2^2 + \alpha^3\left(36L_2^2 + \frac{108\tau}{1-\rho^2}L_2^2 + 4L_1 G^2 + 2L_2 G\right) \\
&\quad + \frac{4\alpha^2}{K\alpha_g^2}\underbrace{\left(\frac{4\alpha}{\xi_1} + 2\alpha^3 L_1 + \alpha(\frac{9}{\xi_1} + 9\xi_1)(c_1 + c_2) + 8\alpha^2 c_4^2\tau^2\right)}_{\mathcal{E}_5}\left[c_3^2 + \frac{2c_3 L_2\rho}{1-\rho} + 4c_1^2(K-1)H^2\right] \\
&\quad + 4\alpha B(\epsilon, \epsilon_1)G + 6\alpha\xi_1(c_1 + c_2)\Gamma^2(\epsilon, \epsilon_1)
\end{aligned} \tag{73}
$$

If we choose step-size such that $\mathcal{E}_4 = 30\alpha\xi_1(c_1 + c_2) + 16\alpha^2\tau^2 c_4^2 \le 32\alpha\xi_1(c_1 + c_2)$ and $\mathcal{E}_5 = \frac{4\alpha}{\xi_1} + 2\alpha^3 L_1 + \alpha(\frac{9}{\xi_1} + 9\xi_1)(c_1 + c_2) + 8\alpha^2 c_4^2\tau^2 \le \alpha(\frac{14}{\xi_1} + 14\xi_1)(c_1 + c_2)$, i.e.,

$$
\alpha \le \min\{\frac{\xi_1(c_1 + c_2)}{8\tau^2 c_4^2}, 1, \frac{(\frac{1}{\xi_1} + \xi_1)(c_1 + c_2)}{2L_1 + 8c_4^2\tau^2}\},
$$

which is sufficient to hold when $\alpha \le \frac{\xi_1(c_1 + c_2)}{2L_1 + 8\tau^2 c_4^2}$, then we have:

$$
\begin{aligned}
E_{t-2\tau}\left\|\bar{\theta}_{t+1} - \theta^*\right\|^2 &\le \mathbb{E}_{t-2\tau}\left\|\bar{\theta}_t - \theta^*\right\|^2 + 2\alpha\mathbb{E}_{t-2\tau}\left\langle\bar{g}(\bar{\theta}_t), \bar{\theta}_t - \theta^*\right\rangle + 4\alpha^2\mathbb{E}_{t-2\tau}\left\|\bar{g}(\bar{\theta}_t)\right\|^2 \\
&\quad + 32\alpha\xi_1(c_1 + c_2)\mathbb{E}_{t-2\tau}\left\|\bar{\theta}_t - \theta^*\right\|^2 \\
&\quad + \frac{9 + 28\tau^2}{NK}\alpha^2 d_2^2 + \alpha^3\left(36L_2^2 + \frac{108\tau}{1-\rho^2}L_2^2 + 4L_1 G^2 + 2L_2 G\right) \\
&\quad + \frac{4\alpha^3}{K\alpha_g^2}(\frac{14}{\xi_1} + 14\xi_1)(c_1 + c_2)\left[c_3^2 + \frac{2c_3 L_2\rho}{1-\rho} + 8c_1^2(K-1)H^2\right] \\
&\quad + 4\alpha B(\epsilon, \epsilon_1)G + 6\alpha\xi_1(c_1 + c_2)\Gamma^2(\epsilon, \epsilon_1).
\end{aligned} \tag{74}
$$

$\square$

### I.2.6 Parameter Selection

With Lemma 16, we have:

$$
\begin{aligned}
&\mathbb{E}_{t-2\tau}\left\|\bar{\theta}_{t+1} - \theta^*\right\|^2 \\
&\le (1 + 32\alpha\xi_1(c_1 + c_2))\mathbb{E}_{t-2\tau}\left\|\bar{\theta}_t - \theta^*\right\|^2 + 2\alpha\mathbb{E}_{t-2\tau}\left\langle\bar{g}(\bar{\theta}_t), \bar{\theta}_t - \theta^*\right\rangle + 4\alpha^2\mathbb{E}_{t-2\tau}\left\|\bar{g}(\bar{\theta}_t)\right\|^2 \\
&\quad + \frac{9 + 28\tau^2}{NK}\alpha^2 d_2^2 + \alpha^3\left(36L_2^2 + \frac{108\tau}{1-\rho^2}L_2^2 + 4L_1 G^2 + 2L_2 G\right) \\
&\quad + \frac{4\alpha^3}{K\alpha_g^2}(\frac{14}{\xi_1} + 14\xi_1)(c_1 + c_2)\left[c_3^2 + \frac{2c_3 L_2\rho}{1-\rho} + 8c_1^2(K-1)H^2\right] \\
&\quad + 4\alpha B(\epsilon, \epsilon_1)G + 6\alpha\xi_1(c_1 + c_2)\Gamma^2(\epsilon, \epsilon_1).
\end{aligned} \tag{75}
$$

**Proposition 4.** *If $\alpha$ satisfies the requirement as Lemma 16, choose $\xi_1 = \frac{(1-\gamma)\bar{\omega}}{32(c_1 + c_2)}$ and $\tau = \lceil\frac{\tau^{\mathrm{mix}}(\alpha_T^2)}{K}\rceil$, we have:*

$$
\nu_1\mathbb{E}_{t-2\tau}\left\|V_{\bar{\theta}_t} - V_{\theta^*}\right\|_{\bar{D}}^2 \le (\frac{1}{\alpha} - \nu_1)\mathbb{E}_{t-2\tau}\left\|\bar{\theta}_t - \theta^*\right\|^2 - \frac{1}{\alpha}\mathbb{E}_{t-2\tau}\left\|\bar{\theta}_{t+1} - \theta^*\right\|^2 + \frac{9 + 28\tau^2}{NK}\alpha d_2^2
$$

$$+ \alpha^2 \left( 36L_2^2 + \frac{108\tau}{1-\rho^2} L_2^2 + 4L_1 G^2 + 2L_2 G \right)$$

$$+ \frac{\alpha^2 c_6}{K} \left[ c_3^2 + \frac{2c_3 L_2 \rho}{1-\rho} + 8c_1^2 (K-1)H^2 \right] + 4B(\epsilon, \epsilon_1)G + \nu_1 \Gamma^2(\epsilon, \epsilon_1) \tag{76}$$

where $\nu_1 = \frac{\nu}{4} = \frac{(1-\gamma)\bar{\omega}}{4}$ and $c_6 \triangleq \frac{4}{\alpha_g^2}(\frac{14}{\xi_1} + 14\xi_1)(c_1 + c_2)$.

*Proof.* Incorporating $\xi_1 = \frac{(1-\gamma)\bar{\omega}}{32(c_1+c_2)}$, $c_6 \triangleq \frac{4}{\alpha_g^2}(\frac{14}{\xi_1} + 14\xi_1)(c_1 + c_2)$ and $6\xi_1(c_1 + c_2) \leq \nu_1$ into Eq (75), we have

$$\mathbb{E}_{t-2\tau}\left\|\bar{\theta}_{t+1} - \theta^*\right\|^2 \leq \mathbb{E}_{t-2\tau}\left\|\bar{\theta}_t - \theta^*\right\|^2 + 2\alpha\mathbb{E}_{t-2\tau}\left\langle \bar{g}(\bar{\theta}_t), \bar{\theta}_t - \theta^*\right\rangle + 4\alpha^2\mathbb{E}_{t-2\tau}\left\|\bar{g}(\bar{\theta}_t)\right\|^2$$

$$+ \alpha(1-\gamma)\bar{\omega}\mathbb{E}_{t-2\tau}\left\|\bar{\theta}_t - \theta^*\right\|^2$$

$$+ \frac{9+28\tau^2}{NK}\alpha^2 d_2^2 + \alpha^3 \left( 36L_2^2 + \frac{108\tau}{1-\rho^2} L_2^2 + 4L_1 G^2 + 2L_2 G \right)$$

$$+ \frac{\alpha^3 c_6}{K} \left[ c_3^2 + \frac{2c_3 L_2 \rho}{1-\rho} + 8c_1^2 (K-1)H^2 \right]$$

$$+ 4\alpha B(\epsilon, \epsilon_1)G + \alpha\nu_1 \Gamma^2(\epsilon, \epsilon_1)$$

$$\overset{(a)}{\leq} \mathbb{E}_{t-2\tau}\left\|\bar{\theta}_t - \theta^*\right\|^2 - 2\alpha(1-\gamma)\bar{\omega}\mathbb{E}_{t-2\tau}\left\|\bar{\theta}_t - \theta^*\right\|^2 + 16\alpha^2\mathbb{E}_{t-2\tau}\left\|V_{\bar{\theta}_t} - V_{\theta^*}\right\|_{\bar{D}}^2$$

$$+ \alpha(1-\gamma)\bar{\omega}\mathbb{E}_{t-2\tau}\left\|\bar{\theta}_t - \theta^*\right\|^2$$

$$+ \frac{9+28\tau^2}{NK}\alpha^2 d_2^2 + \alpha^3 \left( 36L_2^2 + \frac{108\tau}{1-\rho^2} L_2^2 + 4L_1 G^2 + 2L_2 G \right)$$

$$+ \frac{\alpha^3 c_6}{K} \left[ c_3^2 + \frac{2c_3 L_2 \rho}{1-\rho} + 8c_1^2 (K-1)H^2 \right]$$

$$+ 4\alpha B(\epsilon, \epsilon_1)G + \alpha\nu_1 \Gamma^2(\epsilon, \epsilon_1)$$

$$= \mathbb{E}_{t-2\tau}\left\|\bar{\theta}_t - \theta^*\right\|^2 - \frac{\alpha(1-\gamma)\bar{\omega}}{2}\mathbb{E}_{t-2\tau}\left\|\bar{\theta}_t - \theta^*\right\|^2 - \frac{\alpha(1-\gamma)\bar{\omega}}{2}\mathbb{E}_{t-2\tau}\left\|\bar{\theta}_t - \theta^*\right\|^2$$

$$+ 16\alpha^2\mathbb{E}_{t-2\tau}\left\|V_{\bar{\theta}_t} - V_{\theta^*}\right\|_{\bar{D}}^2 + \frac{9+28\tau^2}{NK}\alpha^2 d_2^2 + \alpha^3 \left( 36L_2^2 + \frac{108\tau}{1-\rho^2} L_2^2 + 4L_1 G^2 + 2L_2 G \right)$$

$$+ \frac{\alpha^3 c_6}{K} \left[ c_3^2 + \frac{2c_3 L_2 \rho}{1-\rho} + 8c_1^2 (K-1)H^2 \right]$$

$$+ 4\alpha B(\epsilon, \epsilon_1)G + \alpha\nu_1 \Gamma^2(\epsilon, \epsilon_1)$$

$$\leq \mathbb{E}_{t-2\tau}\left\|\bar{\theta}_t - \theta^*\right\|^2 - \frac{\alpha(1-\gamma)\bar{\omega}}{2}\mathbb{E}_{t-2\tau}\left\|\bar{\theta}_t - \theta^*\right\|^2 - \frac{\alpha(1-\gamma)\bar{\omega}}{2}\mathbb{E}_{t-2\tau}\left\|V_{\bar{\theta}_t} - V_{\theta^*}\right\|_{\bar{D}}^2$$

$$+ 16\alpha^2\mathbb{E}_{t-2\tau}\left\|V_{\bar{\theta}_t} - V_{\theta^*}\right\|_{\bar{D}}^2 + \frac{9+28\tau^2}{NK}\alpha^2 d_2^2 + \alpha^3 \left( 36L_2^2 + \frac{108\tau}{1-\rho^2} L_2^2 + 4L_1 G^2 + 2L_2 G \right)$$

$$+ \frac{\alpha^3 c_6}{K} \left[ c_3^2 + \frac{2c_3 L_2 \rho}{1-\rho} + 8c_1^2 (K-1)H^2 \right]$$

$$+ 4\alpha B(\epsilon, \epsilon_1)G + \alpha\nu_1 \Gamma^2(\epsilon, \epsilon_1)$$

$$\overset{(b)}{\leq} \mathbb{E}_{t-2\tau}\left\|\bar{\theta}_t - \theta^*\right\|^2 - \frac{\alpha(1-\gamma)\bar{\omega}}{2}\mathbb{E}_{t-2\tau}\left\|\bar{\theta}_t - \theta^*\right\|^2 - \frac{\alpha(1-\gamma)\bar{\omega}}{4}\mathbb{E}_{t-2\tau}\left\|V_{\bar{\theta}_t} - V_{\theta^*}\right\|_{\bar{D}}^2$$

$$+ \frac{9+28\tau^2}{NK}\alpha^2 d_2^2 + \alpha^3 \left( 36L_2^2 + \frac{108\tau}{1-\rho^2} L_2^2 + 4L_1 G^2 + 2L_2 G \right)$$

$$+ \frac{\alpha^3 c_6}{K} \left[ c_3^2 + \frac{2c_3 L_2 \rho}{1-\rho} + 8c_1^2 (K-1)H^2 \right]$$

$$+ 4\alpha B(\epsilon, \epsilon_1)G + \alpha\nu_1 \Gamma^2(\epsilon, \epsilon_1)$$

$$\leq (1 - 2\alpha\nu_1)\mathbb{E}_{t-2\tau}\left\|\bar{\theta}_t - \theta^*\right\|^2 - \alpha\nu_1\mathbb{E}_{t-2\tau}\left\|V_{\bar{\theta}_t} - V_{\theta^*}\right\|_{\bar{D}}^2 + \frac{9 + 28\tau^2}{NK}\alpha^2 d_2^2$$

$$+ \alpha^3\left(36L_2^2 + \frac{108\tau}{1-\rho^2}L_2^2 + 4L_1G^2 + 2L_2G\right)$$

$$+ \frac{\alpha^3 c_6}{K}\left[c_3^2 + \frac{2c_3 L_2 \rho}{1-\rho} + 8c_1^2(K-1)H^2\right] + 4\alpha B(\epsilon, \epsilon_1)G + \alpha\nu_1\Gamma^2(\epsilon, \epsilon_1) \tag{77}$$

where $(a)$ is due to Lemma 3 and the selection of parameter; (b) is due to $16\alpha^2 \leq \frac{\alpha(1-\gamma)\bar{\omega}}{4}$. Rearranging the terms and using the fact $1 - 2\alpha\nu_1 \leq 1 - \alpha\nu_1$, we have:

$$\alpha\nu_1\mathbb{E}_{t-2\tau}\left\|V_{\bar{\theta}_t} - V_{\theta^*}\right\|_{\bar{D}}^2 \leq (1 - \alpha\nu_1)\mathbb{E}_{t-2\tau}\left\|\bar{\theta}_t - \theta^*\right\|^2 - \mathbb{E}_{t-2\tau}\left\|\bar{\theta}_{t+1} - \theta^*\right\|^2 + \frac{9 + 28\tau^2}{NK}\alpha^2 d_2^2$$

$$+ \alpha^3\left(36L_2^2 + \frac{108\tau}{1-\rho^2}L_2^2 + 4L_1G^2 + 2L_2G\right)$$

$$+ \frac{\alpha^3 c_6}{K}\left[c_3^2 + \frac{2c_3 L_2 \rho}{1-\rho} + 8c_1^2(K-1)H^2\right] + 4\alpha B(\epsilon, \epsilon_1)G + \alpha\nu_1\Gamma^2(\epsilon, \epsilon_1) \tag{78}$$

Then we finish the proof by dividing $\alpha$ on both sides. $\qquad\square$

With these Lemmas, we are now ready to prove Theorem 2.

### I.3   Proof of Theorem 2.

Given a fixed local step-size $\alpha_l \leq \frac{1}{4\sqrt{2}c_1(K-1)}$, decreasing effective step-sizes $\alpha_t = \frac{8}{\nu(a+t+1)} = \frac{8}{(1-\gamma)\bar{\omega}(a+t+1)}$, decreasing global step-sizes $\alpha_g^{(t)} = \frac{\alpha_t}{K\alpha_l}$ and weights $w_t = (a+t)$, we have:

$$\mathbb{E}\left\|V_{\tilde{\theta}_T} - V_{\theta_i^*}\right\|_{\bar{D}}^2 \leq \tilde{\mathcal{O}}\left(\frac{\tau^2 G^2}{K^2 T^2} + \frac{c_{\text{quad}}(\tau)}{\nu^2 NKT} + \frac{c_{\text{lin}}(\tau)}{\nu^4 KT^2} + \frac{B(\epsilon, \epsilon_1)G}{\nu} + \Gamma^2(\epsilon, \epsilon_1)\right) \tag{79}$$

*Proof.* We take the step-size $\alpha_t = \frac{8}{\nu(a+t+1)} = \frac{2}{\nu_1(a+t+1)}$ for $a > 0$. In addition, we define weights $w_t = (a+t)$ and define

$$\tilde{\theta}_T = \frac{1}{W}\sum_{t=1}^{T} w_t \bar{\theta}_t$$

where $W = \sum_{t=1}^{T} w_t \geq \frac{1}{2}T(a+T)$. By convexity of positive definite quadratic forms ($\lambda_{\min}(\Phi^T \bar{D}\Phi) \geq \bar{\omega} > 0$), we have

$$\nu_1\mathbb{E}\left\|V_{\tilde{\theta}_T} - V_{\theta^*}\right\|_{\bar{D}}^2$$

$$\leq \frac{\nu_1}{W}\sum_{t=1}^{T}(a+t)\mathbb{E}\left\|V_{\bar{\theta}_t} - V_{\theta^*}\right\|_{\bar{D}}^2$$

$$\leq \frac{\nu_1}{W}\sum_{t=1}^{2\tau-1}(a+t)\mathbb{E}\left\|V_{\bar{\theta}_t} - V_{\theta^*}\right\|_{\bar{D}}^2 + \frac{\nu_1}{W}\sum_{t=2\tau}^{T}(a+t)\mathbb{E}\left\|V_{\bar{\theta}_t} - V_{\theta^*}\right\|_{\bar{D}}^2$$

$$\leq \nu_1\frac{(2\tau-1)(a+2\tau-1)G^2}{W} + \frac{\nu_1}{W}\sum_{t=2\tau}^{T}(a+t)\mathbb{E}\left\|V_{\bar{\theta}_t} - V_{\theta^*}\right\|_{\bar{D}}^2$$

$$\overset{(76)}{\leq} \nu_1\frac{(2\tau-1)(a+2\tau-1)G^2}{W} + \frac{\nu_1(a+2\tau)(a+2\tau+1)G^2}{2W}$$

$$+ \frac{1}{W} \sum_{t=2\tau}^{T} \left[ \frac{(9+28\tau^2)d_2^2}{NK}(a+t)\alpha_t + (a+t)\alpha_t^2 \left( 36L_2^2 + \frac{108\tau}{1-\rho^2}L_2^2 + 4L_1 G^2 + 2L_2 G \right) \right]$$

$$+ \frac{1}{W} \sum_{t=2\tau}^{T} \frac{(a+t)\alpha^2 c_6}{K} \left[ c_3^2 + \frac{2c_3 L_2 \rho}{1-\rho} + 8c_1^2(K-1)H^2 \right]$$

$$+ \frac{1}{W} \sum_{t=2\tau}^{T} \left[ 4(a+t)B(\epsilon,\epsilon_1)G + (a+t)\nu_1 \Gamma^2(\epsilon,\epsilon_1) \right] \tag{80}$$

where $\left\| V_{\bar{\theta}_{2\tau}} - V_{\theta^*} \right\|_{\bar{D}}^2 \le G^2$. We know that $\frac{1}{W}\sum_{t=2\tau}^{T}(a+t)\alpha_t^2 \le \frac{1}{W}\sum_{t=1}^{T}(a+t)\frac{4}{\nu_1^2(a+t)^2} \le \frac{8\log(a+T)}{\nu_1^2 T^2}$ and that $\frac{1}{W}\sum_{t=2\tau}^{T}(a+t)\alpha_t \le \frac{4}{\nu_1 T}$. Plugging in these inequalities into Eq (80), we have:

$$\nu_1 \mathbb{E}\left\| V_{\bar{\theta}} - V_{\theta^*} \right\|_{\bar{D}}^2$$

$$\le \frac{3\nu_1(a+2\tau)(a+2\tau+1)G^2}{2W} + \frac{4(9+28\tau^2)d_2^2}{\nu_1 NKT}$$

$$+ \frac{8\log(a+T)}{\nu_1^2 T^2}\left( 36L_2^2 + \frac{108\tau}{1-\rho^2}L_2^2 + 4L_1 G^2 + 2L_2 G \right)$$

$$+ \frac{8c_6 \log(a+T)}{\nu_1^2 T^2 K}\left[ c_3^2 + \frac{2c_3 L_2 \rho}{1-\rho} + 8c_1^2(K-1)H^2 \right]$$

$$+ 4B(\epsilon,\epsilon_1)G + \nu_1 \Gamma^2(\epsilon,\epsilon_1)$$

$$= \frac{3\nu_1(a+2\tau)(a+2\tau+1)G^2}{2W} + \frac{4(9+28\tau^2)d_2^2}{\nu_1 NKT}$$

$$+ \frac{8\log(a+T)}{\nu_1^2 T^2 K}\underbrace{\left[ K\left( 36L_2^2 + \frac{108\tau}{1-\rho^2}L_2^2 + 4L_1 G^2 + 2L_2 G \right) + c_6\left( c_3^2 + \frac{2c_3 L_2 \rho}{1-\rho} + 8c_1^2(K-1)H^2 \right) \right]}_{c_{\text{lin}}(\tau)}$$

$$+ 4B(\epsilon,\epsilon_1)G + \nu_1 \Gamma^2(\epsilon,\epsilon_1) \tag{81}$$

where $c_{\text{quad}}(\tau) = 4d_2^2(9+28\tau^2)$. Dividing $\nu_1$ on the both sides, changing $\nu_1$ into $\nu$ ($\nu = (1-\gamma)\bar{\omega}$) and noting that $c_6 = \frac{4}{\alpha_g^2}(\frac{14}{\xi_1}+14\xi_1)(c_1+c_2) = \mathcal{O}(\frac{1}{\nu})$, we have:

$$\mathbb{E}\left\| V_{\tilde{\theta}_T} - V_{\theta^*} \right\|_{\bar{D}}^2 \le \tilde{\mathcal{O}}\left( \frac{\tau^2 G^2}{K^2 T^2} + \frac{c_{\text{quad}}(\tau)}{\nu^2 NKT} + \frac{c_{\text{lin}}(\tau)}{\nu^4 KT^2} + \frac{B(\epsilon,\epsilon_1)G}{\nu} + \Gamma^2(\epsilon,\epsilon_1) \right). \tag{82}$$

We finish the proof by using the inequality, $\mathbb{E}\left\| V_{\tilde{\theta}_T} - V_{\theta_i^*} \right\|_{\bar{D}}^2 \le 2\mathbb{E}\left\| V_{\tilde{\theta}_T} - V_{\theta^*} \right\|_{\bar{D}}^2 + 2\mathbb{E}\left\| V_{\theta_i^*} - V_{\theta^*} \right\|_{\bar{D}}^2$ and combining with the third point in Theorem 1. $\qquad\square$

## J    Simulation Results

### J.1    Simulation results for the I.I.D. setting

In this subsection, we provide numerical results for $\texttt{FedTD}(0)$ under the i.i.d. sampling setting to verify the theoretical results of Theorem 4. In particular, the MDP $\mathcal{M}^{(1)}$ of the first agent is randomly generated with a state space of size $n = 100$. The remaining MDPs are perturbations of $\mathcal{M}^{(1)}$ with the heterogeneity levels $\epsilon = 0.1$ and $\epsilon_1 = 0.1$. The number of local steps is chosen as $K = 20$. We evaluate the convergence in terms of the running error $e_t = \|\bar{\theta}_t - \theta_1^*\|^2$. Each experiment is run 10 times. We plot the mean and standard deviation across the 10 runs in Figure 3.

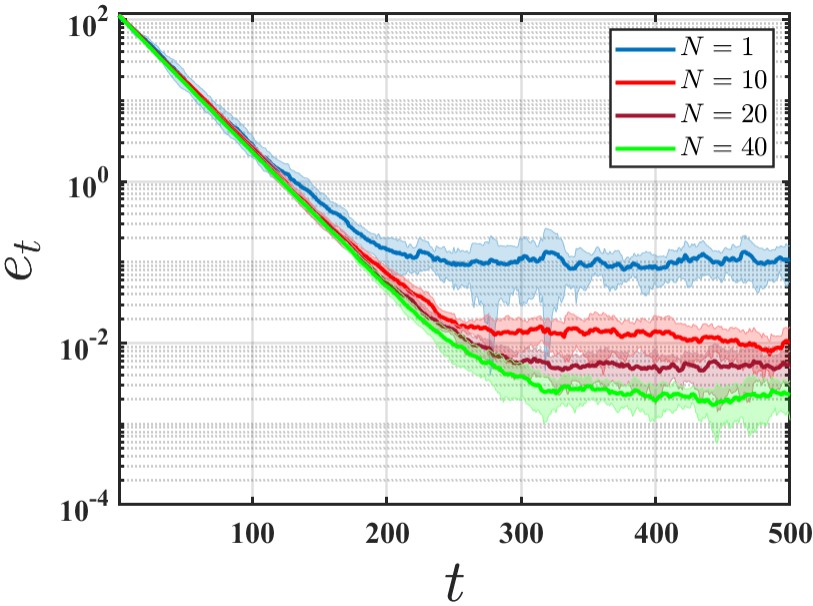

Figure 3: Performance of $\texttt{FedTD}(0)$ with i.i.d. sampling with varying number of agents $N$. Solid lines denote the mean and shaded regions indicate the standard deviation over ten runs.

As shown in Fig 3, $\texttt{FedTD}(0)$ converges for all choices of $N$. Larger values of $N$ decreases the error, which is consistent with our theoretical analysis in Theorem 4.

### J.2 Simulation results for the Markovian setting

In this subsection, we provide numerical results for `FedTD`(0) under the Markovian sampling setting to verify the theoretical results of Theorem 2. Here we generate all MDPs in the same way as the i.i.d setting and choose the number of local steps as $K = 20$. All the remaining parameters are kept the same as those in the subsection J.1.

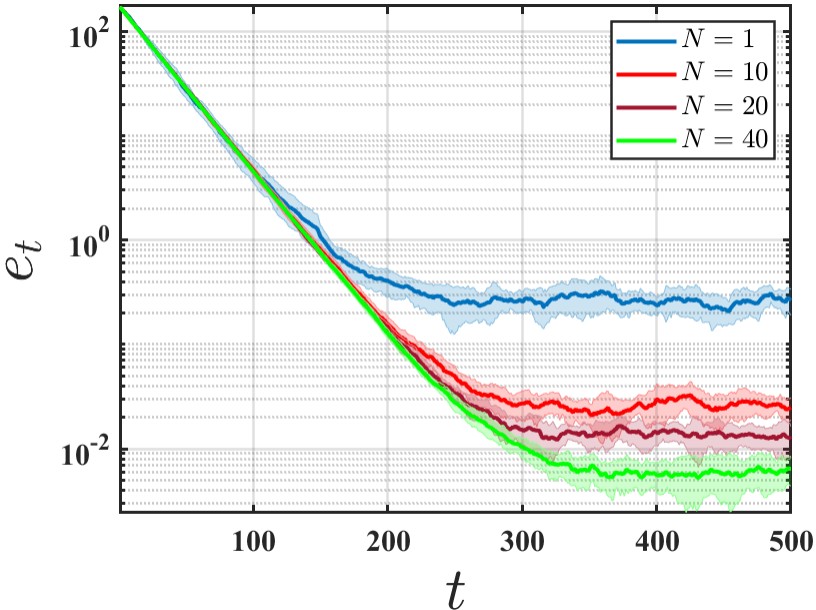

Figure 4: Performance of `FedTD`(0) with the Markovian sampling with varying number of agents $N$. Solid lines denote the mean and shaded regions indicate the standard deviation over ten runs.

As shown in Fig 4, `FedTD`(0) converges for all choices of $N$. Larger values of $N$ decreases the error, which is consistent with our theoretical analysis in Theorem 2.

### J.3 Simulation on the effect of the heterogeneity level for the Markovian setting.

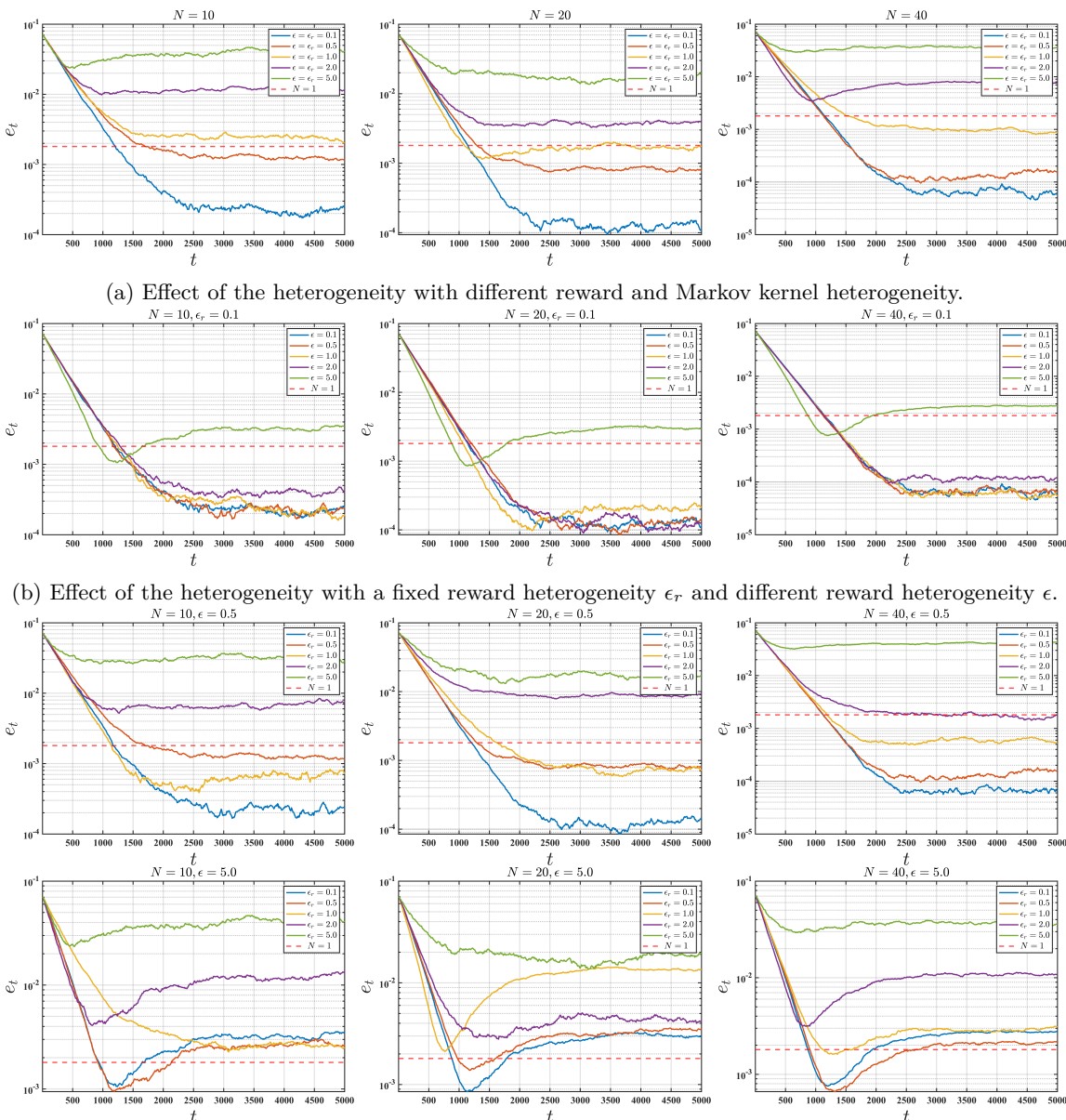

(a) Effect of the heterogeneity with different reward and Markov kernel heterogeneity.

(b) Effect of the heterogeneity with a fixed reward heterogeneity $\epsilon_r$ and different reward heterogeneity $\epsilon$.

(c) Effect of the heterogeneity with a fixed Markov kernel heterogeneity $\epsilon$ and different reward heterogeneity $\epsilon_r$.

As shown in Fig $(a) \sim (c)$, we can conclude that increasing the level of heterogeneity level will increase the size of the ball to which FedTD(0) converge, which is completely consistent with our theoretical analysis in Theorem 2.

