# OpenReview forum: "Federated TD Learning with Linear Function Approximation under Environmental Heterogeneity"
_TMLR — Accepted by TMLR_

### Review · Reviewer_nkSm · 2023-12-21

**Summary Of Contributions:**

The paper considers federated reinforcement learning under environmental heterogeneity, looking into the policy evaluation problem. The authors consider a model where N agents interact with environments sharing the same state and action space, but with different reward functions and state transition probabilities. In the model, the agents may communicate through a central server. The key question studied is whether exchanging information can speed up the process of evaluating a common policy. The authors hence provide a finite-time analysis of a (federated) temporal difference learning algorithms, applying linear function approximation. The method accounts for the sampling and heterogeneity in the agents’ environments.

**Audience:**

Yes

**Broader Impact Concerns:**

No concerns on my part.

**Claims And Evidence:**

Yes

**Requested Changes:**

Please first see abpve regarding better / more detailed presentation of the motivation behind the paper (and possible domains).

Second, in terms of the exposition, I’ve found Figure 1 very helpful. I suggest extending that description. Additionally, the list of contributions of the paper is very clear. However, I would have appreciated a more details introduction to the topic of Federated Learning and RL. Readers are likely to come from all sorts of backgrounds, and it would be great to give more details on this topic. In particular, I’d also suggest exploring some connections to multitask learning in more detail; I realize this is about federated RL, but you base model is having MDPs that are similar in state and action spaces, and the theme of using what is known about the setting to speed-up learning has been explored in a broader context.

**Strengths And Weaknesses:**

All in all, I really love the topic of the paper- a multiagent view on analyzing environments and policies that have some similarities and some differences. I think the model proposed by the authors is quite natural. Nonetheless, I think it could be better motivated in the paper. It’d be good for example to list down multiple possible domains where this could occur, and provide an example or two per each domain, possibly in the form of a table. For instance I can easily think of some domains (e..g supply chains, where customer demands and stock item transfers are held fixed, but the probabilities / quantities of demand or supply have different probability distribution, or traffic networks where the graph topology is fixed, but the travel times come from a different distribution).

The proposed method combines several building blocks in a creative  way: perturbation bounds on TD fixed points depending on the differences in the underlying MDPs, using a “virtual” MDP to approximate the dynamics of the federated algorithm, and using the “virtual” MDP to connect to federated optimization. A key result in the paper is that for low-heterogeneity regimes, exchanging model estimates leads to linear convergence speedups in the number of agents.

All in all, a very interesting paper on an important topic. Presentation could be improved a bit, but the paper is readable and clear (with a lot of details left in the later sections, which I think is a good call).

---

> ### Author Response · Authors · 2024-03-08
> **Response to the motivating examples**
>
> Thank you for your constructive suggestions on our paper. In response to your suggestions, we have now included a fairly elaborate discussion on the diverse possible applications (e.g., supply chains, transportation, health care, and recommendation systems) of our FRL framework in the introduction of our revised paper.

---

> ### Author Response · Authors · 2024-03-08
> **Response to the relative works**
>
> Thank you again for your very useful suggestions. In response to these suggestions, we have now made two main changes to our revised paper. First, we have now added an informal description of a standard RL framework in the introduction - we believe this description will provide the readers with the context needed to understand the motivation behind our study. Second, under the related works section, namely Section 1.2, we have now discussed relevant work on multi-task learning. However, as we mention in the paper, these papers neither consider iterative stochastic approximation under Markovian sampling nor the effects of intermittent communication as captured by local steps. As such, our work differs considerably from the multi-task literature in terms of both the algorithms we study, and our analysis techniques.

---

### Review · Reviewer_N4hB · 2023-12-21

**Summary Of Contributions:**

This work focuses on federated RL, in particular the TD learning task with linear function approximation. The setting involves $N$ agents, each with its own environment (with heterogeneous transition kernels and rewards) to interact, and seek to evaluate a shared policy by communicating with a central server. The main results establishes an upper bound of the estimation error, that involves (i) a non-vanishing bias term due to environmental heterogeneity and (ii) vanishing terms that features a linear speedup with the number of agents. Numerical results are provided to verify the findings.

**Audience:**

Yes

**Claims And Evidence:**

Yes

**Requested Changes:**

See Strengths And Weaknesses.

**Strengths And Weaknesses:**

Overall the paper is well written, organized and easy to follow. The explanation of technical contributions is clear and intuitive. However, some implications of the results are not immediately clear to me, and it would be much appreciated if the authors can provide some further discussions on them:

1. What would be a reasonable choice of the number of local updates $K$, given that the total number of iterates $KT$ is fixed? The ideal choice of $K$ should be as large as possible without significantly worsening the error bound, so as to promote communication efficiency.

2. The bias term from environmental heterogeneity appears to scale quite pessimistically with $\epsilon$ and $\epsilon_1$, and involves some terms that are hard to interpret (e.g., the conditional number of $\bar{A}_i$). Would it be possible to derive the bound with terms such as $\\|V\_{\theta_i^\star} - V\_{\theta_j^\star}\\|$ instead?

3. The first experiment investigates the effect of linear speedup with different number of agents. It is mentioned that the first MDP is randomly generated and the remaining MDPs are perturbations of the first one. If they are generated in an i.i.d. random way, then the improvement may actually results from the benign statistics property that the virtual MDP is converging to the first MDP.

---

> ### Author Response · Authors · 2024-03-08
> **Response to Q1**
>
> Thank you for raising this important point of selecting the number of local steps $K$ judiciously. While it might appear that $K$ can be made arbitrarily large without worsening the bounds, this is not quite true. Let us elaborate on this point. First, recall that as per our notation, $N$ represents the number of agents, $T$ represents the number of communication rounds, and $K$ represents the number of local steps per round. Thus, the total number of samples collected per agent is $KT$, and overall across the system is $NKT$. Ideally, one would hope for the mean-square error to scale inversely with $NKT$ - our main result Theorem 2 does indeed exhibit such a dominant $O(1/(NKT))$ term up to an additive (unavoidable) heterogeneity bias. Although this dominant term goes down with $K$, there is an additional (eventually negligible) term on the order of $O(1/T^2)$ that also shows up in our final bound.  Now suppose we fix $KT=R$. Our overall bound would then take the following form:
> $$ \textrm{Mean-Square Error} \leq \tilde{O}\left(\frac{1}{NR} \right) + \tilde{O}\left(\frac{K^2}{R^2} \right) + \textrm{Heterogeneity Bias} .$$
>
> The above bound clearly captures the price paid for increasing $K$: we see that it causes the second term on the R.H.S. to increase. Although this term will eventually get dominated by the first term for sufficiently large $R$, we note that $K$ cannot be increased indefinitely. The main takeaway here is that although increasing $K$ helps reduce communication costs and the dominant term in our bound, it also exacerbates a higher-order term. Thus, there is a price to be paid for infrequent communication, and our bounds reflect this trade-off. At a more technical level, the cause of this trade-off arises from the fact that the local step-sizes/learning rates at the agents need to scale inversely with the number of local steps $K$ to ensure that the drift between agents' parameters do not blow up during the period when there is no communication.
>
> In addition to the above, it is also worth pointing out that although one can think of $K$ as a hyper-parameter to be tuned to tweak algorithm performance, in reality, $K$ will typically be dictated by the available resources of each client.
>
> In the revised paper, we have now added a brief discussion of the above points.

---

> ### Author Response · Authors · 2024-03-08
> **Response to Q3**
>
> Thanks for asking this interesting question. The remaining MDPs are indeed generated in an i.i.d way. However, there are two key points we would like to make.
>
> First, the scale of our experiment, involving a total number of MDPs/agents ($N$) up to 30 or 40, is relatively small. This scale is insufficient to trigger the Law of Large Numbers, which necessitates a significantly larger sample size to ensure that the empirical average of the results obtained from many trials converges to the expected value. As such, it is quite unlikely that the virtual MDP has converged to a vicinity of the first MDP in our experiments.
>
> Second, for the sake of argument, suppose that the transition kernels and reward functions of the virtual MDP are indeed $\epsilon$-close to those of the first MDP. Crucially, note that this closeness pertains to the MDP parameters, **not the optimal TD parameters**.  As elaborated in response to a previous question you raised, each agent $i$'s TD(0) fixed point is determined by the equation $\bar{A}^{(i)}\theta_i^* = \bar{b}^{(i)}$, where $\bar{A}^{(i)}$ and $\bar{b}^{(i)}$ are specific to each MDP. Even minor perturbations in $\bar{A}^{(i)}$ and $\bar{b}^{(i)}$, especially under conditions of ill-conditioning, can lead to significant disparities in the agents' TD(0) fixed points because of the inverse calculation. In short, even if the Law of Large numbers does kick in, and the virtual MDP does happen to be $\epsilon$-close to the first MDP, this does not immediately imply that the heterogeneity bias - arising due to differences in TD(0) fixed points - will be small.
>
> Thus, we do not think that our current experimental results undermine the key messages we wish to convey. That said, if the Reviewer wishes to see some other experiments, we are happy to provide them.

---

> ### Author Response · Authors · 2024-03-08
> **Response to Q2**
>
> Thank you for bringing up this important question. In what follows, we will first argue that a heterogeneity-induced bias term - of the form that shows up in Theorem 2 - is unavoidable, in general. Next, we will speak to the specific nature of this term.
>
> First, some intuition. Notice that the bound we provide in Theorem 2 captures the expected value-function suboptimality between the output of our FedTD algorithm $\bar{\theta}_T$ and the optimal parameter $\theta^*_i$ of a particular agent $i$. Now given that our algorithm combines data from the MDPs of every agent, and these MDPs are non-identical, there is no reason to expect that the output $\bar{\theta}_T$ of our algorithm will converge to the optimal parameter of any particular agent $i$. Thus, it stands to reason that a heterogeneity bias term - akin to what shows up in Theorem 2 - is only to be expected. Indeed, this reasoning is formalized in Theorem 3 where we show that a bias term that depends on the difference between the agents' optimal parameters is unavoidable for an algorithm like FedTD.
>
> Now that we have argued the inevitability of the bias term in Theorem 2, let us get into the specifics of this term. The TD learning setting we consider is an instance of linear stochastic approximation where the optimal parameter $\theta_i^*$ for each agent $i$ is the unique solution of the following linear equation $\bar{A}^{(i)} \theta_i ^*= \bar{b}^{(i)}$. Here, $\bar{A}^{(i)}$ and $\bar{b}^{(i)}$ depend on the probability transition kernels and reward functions, respectively, of agent $i$'s MDP; these objects are defined in Section 3 of our paper. In light of Theorem 3, we now know that a bias term depending on the magnitude of the difference $\max_{i,j \in [N]} \Vert \theta_i^* - \theta_j ^*\Vert$ is unavoidable. But quantifying the above difference essentially boils down to doing a perturbation analysis of the solutions of linear equations. Why? Well, fixing a pair of agents $i$ and $j$, one can view $\bar{A}^{(j)}$ and $\bar{b}^{(j)}$ as perturbed versions of $\bar{A}^{(i)}$ and $\bar{b}^{(i)}$, where these perturbations arise from the underlying differences in the agents' probability kernels and reward functions.
>
> In this context, we note that the condition number (ratio of max/min singular values) is the gold standard for assessing the sensitivity of solutions of $Ax=b$ to perturbations of the form $(A+\Delta)x = b$. Further details can be found in the classical texts by Trefethen & Bau and Golub and Van Loan. In light of the above discussion and Theorem 3, we conclude that the appearance of the condition number of $\bar{A}_i$ in our heterogeneity bias term is in fact not surprising. Please let us know if the above train of thought makes sense.
>
> In passing, we also note that one of the key contributions of our work (Theorem 1) is to map the closeness between the agents' MDPs, i.e., the transition kernels and reward functions, to the closeness between the limiting behavior of TD(0) on such MDPs (as characterized by the agents' optimal parameters).
>
> Finally, let us turn to the question you raised about expressing our bounds in terms of $\Vert V_{\theta_i^*} - V_{\theta_j^*} \Vert.$ Yes, one can very easily convert any bound on the iterates to a corresponding bound on the value functions via a norm-equivalence lemma - namely, Lemma 1 in Bhandari et al., COLT 2018. We briefly explain this lemma here by following the notation in Section 3 of our paper. Recall that $\Phi$ is the feature matrix and $D^{(i)}$ is the diagonal matrix containing the elements of the stationary distribution $\pi^{(i)}$ along the diagonal. Let $\Sigma^{(i)}= \Phi^{\top} D^{(i)} \Phi.$ Now under feature normalization and Assumption 3 in our paper, we have $0 <  \omega_i \triangleq \lambda_{min}(\Sigma^{(i)}) \leq \lambda_{max}(\Sigma^{(i)}) \leq 1.$  Using $\hat V_{\theta} =\Phi \theta,$ we then have for any $\theta_1, \theta_2:$
> $$ \Vert \hat V_{\theta_1} - \hat V_{\theta_2} \Vert^2_D = (\theta_1-\theta_2)^{\top} \Sigma^{(i)} (\theta_1-\theta_2).$$ Thus, we immediately obtain:
> $$ \omega_i \Vert \theta_1 - \theta_2 \Vert^2_2 \leq \Vert  \hat V_{\theta_1} - \hat V_{\theta_2} \Vert^2_D \leq \Vert \theta_1 - \theta_2 \Vert^2_2.$$
>
> In light of the above display, it is straightforward to go back and forth between parameter-sub-optimality bounds and value-function sub-optimality bounds.
>
> In the revised paper, we have now added a brief discussion of these points.

---

### Review · Reviewer_By6e · 2024-02-15

**Summary Of Contributions:**

This paper introduces and analyzes FedTD(0)--an algorithm for federated policy evaluation.  Under the proposed framework, $N$ agents each take actions in individual MDPs according to a common policy $\mu$.  The MDPs share the same finite state and action structure, but may have different rewards and transition probabilities.  The goal of an agent is to evaluate its own policy, i.e., estimate its own state-value function.

When employing FedTD(0), each agent perform $K$ steps of TD(0) with linear function approximation starting with the same state-value parameters.  After $K$ steps, each agent broadcasts the difference between their current state-value parameters and the common starting parameters.  Then the agents derive new common state-value parameters by averaging the differences and begin a new round.

The paper proves that so long as the individual MDPs are similar enough that FedTD(0) can in expectation achieve a linear speedup with respect to the number of agents.  This result in and of itself is not surprising, e.g., if all the agents act in the same MDP then they effectively all perform $(N-1)*K$ additional updates per round.  What is much more nuanced though is the analysis of how the performance of FedTD(0) degrades as the MDPs start to differ.

More concretely, the paper assumes that each MDP has long-term dynamics that result in a unique steady-state distribution.  First, it shows that so long as the transition probabilities are relatively within $\varepsilon$ for all pairs of agents and pairs of states that the L1-norm of the difference in steady-state distributions is bounded by $O(n\varepsilon)$.

Second, it shows so long as the L2-norm of the difference between any two agents' MRP reward vectors is bounded by $\varepsilon_1$, then the L2-norm of the difference between their optimal state-value function parameters is also bounded.

This leads to the development of FedTD(0), which can be roughly thought of as having the agents' collaboratively estimating the state-value function in the "average" MDP until the very last round where each individual estimates its own state-value function from a common starting point.  So long as the individual MDPs are pair-wise close, then they will be close to the average and this strategy can be advantageous.

**Audience:**

Yes

**Broader Impact Concerns:**

No concerns.

**Claims And Evidence:**

Yes

**Requested Changes:**

Though it may appear by the relative length of my strengths and weaknesses lists that I am not in favor of accepting this paper, I do believe that it can be dramatically improved with appropriate textual additions.

Specifically, the addition of concrete examples of what constitutes settings that are favorable for FedTD(0) and what settings are not and why.  e.g., what constitutes a good heterogeneous population of agents for different structures of MDPs, and how quickly the performance of FedTD(0) degrade.  This is empirically explored to an extent in Section 5.1., but that can be complemented greatly by exploring how agent populations and MDP structures effect the terms in the bound and how well the bound predicts the performance in practice.

Another particular point of interest is discussing how should a group of agents decide to participate in FedTD(0), what step-sizes that they should use, how to combine the weights into a solution for each agent, or in what situations can an agent identify that it is not advantageous for them to continue cooperating in the FedTD(0) process.

**Strengths And Weaknesses:**

Starting with the strengths, though the math is fairly involved due to the mixing/long-term nature of the MDPs, the idea itself is reasonably straightforward and nicely built up from a perturbation analysis of stationary distributions and linear systems.  The algorithm structure itself is also intuitive and operationally it is a very simple extension of TD(0).

The paper has a number of weaknesses, but I hope that most of them can be mitigated with textual changes.

First, in my opinion the paper does over claim and misrepresent some of its contributions.  An example of this is the statement "Theorem 2 is significant in that it is the first result of its kind in MARL/FRL with heterogeneous environments...".  I do not view this setting as multi-agent RL as an agent's actions has no effect the others' rewards.  In a sense, the agents in this setting do not even act as the shared policy is predetermined, and all must update and communicate their state-value parameters exactly as prescribed by the algorithm.  Additionally, the analysis of the algorithm implies that its performance only trends favorably when scaling the number of agents in the population stays relatively homogeneous.  Another example, it is also not totally clear how to apply FedTD(0) to actually perform RL.  Where as in the single-agent setting state-value estimation is a critical component of learning good policies, the agents in this setting cannot independently adjust their policy if they wish to continue participating in the shared policy evaluation.

Second, though the relative error bound on the probability transitions and norm-bound on the corresponding MRPs' rewards are easy to describe mathematically, it is hard to imagine situations where multiple agents will act in the exact same state/action-space where these components differ slightly.  Some concrete examples of applications would be helpful.

Third, related to the most recent point, it is also difficult to follow exactly how the error bounds are related to the assumed probability/reward bounds.  In particular, the Eigenvalue/condition numbers of the matrices corresponding to the MDPs relate to non-intuitive structural aspects, and exactly how these combine when analyzing the "average" MDP is hard to imagine.

Fourth, the algorithm as stated is to some extent unimplementable in that the step-size parameters are, as far as I can tell, a function of the steady-state distribution of the average MDP.  i.e., it seems that the agents' would need to perform a non-trivial collaborative computation to first agree on the step-sizes.

Finally, if my understanding is correct it is rather detrimental that the performance of the agents as a whole is only guaranteed to be as good as that of the two most dissimilar agents.  i.e., if there are N-1 agents acting in the same MDP and one acting in a very dissimilar MDP then all agents might be better off acting alone.

---

> ### Author Response · Authors · 2024-03-15
> **Response to the comments on not surprising results**
>
> We respectfully disagree with the Reviewer's view that our results are obvious and not surprising. We agree that such behavior is desirable - but proving it was a huge undertaking. Even in the more simple homogeneous setting [2], proving the linear speedup result took nearly $30-40$ pages of rigorous analysis. Notably, this work focusing on deriving this ''unsurprising'' result was selected for a long presentation at ICML 2022.
>
> There are many technical challenges in our settings that had to be overcome. Why?  Establishing the scaling of our results with $N$ and $K$ is largely dependent on the variance reduction lemma 13, i.e., the variance of estimating the gradients scales with the increase in both agents $N$ and local updates $K$. First, our problem setup involves *temporally correlated Markovian data from heterogeneous MDPs*, which adds layers of complexity to the analysis. The complexities that such sampling creates in the **centralized** setting are clearly explained in the seminal paper [Bhandari, Russo and Singal]. In addition, in our setting, we also have inter-agent correlations among local parameters ($\theta_{t,k_1}^{(i)}$ and $\theta_{t,k_2}^{(i)}$). This is in contrast to the standard assumptions common to the distributed/FL optimization literature (e.g. FedAvg and FedProx), where gradient noise is considered independent across agents and updates, our scenario complicates the derivation of this variance reduction lemma. The presence of correlated observations at different local steps ($O_{t,k_1}^{(i)}$ and $O_{t,k_2}^{(i)}$) for any agent does not make it obvious that variances can be scaled with $K$. One might argue that leveraging the geometric mixing property could address these correlations, as it predicts a geometric decay in correlation post-mixing time. Yet, this does not clarify the scenario when the number of local updates $K$ is less than the mixing time, leaving a significant correlation between observations. Despite these uncertainties, we have demonstrated that Lemma 13 remains applicable in the Markov setting, without necessitating any specific conditions regarding the relationship between $K$ and the mixing time. Second, it was far from obvious whether the variances or the final convergence results can be scaled by the number of agents $N$--linear speedup. One might mention that it is easy to imagine the result of scaling with $N$ since the observations are independent across agents. However, our scenario, marked by environmental heterogeneity, necessitates the employment of server-side projection [line 10 in Algorithm 1] to mitigate the significant bias in estimating the value function across all agents. This necessary step introduces the possibility of disrupting the expected linear speedup. Our findings were achieved by overcoming these intricate challenges.
>
> We list the differences and novelty between our work and existing work as below:
>
> **Novelty Relative to [1, 2, 3]**:
>
> *  [1] only considered a homogeneous setting under i.i.d. sampling, i.e., identical MDPs. Therefore, there was no need for them to address the challenge of significant statistical correlation among parameters ($\theta_{t,k_1}^{(i)}$ and $\theta_{t,k_2}^{(i)}$) at different time steps, which arises from multiple local updates and Markovian sampling. More crucially, they did not need to tackle the concerns regarding environmental heterogeneity. In other words, they did not need to figure out whether bounded environmental heterogeneity implies the bounded distance between each agent's TD fixed point, which we proved in Theorem 1.
>
> * [2] considered Markovian sampling, the analysis is once again limited to a homogeneous setting. As we mentioned before, they did not have to worry about the effect of environmental heterogeneity.
>
> * [3] considered environmental heterogeneity, their analysis is limited to a much simpler tabular setting without function approximation. Even in this simple setting, [3] is unable to prove a linear speedup result, i.e., scaled by $K$, (unlike our result which does establish a linear speed-up in the more complicated case).
>
> We reiterate that \textit{ our work is the first to establish a linear speedup result under function approximation, multiple local updates, Markovian sampling, and environmental heterogeneity}. Regarding the first question, about our result being obvious, the papers listed above establish weaker results with stronger assumptions. We believe this is further evidence as to why our result is not obvious.
>
> [1] "Finite-time analysis of distributed TD (0)
> with linear function approximation on multi-agent reinforcement learning". --Doan et al., ICML 2019.
>
> [2] "Federated Reinforcement Learning: Linear Speedup Under Markovian Sampling" -- Khodadadian et al., ICML 2022.
>
> [3] "Federated Reinforcement
> Learning with Environment Heterogeneity" --Jin et al., AISTATS 2022.

---

> ### Author Response · Authors · 2024-03-15
> **Response to comments on usage of "Multi-agent"**
>
> We believe that there may have been some confusion induced by us including the term "multi-agent'' in this work. Our setting is different from another line of multi-agent RL problems, where each agent interacts with each other and takes action based on others' states, actions, or rewards. Instead, in our setting, each agent operates independently in their own environment and has no interaction with each other. We adopted the term "multi-agent" in alignment with existing literature [1], where agents' actions also do not impact other agents' rewards or observations, with each agent drawing observations from their environment. To clarify and avoid further confusion, we will revise our terminology. Specifically, we will eliminate the term "multi-agent" from our paper and instead describe our contribution as a novel achievement within the FRL domain.
>
> [1] "Finite-time analysis of distributed TD (0) with linear function approximation on multi-agent reinforcement learning". --Doan et al., ICML 2019.

---

> ### Author Response · Authors · 2024-03-16
> **Response to comments on how to adjust each agent's policy**
>
> Indeed, we do not expand upon this point further in our work since a comprehensive analysis of the downstream effects of fine-tuning/personalization is beyond the scope of our current paper. We should note here that results on fine-tuning even in the federated supervised learning setting have appeared only very recently in 2022 (Ref [4]), *almost 6 years after the first FL paper came out!* In contrast, federated RL (FRL) is a nascent field, the first theoretical papers in this area are from last year, and our work is one of the first principled efforts towards studying the challenge of heterogeneity in FRL. So it will naturally take some time to have a better sense of personalization in RL.
>
> Now let us get back to the question of whether FedTD(0) + fine-tuning can help beyond local learning. To begin to understand this question, let us note that our final sub-optimality gap for each agent is of the form $O\left(\frac{1}{NKT}\right) + O(\epsilon, \epsilon_1)$, where $N$ is the number of agents, $K$ is the number of local steps, $T$ is the number of rounds, and the $O(\epsilon, \epsilon_1)$ term captures the effect of kernel and reward heterogeneity collectively. The first term is *statistically optimal* (since the total number of samples across agents is precisely $NKT$), and the second term is *fundamentally unavoidable* (as shown in Theorem 3). Thus, our overall guarantee is the best one can hope for. Our final bound as discussed above is entirely consistent with results in FL where the global loss function $f(\mathbf{x})$ differs from each agent $i$'s local loss function $f_i(\mathbf{x})$ by $O(\epsilon)$.
>
> There are two main takeaways from the above discussion.
>
> *  First, since arriving at the above result requires no assumption/knowledge of the heterogeneity levels $\epsilon, \epsilon_1$, our final bound captures the true nature of the problem. If $O(\epsilon, \epsilon_1)$ is large relative to the first statistical term, then there is nothing to be gained from federation; local learning makes more sense. This seems intuitive. On the other hand, if $O(\epsilon, \epsilon_1)$ is small relative to the statistical term, then the output of FedTD(0), namely $\hat{\theta}_T$, can indeed provide a much better approximation of an agent $i$'s optimal point $\theta_i^*$ relative to what it could obtain by learning locally. Personalization can only help from this point onwards: this is because if the agent $i$ starts from $\hat{\theta}_T$ and now uses *only its own data*, the $O(\epsilon, \epsilon_1)$ bias term that shows up in Theorem 2 will only affect the final bound as an "initial condition effect'' - this effect can be washed away as agent $i$ uses more local samples.
>
> If there is no prior understanding of whether the $O(\epsilon, \epsilon_1)$ term is large or small, then we don't have a current sense of what can be done. Perhaps one can aim to somehow adaptively cluster agents into groups such that within each group, $O(\epsilon, \epsilon_1)$ is small.
>
> * Second, as we mentioned earlier, our final bound is what one typically achieves in heterogeneous FL as well. So techniques/results for fine-tuning in FL should likely have implications for our setting as well. In this regard, one could appeal to the conclusions from the following interesting paper:
>
> [4] ``Federated Asymptotics: a model to compare federated learning algorithms'' - Cheng et al., AISTATS 2023.

---

> ### Author Response · Authors · 2024-03-16
> **Response to comments on the motivating examples**
>
> Thank you for your constructive suggestions on our paper. In response to your questions, we have now included a fairly elaborate discussion on the diverse possible applications (e.g., supply chains, transportation, health care, and recommendation systems) of our FRL framework in the introduction of our revised paper.

---

> ### Author Response · Authors · 2024-03-16
> **Response to the question about the non-intuitive structural**
>
> In response to the question on how the error bounds being related to the assumed probability/reward bounds, the main takeaway is that the term $\Gamma(\epsilon, \epsilon_1)$ in Theorem 1 will linearly depend on the heterogeneity constant $\epsilon$ and $\epsilon_1$ (up to some negligible high-order terms) as long as $\bar{A}_i$ is well-conditioned. In other words, the distance between each agent's TD fixed point is directly proportional to the heterogeneity in the Markov Kernel $\epsilon$ and the reward heterogeneity $\epsilon_1$. Likewise, $Q(\epsilon, \epsilon_1)$ in Theorem 2 is linearly dependent on the heterogeneity constants $\epsilon$ and $\epsilon_1.$
>
> Next, let us try to understand the condition numbers in Theorem 1. The TD learning setting we consider is an instance of linear stochastic approximation where the optimal parameter $\theta_i^*$ for each agent $i$ is the unique solution of the following linear equation $\bar A_i \theta_i^* = \bar b_i$. Here, $\bar A_i$ and $\bar b_i$ depend on the probability transition kernels and reward functions, respectively, of agent $i$'s MDP; these objects are defined in Section 3 of our paper. In light of Theorem 3, we now know that a bias term depending on the magnitude of the difference $\max_{i,j \in [N]} \Vert \theta_i^* - \theta_j^* \Vert$ is unavoidable. But quantifying the above difference essentially boils down to doing a perturbation analysis of the solutions of linear equations. Why? Well, fixing a pair of agents $i$ and $j$, one can view $\bar{A}_j$ and $\bar{b}_j$ as perturbed versions of $\bar{A}_i$ and $\bar{b}_i$, where these perturbations arise from the underlying differences in the agents' probability kernels and reward functions.
>
>
> In this context, we note that the condition number (ratio of max/min singular values) is the gold standard for assessing the sensitivity of solutions of $Ax=b$ to perturbations of the form $(A+\Delta)x = b$. Further details can be found in the classical texts by Trefethen \& Bau and Golub and Van Loan. In light of the above discussion and Theorem 3, we conclude that the appearance of the condition number of $\bar{A}_i$ in our heterogeneity bias term is in fact not surprising. Please let us know if the above train of thought makes sense.

---

> ### Author Response · Authors · 2024-03-16
> **Response to the question about how to choose the step-size**
>
> First, the requirement in terms of the step-sizes follows the standard optimization literature, where the step-size is chosen to be smaller than the reciprocal of the Lipschitz constant. This requirement is a common and unavoidable condition for ensuring convergence in optimization problems (either supervised problems or RL problems).  Secondly, in reality, we do not need to know this constant. In experimental setups, it's very easy to select sufficiently small step-sizes that satisfy this condition, such as using time-decaying step-sizes. This approach simplifies the implementation and avoids the need for complex initial computations among agents to agree on step-sizes.

---

> ### Author Response · Authors · 2024-03-16
> **Response to the question about the heterogeneity**
>
> We believe that there may be a misunderstanding here. In Theorem 2, we establish a performance boundary applicable to every agent individually. Specifically, the optimality condition we utilize, represented on the left-hand side of Eq (2), measures the largest deviation between the output of our algorithm $\bar{\theta}$ and each agent's optimal $\theta_i^*$. Consequently, the right-hand side of Eq. (2) is based on the maximal distance among the MDPs of the agents, which is quite intuitive. Our analysis does not conclude that our algorithm's performance significantly deviates from the target outcomes of the $N-1$ agents operating in similar MDPs.
>
> Regarding the scenario where most agents share a similar MDP while one operates in a markedly different one, our current frameworks do not offer an affirmative conclusion. Our initial guess is that we could potentially use some anomaly detection schemes on the server side or clustering algorithms to ensure that the algorithm's output converges to the TD fixed points of these similar MDPs. It is a very interesting problem but we will leave it for future work.

---

### Decision · Action_Editor_DLEY · 2024-05-01

**Recommendation:** Accept as is

**Comment:**

Overall two reviewers recommended an accept and one reviewer recommended a reject. The primary reasoning for the reviewer recommending a reject is that the paper does not provide enough practical scenarios where the development in the paper will be useful. While this is true, I believe the paper still meets the criteria of TMLR in terms of correctness of results and there being some folks in the community who would be interested in the paper. The reviewers all seem to agree on this criteria.

**Audience:**

Individuals in the intersection of RL and Federated Learning.

**Claims And Evidence:**

The claims made in the paper are somewhat well supported. The reviewers raise an issue where they feel the paper oversells its contributions however on a technical level the paper seems correct.